# Strand-resolved mutagenicity of DNA damage and repair

Craig J. Anderson[1], Lana Talmane[1], Juliet Luft[1], John Connelly[1,2,3,4], Michael D. Nicholson[5], Jan C. Verburg[1], Oriol Pich[6], Susan Campbell[1], Marco Giaisi[7], Pei-Chi Wei[7], Vasavi Sundaram[8], Frances Connor[9], Paul A. Ginno[10], Takayo Sasaki[11], David M. Gilbert[11], Liver Cancer Evolution Consortium*, Núria López-Bigas[6,12,13,14], Colin A. Semple[1], Duncan T. Odom[9,10 ✉], Sarah J. Aitken[2,9,15,16 ✉] & Martin S. Taylor[1 ✉]

DNA base damage is a major source of oncogenic mutations[1]. Such damage can produce strand-phased mutation patterns and multiallelic variation through the process of lesion segregation[2]. Here we exploited these properties to reveal how strand-asymmetric processes, such as replication and transcription, shape DNA damage and repair. Despite distinct mechanisms of leading and lagging strand replication[3,4], we observe identical fidelity and damage tolerance for both strands. For small alkylation adducts of DNA, our results support a model in which the same translesion polymerase is recruited on-the-fly to both replication strands, starkly contrasting the strand asymmetric tolerance of bulky UV-induced adducts[5]. The accumulation of multiple distinct mutations at the site of persistent lesions provides the means to quantify the relative efficiency of repair processes genome wide and at single-base resolution. At multiple scales, we show DNA damage-induced mutations are largely shaped by the influence of DNA accessibility on repair efficiency, rather than gradients of DNA damage. Finally, we reveal specific genomic conditions that can actively drive oncogenic mutagenesis by corrupting the fidelity of nucleotide excision repair. These results provide insight into how strand-asymmetric mechanisms underlie the formation, tolerance and repair of DNA damage, thereby shaping cancer genome evolution.

There is an elegant symmetry to the structure and replication of DNA, in which the two strands separate and each acts as a template for the synthesis of new daughter strands. Despite this holistic symmetry, many activities of DNA are strand asymmetric: (1) during replication, different enzymes mainly synthesize the leading and lagging strands[3,4,6,7], (2) RNA transcription uses only one strand of the DNA as a template[8], (3) one side of the DNA double helix is more associated with transcription factors[9], and (4) alternating strands of DNA face towards or away from the nucleosome core[10,11]. These processes can each impart strand asymmetric mutational patterns that reflect the cumulative DNA transactions of the cells in which the mutations accrued[1,9,10,12,13].

Cancer genomes are the result of diverse mutational processes[1,14], often accumulated over decades, making it challenging to identify and subsequently interpret their relative roles in generating spatial and temporal mutational asymmetries. The relative contribution of DNA damage, surveillance and repair processes to observed patterns

of mutational asymmetry remains poorly understood, although mapping of DNA damage[15–18] and repair intermediates[19,20] have provided key insights.

To understand the mechanistic asymmetries of DNA damage and repair on a genome-wide basis, we have exploited an established mouse model of liver carcinogenesis[21,22], in which mutations are induced through a single DNA-damaging exposure to diethylnitrosamine (DEN; an alkylating agent that is bioactivated by the hepatocyte-expressed enzyme Cyp2e1). The exposure results in mutagenic DNA base damage, referred to as DNA lesions, that are inherited and resolved as mutations in subsequent cell cycles[2]. This phenomenon of lesion segregation, in which damaged lesion-containing strands segregate into separate daughter cells, results in pronounced, chromosome-scale mutational asymmetry. In a clonally expanded cell population, such as a tumour, this asymmetry can identify which damaged DNA strand was inherited by the ancestor of each tumour (Fig. 1a). Using this approach, we can

[1]Medical Research Council Human Genetics Unit, Institute of Genetics and Cancer, University of Edinburgh, Edinburgh, UK. [2]Medical Research Council Toxicology Unit, University of Cambridge, Cambridge, UK. [3]Edinburgh Pathology, Institute of Genetics and Cancer, University of Edinburgh, Edinburgh, UK. [4]Laboratory Medicine, NHS Lothian, Edinburgh, UK. [5]CRUK Scotland Centre, Institute of Genetics and Cancer, University of Edinburgh, Edinburgh, UK. [6]Institute for Research in Biomedicine (IRB Barcelona), The Barcelona Institute of Science and Technology, Barcelona, Spain. [7]Brain Mosaicism and Tumorigenesis (B400), German Cancer Research Center (DKFZ), Heidelberg, Germany. [8]European Molecular Biology Laboratory, European Bioinformatics Institute, Hinxton, UK. [9]Cancer Research UK Cambridge Institute, University of Cambridge, Cambridge, UK. [10]Division of Regulatory Genomics and Cancer Evolution (B270), German Cancer Research Center (DKFZ), Heidelberg, Germany. [11]San Diego Biomedical Research Institute, San Diego, CA, USA. [12]Universitat Pompeu Fabra (UPF), Barcelona, Spain. [13]Institució Catalana de Recerca i Estudis Avançats (ICREA), Barcelona, Spain. [14]Centro de Investigación Biomédica en Red en Cáncer (CIBERONC), Instituto de Salud Carlos III, Madrid, Spain. [15]Department of Pathology, University of Cambridge, Cambridge, UK. [16]Department of Histopathology, Cambridge University Hospitals NHS Foundation Trust, Cambridge, UK. *A list of authors and their affiliations appears at the end of the paper. ✉e-mail: d.odom@dkfz-heidelberg.de; sa696@cam.ac.uk; martin.taylor@ed.ac.uk

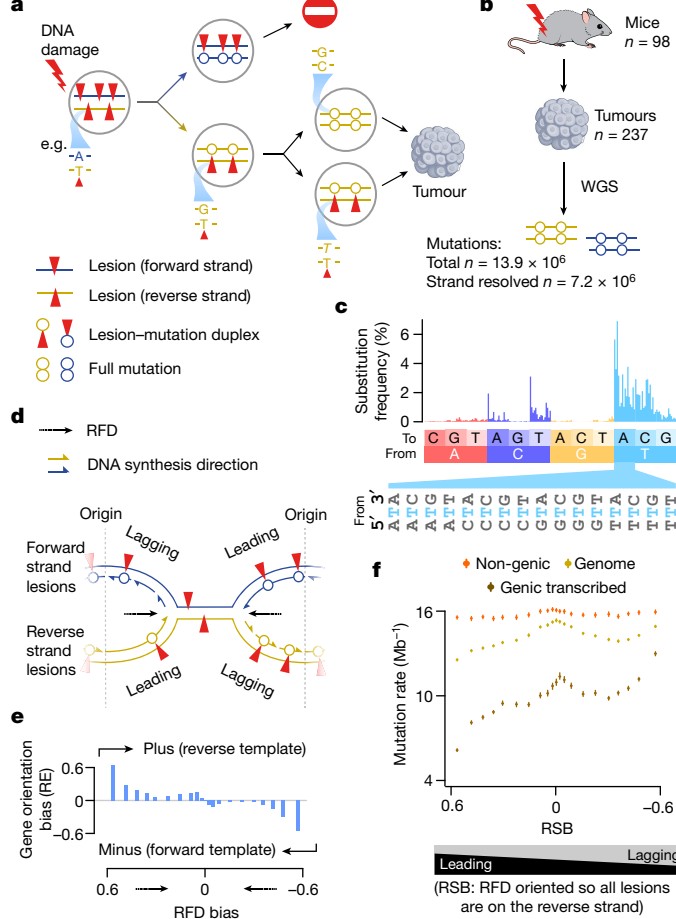

**Fig. 1 | Apparent replication-associated mutational asymmetry can be explained by transcription coupled repair. a**, Schematic of DNA lesion segregation[2]. Mutagen exposure induces lesions (red triangles) on both DNA strands (forward in blue; reverse in gold). Lesions that persist until replication serve as a reduced fidelity template. The two sister chromatids segregate into distinct daughter cells, so new mutations are not shared between daughter cells of the first division. Lesions that persist for multiple cell generations can generate multiallelic variation through repeated replication over the lesion (in italic). **b**, Summary of tumour generation and mutations called from whole-genome sequencing (WGS; Methods). **c**, Lesion strand resolved mutation spectra of all tumours ($n = 237$), representing the relative frequency of strand-specific single-base substitutions and their sequence context (192 categories). **d**, During the first DNA replication after DNA damage, template lesions (red triangles) are encountered by both the extending leading and the lagging strands. **e**, The relative enrichment (RE) of liver-expressed genes in the plus versus minus orientation (RE = (plus − minus)/(plus + minus)) across 21 quantile bins of replication fork directionality (RFD) bias (x axis). **f**, Mutation rates (y axis) for the whole genome (gold) stratified into 21 quantile bins of replication strand bias (RSB; x axis) show a higher mutation rate for the lagging strand than the leading strand replication on a lesion-containing template. This effect is enhanced in expressed genes (tan) and negligible in non-genic regions (orange). Whiskers show 95% bootstrap confidence intervals.

determine the lesion-containing strand for approximately 50% of the autosomal genome and the entire X chromosome for each tumour[2] (Extended Data Fig. 1). We analysed data from 237 clonally distinct tumours from 98 mice and could resolve the lesion strand for over 7 million base substitution mutations (Fig. 1b). Most (more than 75%) of the mutations are from T nucleotides on the lesion strand (Fig. 1c), consistent with previous analyses of DEN-induced tumours[2,22], and biochemical evidence of frequent mutagenic alkylation adducts on thymine[23].

The range of mutagenic alkylation adducts generated by activated DEN overlaps those from tobacco smoke exposure, unavoidable endogenous mutagens and alkylating chemotherapeutics such as temozolomide[23–25]. More generally, the mechanism of lesion segregation, which the strand-resolved analysis relies on, appears to be a ubiquitous property of base-damaging mutagens[2]. Here we newly exploit these strand-resolved lesions as a powerful tool to quantify how mitotic replication, transcription and DNA–protein binding mechanistically shape DNA damage, genome repair and mutagenesis.

## The mutational symmetry of replication

These well-powered and experimentally controlled in vivo data provide a unique opportunity to evaluate whether DNA damage on the template for leading strand replication results in the same rate and spectrum of mutations as on the lagging strand template. There are several reasons why they might differ. First, leading and lagging strand replication use distinct replicative enzymes[3,4,6,7], which may differ in how they handle unrepaired damage on the DNA template strand. Second, it is unknown whether the leading and lagging strand polymerases recruit different translesion polymerases, which could generate distinct error profiles. Third, substantially longer replication gaps are expected on the leading strand, if there is polymerase stalling[26]. Consequently, leading and lagging strands are thought to differ in their lesion bypass[5] and post-replicative gap filling[27,28].

On the basis of hepatocyte-derived measures of replication fork directionality (using Repli-seq and OK-seq, see Methods; Extended Data Fig. 2) and patterns of mutation asymmetry, we inferred whether the lesion-containing strand preferentially templated the leading or lagging replication strand (Fig. 1d). This was separately resolved for each genomic locus on a per tumour basis. Our initial analysis demonstrated a significantly higher mutation rate for lagging strand synthesis over a lesion-containing template (Pearson's correlation coefficient cor = −0.86, $P = 3.2 \times 10^{-9}$; Fig. 1e). However, gene orientation — and thus the directionality of transcription — also correlates with replication direction[29,30] and DEN lesions are subject to transcription-coupled repair (TCR)[2]. We therefore measured transcriptome-wide gene expression in the mouse liver on postnatal day 15 (P15), corresponding to the timing of DEN mutagenesis. This confirmed that the direction of transcription is strongly biased to match replication fork movement, and the effect is disproportionally evident in regions of extreme replication bias (Fig. 1e).

To disentangle the effects of transcription from replication, we measured mutation rates, jointly stratifying the genome by transcription state, replication strand bias, replication timing and genic annotation (Fig. 1f and Extended Data Fig. 3). Although transcribed regions exhibit a strong correlation of mutation rate with replication strand bias (Pearson's cor = −0.86, $P = 3.1 \times 10^{-7}$), genome-wide multivariate regression shows that the strongest independent effect on the DEN-induced mutation rate is transcription over the lesion-containing strand ($P < 1 \times 10^{-300}$), followed by replication time ($P = 6 \times 10^{-162}$). As mismatch repair is biased towards earlier replicating genomic regions[31], it may be partially responsible for correcting some mismatch–lesion heteroduplexes. We considered genic and non-genic regions of the genome across 21 quantiles of replication timing and found that, although there is a correlation between mutation rate and replication time supportive of mismatch repair, its role is minor relative to TCR (Extended Data Fig. 4). Replication strand bias has the smallest effect on mutation rate of tested measures (Extended Data Fig. 3j). Outside of genic regions, the correlation of replication strand bias with mutation rate is negligible (Fig. 1f and Extended Data Fig. 3j). This unexpected consistency in the rate of mutations generated by replication over alkyl lesions points to a shared mechanism of lesion bypass for the leading and lagging strands, possibly involving recruitment of the same translesion polymerases.

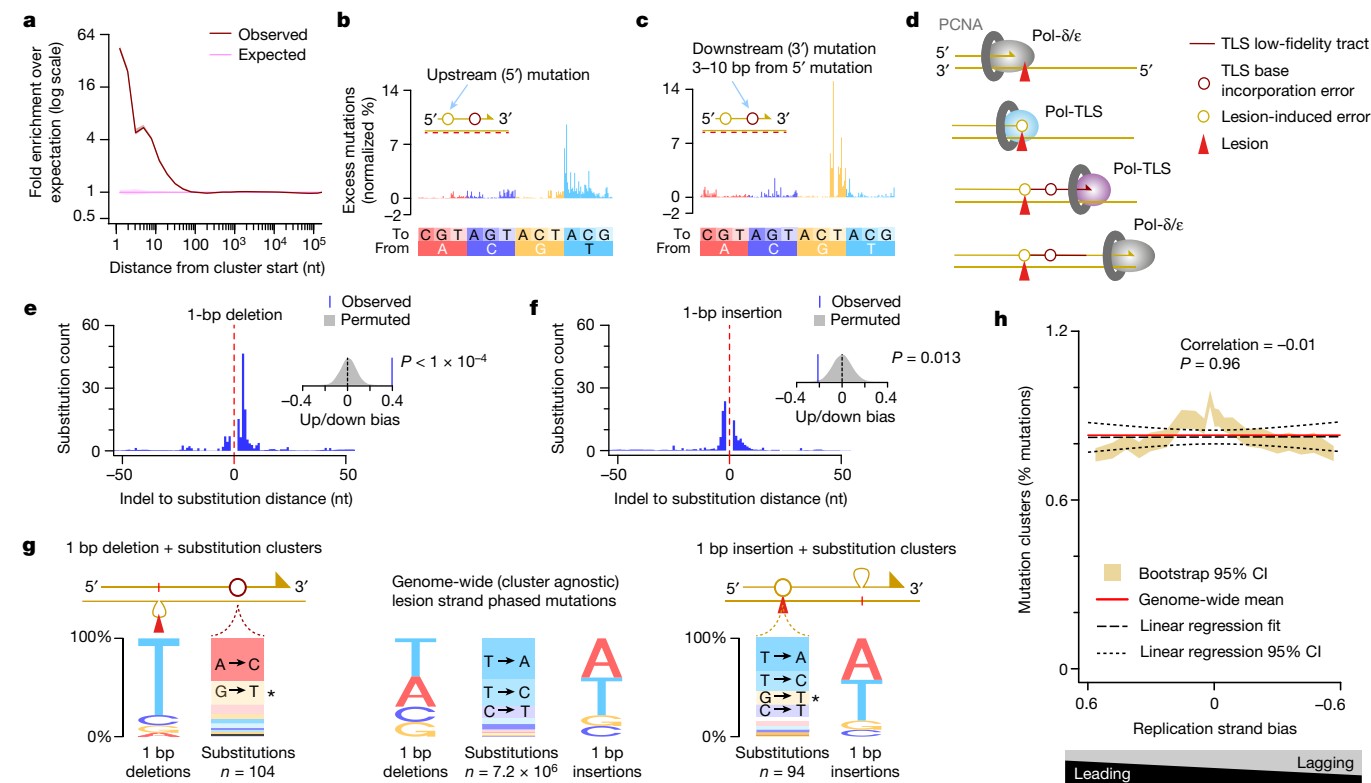

**Fig. 2 | Translesion synthesis drives collateral mutagenesis on both the leading and the lagging strands. a**, Closely spaced mutations (brown) occur more frequently than expected based on permutation of mutations between tumours (pink; bootstrap 95% CI is shaded, too small to visualize). **b**, Residual mutation signature (after subtracting expected mutations) for cluster upstream mutations. Cluster orientation by the lesion-containing strand (red dashed line; Methods). **c**, Residual signature of downstream cluster mutations, plotted as per **b**. **d**, Schematic illustrating mutagenic translesion synthesis (TLS) (yellow circle) and collateral mutagenesis (brown circle). **e**, Substitutions are highly clustered downstream of 1 bp deletions. The inset shows the density plot for 10,000 random permutations of lesion strand assignment (grey) compared with the observed level of upstream/downstream bias. Only clusters where the substitution could be definitively assigned to an upstream or downstream location were considered. Two-sided $P$ values were empirically derived from the permutations. nt, nucleotide. **f**, Single-base insertions are

also clustered with substitutions, but biased to upstream of the insertion; plotted as per **e**. **g**, One-base pair deletions with a downstream substitution within 10 bp (left panel) show significant bias towards deletion of T (rather than A) from the lesion-containing strand compared with the rate genome wide (centre panel, two-sided Fisher's exact test odds = 16.5, $P = 1.04 \times 10^{-16}$). Downstream substitutions are also highly distorted from the genome-wide profile (two-sided Chi-squared test $P = 8.5 \times 10^{-46}$). By contrast, insertion mutations and their proximal substitutions resemble the genome-wide profiles, with the notable additional contribution from the G→T substitutions (*) that also associate with both substitution and 1 bp deletion clusters. **h**, The rate of mutation clusters is not correlated with replication strand bias; consistently, approximately 0.8% of substitution mutations are found in clusters spanning 10 nt or fewer, indicating a similar rate of TLS for both the leading and the lagging strands.

## Strand-resolved collateral mutagenesis

It has been proposed that when translesion polymerases replicate across damaged bases, they can generate proximal tracts of low-fidelity synthesis[32–34]. In bacteria and yeast, this mechanism produces clusters of mutations[35,36] and such collateral mutagenesis has recently been reported in vertebrates[37]. Consistent with these models, we found that mutations within 10 nt of each other are significantly elevated over permuted expectation (two-sided Fisher's test, odds ratio 11.9, $P < 2.2 \times 10^{-16}$). This enrichment is most pronounced at 1–2 nt spacing, decreases after one DNA helical turn (approximately 10 nt) and decays to background within 20 nt (Fig. 2a and Extended Data Fig. 5). These short clusters are overwhelmingly isolated pairs of mutations (98% pairs, 2% trios) phased on the same chromosome (Extended Data Fig. 5e).

We oriented the clusters by their lesion-containing strand, and designated the first mutation site to be replicated over on the lesion-containing template as the upstream (5′) mutation and subsequent mutations were designated downstream (3′). Upstream mutations showed a mutation spectrum closely resembling the tumours

as a whole (Fig. 2b and Extended Data Fig. 5a,b,i), indicating that it represents a typical lesion-templated substitution.

By contrast, downstream mutations have distinct mutation spectra (Extended Data Fig. 5c). Those located more than two nucleotides downstream show a strong preference for G→T substitutions (Fig. 2c and Extended Data Fig. 5h,l–n). As mutations are called relative to the lesion-containing template strand, this indicates the preferential misincorporation of A nucleotides opposite a template G nucleotide, thus newly revealing the intrinsic error profile of an extending translesion polymerase. Mutation pairs with closer spacing (2 nt or fewer) exhibit somewhat divergent mutation signatures (Extended Data Fig. 5h,j,k), probably reflecting both sequence-composition constraints and processes such as the transition between alternate translesion polymerases (Fig. 2d).

Extending these observations of collateral translesion mutagenesis, we found significant clustering of insertion and deletion mutations with base substitutions (insertion/deletion mutation within 100 bp of a substitution, two-sided Fisher's test odds ratio 103, $P < 2.2 \times 10^{-16}$ compared with permuted expectation; Fig. 2e,f and Extended Data Fig. 6a–i). Single-base deletions preferentially remove T nucleotides

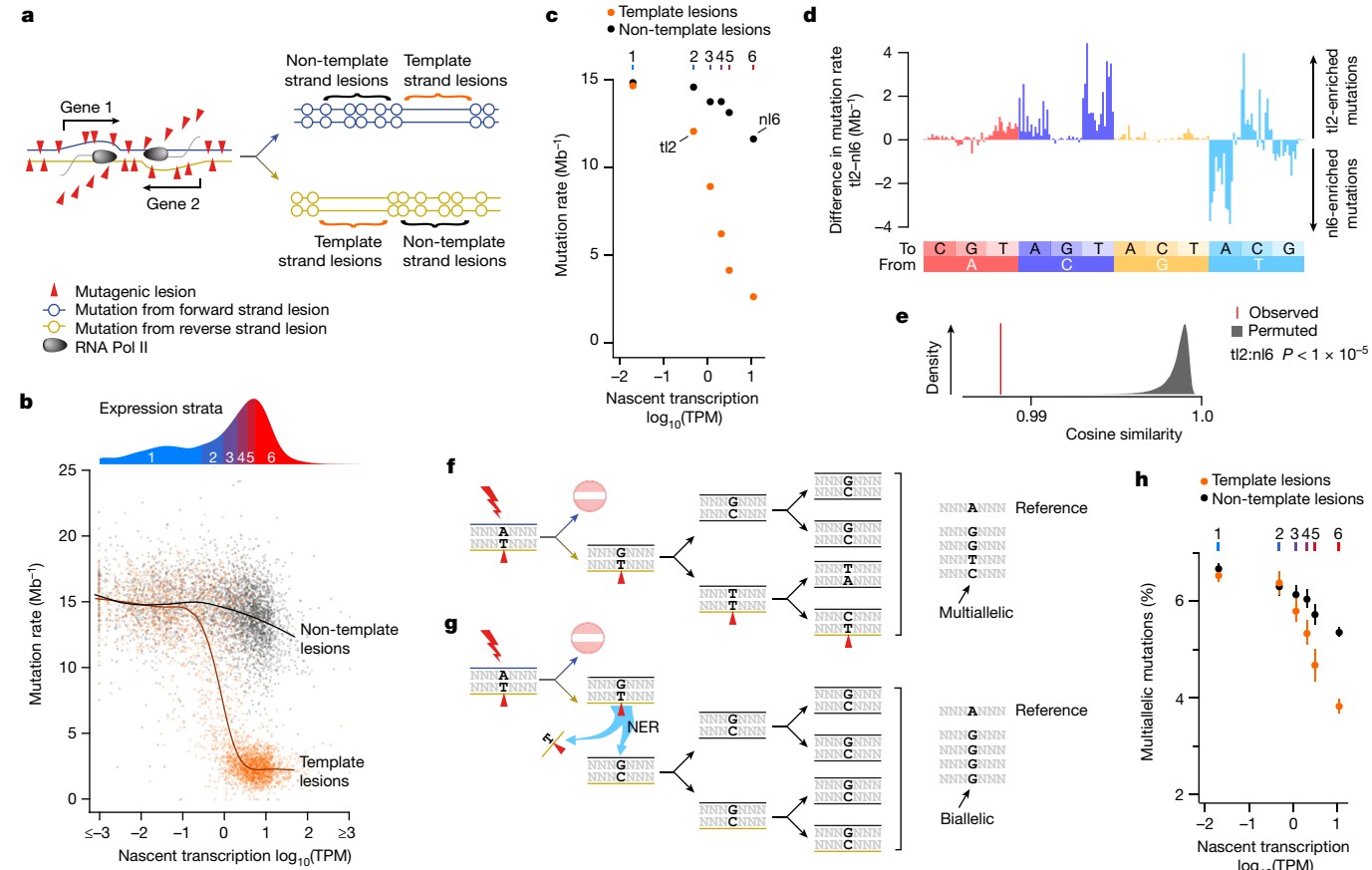

**Fig. 3 | Multiallelic variation demonstrates transcription-associated repair of the non-template DNA strand. a**, DNA lesions (red triangles) on the transcription template strand can cause RNA polymerase to stall and trigger transcription-coupled NER. Cells that inherit the template strand of active genes have a depletion of mutations through the gene body. **b**, Mutation rate (*y* axis) for individual genes relative to their nascent transcription rate (*x* axis) estimated from intronic reads. Mutation rates for each gene (*n* = 3,392) are calculated separately for template (orange) and non-template (black) strand lesions. The curves show best-fit splines. Genes are grouped into six expression strata (used in subsequent analyses), indicated by the density distribution (top). TPM, transcripts per million. **c**, Mutation rates for genes grouped into expression strata (1–6; top axis), calculated separately for template strand lesions (orange) and non-template strand lesions (black). Whiskers indicate 95% bootstrap confidence intervals (too small to resolve). Labels indicate data

used in subsequent mutation spectra panels (**d**,**e**). **d**, Despite similar mutation rates, the mutation spectrum differs between non-template lesion stratum 6 (nl6) and template lesion stratum 2 (tl2). **e**, Permutation testing confirms that the mutation spectra differs between the transcription template and the non-template strand, even when overall mutation rates are similar. Comparison of tl2 and nl6 mutation spectra (red) and after gene-level permutation of categories. $n = 10^5$ permutations (grey). **f**, Lesions (red triangles) that persist for multiple cell generations can generate multiallelic variation through repeated replication over the lesion. **g**, Lesions rapidly removed by NER persist for fewer cell cycles, generating less multiallelic variation. **h**, The multiallelic rate (*y* axis) for template strand lesions (orange) is reduced with increasing transcription (*x* axis). The same is apparent for non-template lesions (black), indicating that enhanced repair of non-template lesions is also associated with greater transcription. Whiskers show bootstrap 95% confidence intervals.

from the lesion strand both genome wide and in mutation clusters (Fig. 2g; two-sided Fisher's test odds ratio 16.5, $P = 1.04 \times 10^{-16}$), which indicates a base-skipping mode of lesion bypass. These single-base deletions are associated with downstream substitutions within 10 nt that include the G→T substitutions already identified as a signature of collateral translesion mutagenesis, but more prominently a distinct substitution signature of A→C on the lesion strand (Fig. 2g). In contrast to deletions, nucleotide insertions are clustered downstream of typical DEN adduct-induced base substitutions, pointing to collateral insertion mutagenesis by translesion polymerases (Fig. 2g and Extended Data Fig. 6h,i).

Three lines of evidence support a model in which the same translesion polymerases are recruited with equal efficiency and processivity to both the leading and the lagging strands. First, the leading and lagging strands have essentially identical relative rates of mutation clusters (Fig. 2h). Second, the mutation spectra of the downstream mutations are the same (Extended Data Fig. 5o). Third, the length distribution of clusters matches between leading strand-biased and lagging

strand-biased regions (no significant difference in size distribution, Kolmogorov–Smirnov test ($P = 0.15$) despite more than 98% power to detect a difference in the distribution of cluster lengths of 4% or more; Extended Data Fig. 5p,q).

Having established the replicative symmetry of damage-induced mutagenesis and determined the relative contributions of replication and transcription on mutation rate, we next looked in detail at the pronounced strand-specific effects of transcription on DNA repair and mutagenesis.

## Multiallelism reveals repair kinetics

Using liver RNA sequencing data (P15 mice), we found that nascent transcription estimates provide a better correlation with mutation rate than steady-state transcript levels (Extended Data Fig. 7a–d), as expected[8]. Increased transcription decreases the mutation rate for template strand lesions up to an expression level of ten nascent transcripts per million (Fig. 3a,b). Beyond this, the mutation rate plateaus

and is not further reduced by additional transcription, suggesting that the remaining mutagenic lesions are largely invisible to TCR (Extended Data Fig. 7c,d).

Unexpectedly, the non-template strands of genic regions also showed a modest reduction in mutation rate with increased transcription (Fig. 3c), but the resulting mutation signature differs from that on the template strand. This discordance suggests that cryptic antisense transcription is not responsible (Fig. 3d,e and Extended Data Fig. 7e–j) and that there is either (1) enhanced (non-TCR) surveillance of lesions on the non-template strand or (2) generally reduced alkylation damage to transcriptionally active regions.

We used another insight from lesion segregation to disentangle patterns of differential damage from differential repair. As DNA lesions from DEN treatment, as with all other tested mutagens[2], can persist for multiple cell cycles, each round of replication could incorporate a different incorrectly paired nucleotide opposite a persistent lesion. This results in multiallelic variation: multiple alleles at the same genomic position within a tumour[2] (Figs. 1a and 3f). Lesions in efficiently repaired regions will persist for fewer generations and therefore have fewer opportunities to generate multiallelic variation, so are expected to exhibit lower multiallelic rate (the fraction of mutations with multiallelic variation) than less efficiently repaired regions (Fig. 3g). By contrast, differential rates of damage, although influencing overall mutation rate, do not systematically distort the persistence of an individual lesion, so would have no influence on rates of multiallelic variation.

Whether mutation suppression on the non-template strand is caused by enhanced repair or reduced damage can now be established through the comparison of multiallelic variation rates. For lesions on the template strand, multiallelic rate decreases with increased transcription (Fig. 3h), reflecting the progressive removal of lesions across multiple cell cycles by TCR, as expected. The multiallelic rate for non-template strand lesions is also reduced with greater transcription (Fig. 3h), revealing enhanced repair rather than decreased damage. Combined with the distinct repair signature of the two strands (Fig. 3d,e and Extended Data Fig. 7j), this demonstrates that in expressed genes, there is transcription-associated repair activity of the non-template strand, in addition to the template strand-specific TCR. We speculate that this may reflect enhanced global nucleotide excision repair (NER) surveillance in the more open chromatin of transcriptionally active genes.

## Steric influences on damage and repair

Transcription-associated repair of non-template lesions (Fig. 3h) highlights the importance of DNA accessibility for repair of DNA damage. Although it is well established that mutation rate is correlated with nucleosome positioning and transcription factor binding[7,9–11,38], our lesion strand resolved measures of mutation and multiallelic rate provide an opportunity to deconvolve the contributions of differential damage from repair in these genomic contexts.

We quantified the DNA accessibility landscape of the genome using ATAC-seq (in the P15 mouse liver; Methods), and annotated it using experimentally defined transcription factor binding (including chromatin immunoprecipitation followed by sequencing (ChIP–seq) mapping of CTCF binding in the P15 mouse liver; Methods) and pre-existing maps of nucleosome positioning[39]. In all contexts, we found that greater DNA accessibility corresponds to both reduced mutation rate and reduced rate of multiallelic variation, implicating the efficient repair of accessible DNA as a major determinant of damage-induced mutation rate (Fig. 4a,b). Indeed, the 10 bp periodicity of mutations in nucleosome-wrapped DNA, as previously seen for other mutagens[11,40], is recapitulated by the multiallelic rate variation that we identified (Extended Data Fig. 8a–c).

Sequence-specific binding proteins, such as transcription factors and CTCF, interact with DNA more transiently than nucleosomes[41]. We found

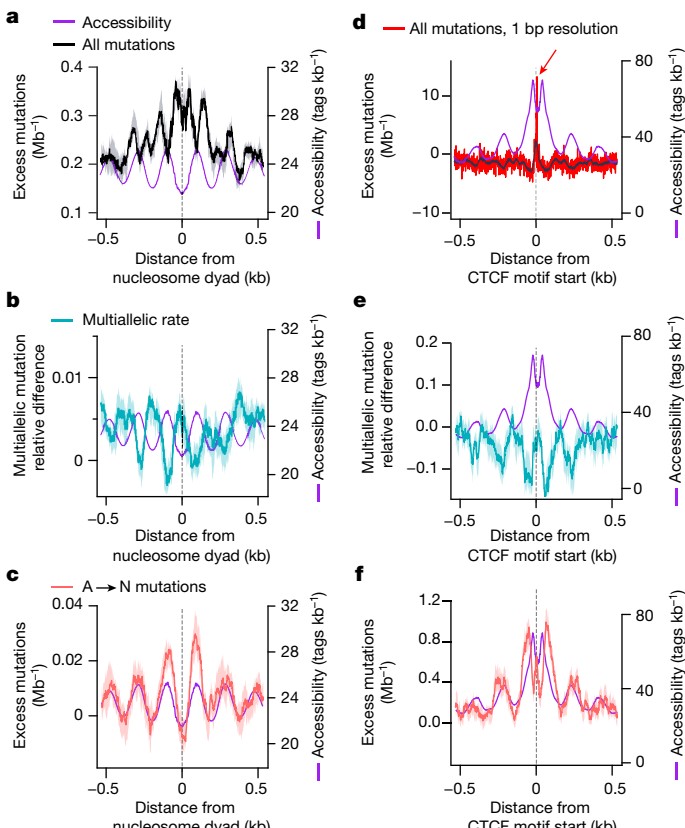

**Fig. 4 | Rapid repair of accessible DNA shapes the mutational landscape, but CTCF binding causes extreme local distortions. a**, Nucleosome occupancy shapes the mutational landscape[56,57], with higher mutation rates (21 bp sliding window) over the nucleosomes (for example, $x = 0$), and lower rates in more-accessible linker regions (accessibility measured by ATAC-seq from P15 mouse liver, in purple with scale on the right axis and larger values corresponding to greater accessibility). Mutation and multiallelic rates are shown with shaded 95% bootstrap confidence intervals (also in subsequent panels). **b**, High rates of multiallelic variation are found at sites of low accessibility and high mutation rate, indicating that high rates of mutation represent slow repair. **c**, The rate of A→N mutations is the inverse of the overall mutation profile, with high rates of A→N corresponding to accessible regions and rapid repair. **d**, Mutation rates are dramatically elevated at CTCF-binding sites (21 bp sliding window, in black; single-base resolution, in red). **e**, High accessibility at CTCF sites again corresponds to low multiallelic variation and low mutation rates (**d**), with the exception of the mutation hotspot (red arrow), which does not show a corresponding increase in multiallelism, indicating that higher rates of damage cause these hotspots. **f**, Mutations of A→N closely track DNA accessibility.

reduced mutation rates and multiallelic variation adjacent to and across their binding sites compared with genome-wide averages (Extended Data Fig. 8h–j), suggesting that transient binding is not a strong impediment to repair processes. High information content nucleotides in sequence-specific binding motifs show exceptionally reduced mutation rates that are not accompanied by corresponding decreases in multiallelic variation (Extended Data Fig. 8i,j). This discordance is consistent with reduced damage (rather than enhanced repair) in these sites. Given the close contacts made between the bases and proteins in these motifs, it raises the possibility that binding proteins offer some protection from lesion formation. Uniquely, the CTCF-binding footprint contains specific sites that exhibit pronounced, lesion strand-specific elevations of mutation rate that are not accompanied by increased multiallelic variation (Fig. 4d,e and Extended Data Fig. 8e–g). This suggests that in this case, the elevated mutation is due to elevated DNA damage, rather than primarily a consequence of suppressed repair.

We identified an anomalous enrichment of apparent A→N mutations in genomic loci that showed highly efficient repair for other nucleotides (Fig. 4c,f). These accessible loci include those adjacent to CTCF and transcription factor-binding sites and linker DNA between nucleosomes (Fig. 4 and Extended Data Figs. 8d and 9). This enrichment of A→N mutations extends into sequence-specific binding sites (Extended Data Figs. 8c and 9e,f). A possible explanation for the enrichment of A→N mutations is that, in some circumstances, the activity of NER is itself mutagenic.

## Nucleotide excision repair is mutagenic

We propose a mechanistic model for mutagenic NER, arising when two lesions occur in close proximity, but on opposite strands of the DNA duplex. Repair of one lesion, which entails excision of an approximately 26 nt single-stranded segment containing the lesion[42,43], would leave a single-stranded gap containing the second lesion on the opposite strand; resynthesis using this as a template would necessitate replication over that remaining lesion (Fig. 5a). As a result, nucleotide misincorporation opposite a T lesion in the single-stranded gap would be erroneously interpreted as a mutation from an A lesion (Fig. 5a) when phasing lesion segregation. We subsequently refer to this mechanism as translesion resynthesis-induced mutagenesis (TRIM), or NER-TRIM specifically in the context of NER.

As NER-TRIM requires lesions on both DNA strands, mutagenic NER can only occur when both lesion-containing strands are duplexed, for example, in the first cell generation following DEN mutagenesis; NER-TRIM would not occur in daughter cells with only one lesion-containing strand per duplex. It follows that regions with the highest − and thus fastest − repair rates are most likely to experience NER-TRIM. This prediction is consistent with our observation of local enrichment of apparent A-lesion mutations in accessible regions with otherwise low rates of mutations and low multiallelic variation (Fig. 4c,f).

Local gradients in repair efficiency are also expected to lead to enrichment of NER-TRIM. The most efficient repair that we observed is transcription-coupled NER, in which there is a steep gradient of repair efficiency between the template and non-template strands. There is a pronounced increase in the rate of apparent A→N mutations on the template strand of expressed genes, whose sigmoidal profile closely mirrors the decrease in T→N mutations on the same strand (Fig. 5b). The saturation of repair at higher expression levels is reflected in a corresponding saturation of NER-TRIM, demonstrating that the rate of template strand A→N mutations is not simply dependent on transcription, but on TCR.

Similar local gradients of repair can also explain the elevated rate of A→N mutations in CTCF and transcription factor-binding sites (Extended Data Fig. 9e,f), where nucleotides adjacent to the binding site are more accessible than those within the binding site. High-efficiency repair of the accessible DNA would result in an excision gap that extends into the binding site, where a more protected lesion then serves as a template for repair resynthesis.

## The TRIM origin of twin sister tumours

A subset of tumours in our dataset provided an opportunity to directly test further predictions of this NER-TRIM model and demonstrated a remarkable propensity for NER-TRIM mutagenesis to drive oncogenic transformation. Of the complete set of DEN-induced tumours[2], 2% (8 of 371) exhibited the same mutation spectra as other tumours but completely lacked the mutational asymmetry of lesion segregation (Extended Data Fig. 10a). This pattern is expected to result from the persistence of mutations derived from lesions on both strands (Fig. 5c and Extended Data Fig. 10b). On the basis of extensive genomic and histological evidence (Extended Data Fig. 10c–h), we conclude

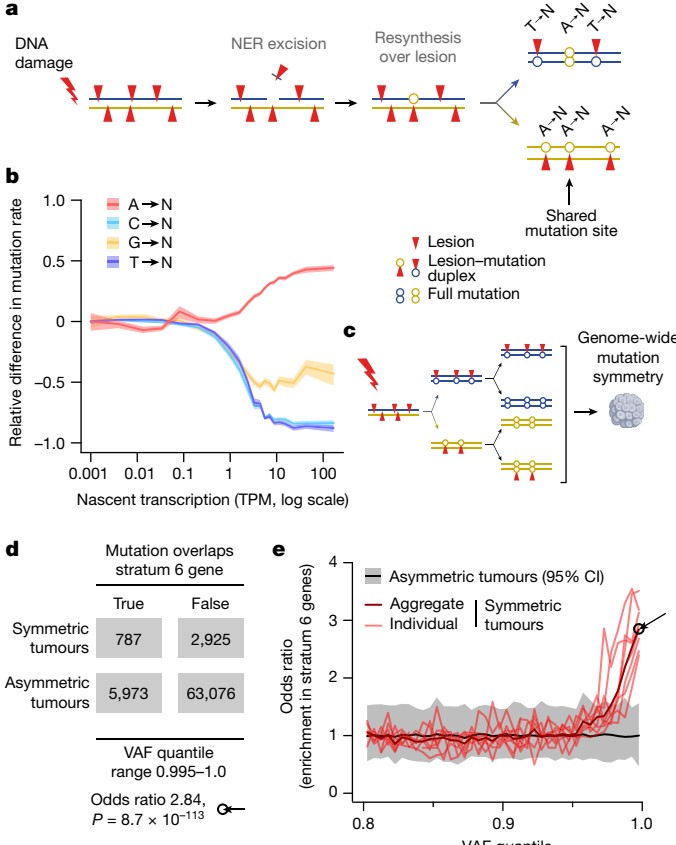

**Fig. 5 | Nucleotide excision repair is mutagenic when lesions on opposing strands are in close proximity. a**, Mechanism of NER translesion resynthesis-induced mutagenesis (NER-TRIM). Lesion-containing single-stranded DNA is excised and consequently a residual lesion in close proximity on the opposite strand would be used as a low-fidelity template for repair synthesis. This creates isolated mutations with opposite strand asymmetry to the genomic locality (for example, A→N within a T→N segment). Most lesion-induced mutations are not shared between daughter lineages, whereas those from NER-TRIM can be shared (black arrow). **b**, The rate of A→N mutations on the genic template strand increases with gene expression, mirroring the decrease in mutations from other bases due to TCR. The relative difference (*y* axis) in mutation rate for each nucleotide is (obs − exp)/(obs + exp); exp is the mutation rate for that nucleotide in non-expressed genes, and obs is the rate observed in the body of genes with the indicated expression level (*x* axis). Rates shown for lesions on the transcription template strand, with 95% confidence interval (shaded areas) from 100 bootstrap samples of genes. **c**, Schematic illustrating the generation of a mutationally symmetric tumour through the survival of both post-mutagenesis daughter genomes. NER-TRIM mutations in symmetric tumours will be characterized by abnormally high VAF as they will be shared by both contributing genomes (Extended Data Fig. 10b). **d**, Contingency table illustrating the enrichment of mutations with high VAF (0.995–1.0 quantile) in highly expressed genes of mutationally symmetric tumours (*n* = 8) compared with asymmetric tumours (*n* = 237). Statistical significance by two-tailed Fisher's exact test. **e**, Symmetric tumours are highly enriched for high VAF mutations in highly expressed genes. Odds ratios (*y* axis) are as in **d**, for VAF quantile bins of 0.005 (*x* axis). The black arrow shows the odds ratio calculated in **d**.

that these eight mutationally symmetrical tumours are each made up of two diploid sister clones derived from both daughters of a mutagenized cell.

Lesion segregation predicts that mutations will be independent and not shared between sister clones (Fig. 1a). However, mutations arising from NER-TRIM are expected to be shared between sister clones (Fig. 5a). The variant allele frequency (VAF) of a somatic mutation is proportional to the fraction of cells in the tumour that contain the

mutation. Consequently, we expect the VAF of shared mutations derived from NER-TRIM to be approximately twice that of mutations found in only one of the two daughter cell lineages. Owing to the absence of mutational asymmetry in these eight tumours, it is not possible to define which individual mutations arose from NER-TRIM. However, as we have shown that NER-TRIM is enriched in highly expressed genes, we tested whether high VAF mutations were biased to those regions in the symmetrical tumours ($n$ = 8) compared with the asymmetric tumours ($n$ = 237). Our results demonstrated a pronounced and significant enrichment, as we predicted, both in aggregate (odds ratio 2.84, two-tailed Fisher's test $P = 8.7 \times 10^{-113}$; Fig. 5d) and individually for each tumour (Fig. 5e), confirming expectations of the NER-TRIM model.

Finally, we note that in the symmetrical sister-clone tumours, the oncogenic driver mutations in the MAPK pathway that typify these DEN-induced tumours[2,22] are all significantly biased to the highest VAF mutations, in contrast to the driver mutations in the asymmetric tumours ($P = 3.61 \times 10^{-5}$ two-tailed Wilcoxon rank-sum test, Bonferroni corrected; Extended Data Fig. 10i–y). This suggests that driver mutations in the symmetrical tumours arose through NER-TRIM and may explain the co-evolution of both sister clones in a single tumour.

## Discussion

In damaged DNA, most mutations arise from replication bypass of unrepaired lesions, which can result in chromosome-scale mutational asymmetry[2]. We leveraged this discovery to explore the mechanisms of mutagenesis and repair in vivo at high resolution, with single-base, single-strand specificity. The persistence of DNA lesions for multiple cell generations leads to the generation of multiallelic variation, its quantification providing insight into repair kinetics that allowed us to discriminate the relative contributions of initial damage from subsequent repair in shaping mutation rate patterns.

It has long been expected that the asymmetry of leading and lagging strand replication would lead to asymmetric replication fidelity on damaged DNA[27,28,44,45], and analysis of UV-induced mutation patterns supports that expectation[5,12]. However, our system, with over $7.2 \times 10^6$ lesion strand-resolved mutations and cell-type-matched measures of replication strand bias, means we are uniquely powered to question the generality of this model. Contrary to expectation, we found a remarkable symmetry of mutation rate for leading and lagging strand replication. Matched patterns of collateral mutagenesis – proximal downstream mutations thought to arise from continued synthesis by translesion (TLS) polymerases[37] – point to the recruitment of identical TLS polymerases for the bypass of small alkylation adducts on both replication strands.

Our deeper exploration of mutation clusters demonstrates spatial shifts in mutation signature 3 bp downstream of nucleotides misincorporated opposite damaged bases, supporting a model for the hand-off between TLS polymerases[46,47]. We also provide evidence of competition between TLS polymerases. Single-base deletions, such as base substitutions, are strongly strand asymmetric. This implicates the skipping of damaged template bases (−1 frameshifting), which in vitro studies show is common for some of the TLS polymerases such as polymerase-κ[48]. These skipping versus low-fidelity incorporation mechanisms of lesion bypass are associated with highly distinct signatures of downstream collateral mutations, arguing that the alternate outcomes reflect the recruitment of distinct combinations of TLS polymerases. The contrast in mutation asymmetry that we found between replication over UV and DEN damage suggests at least two available strategies of mutagenic translesion bypass in mammalian cells. For example, re-priming followed by gap-filling[49], leading to replication strand asymmetric mutagenesis, versus on-the-fly bypass[28], which results in replication strand symmetric mutagenesis.

The balance between these probably vary between different types of damage.

Although we found that replication strand biases do not influence the rate of mutations from alkylation damage, both transcription and DNA accessibility have large effects. To better understand how these other features of the genome influence mutation rates, we analysed multiallelic variation as a powerful means to infer the relative kinetics of repair, and disentangle differential damage from differential repair across the genome. This reveals the transcription-associated repair of genic non-template strands, in addition to the well-established TCR of the template strand[8]. Beyond the effects of transcription, the mutational landscape of damaged genomes closely tracks DNA accessibility. This pattern is mirrored by the rate of multiallelic variation, thus providing in vivo evidence that more efficient repair of accessible DNA, rather than differential DNA damage, is primarily responsible for shaping the distribution of damage-induced mutations.

There are, however, some exceptions to the dominance of repair. We found that within transcription factor-binding sites, close contact between high-information-binding site nucleotides and sequence-specific binding proteins shows evidence of providing protection from base damage. By contrast, a subset of nucleotides specifically within CTCF-binding sites exhibit dramatically elevated mutation rates, and lesion strand phasing confirmed that it was damage induced. The identity of these sites with elevated mutation can only partially be reconciled with the structure of the CTCF–DNA interface. We speculate that this structure may be modified, for example, by interacting with cohesin, leading to bending[50,51] and partial melting of the DNA duplex, resulting in greater exposure of the nucleotide bases to chemical attack.

Finally, we found that genomic regions that are most efficiently repaired are also, counterintuitively, specifically prone to repair-induced mutagenesis. Building on evidence that transcription-coupled NER can be mutagenic in bacteria[52] and quiescent yeast[53], we present multiple orthogonal analyses supporting the conclusion that TRIM occurs in vivo in mammals, although confirming the involvement of NER requires further experimental validation. We also showed that NER-TRIM is not purely dependent on transcription, but more generally results from the repair of lesions in close proximity, on opposite strands. It is therefore expected to occur when damage loads are high or closely spaced, for example, UV damage in promoters and ETS factor-binding sites[54,55]. Although NER-TRIM mutations represent only a small fraction of damage-induced mutations, they are specifically biased to functionally important sites: they are responsible for most driver mutations seen in symmetric tumours and, perhaps most importantly, NER-TRIM preferentially results in the misincorporation of a normal DNA base on the template strand of highly expressed genes. That incorrect normal base is not a substrate for subsequent NER and could therefore lead to efficient miscoding of a protein before genome replication, and in the case of an oncogenic mutation, potentially driving otherwise quiescent cells towards oncogenic transformation.

Our ability to resolve both mutation rate and multiallelism at single-strand, single-base resolution allows us to infer lesion longevity and thus disentangle differential DNA damage from differential repair. This powerful approach provides in vivo insights into how strand-asymmetric mechanisms underlie the formation, tolerance and repair of DNA damage, thereby shaping cancer genome evolution.

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

**Liver Cancer Evolution Consortium**

**Sarah J. Aitken**[2,9,15,16], **Stuart Aitken**[1], **Craig J. Anderson**[1], **Claudia Arnedo-Pac**[2,6], **John Connelly**[1,2,3,4], **Frances Connor**[9], **Maëlle Daunesse**[8], **Ruben M. Drews**[9], **Ailith Ewing**[1], **Christine Feig**[9], **Paul Flicek**[8], **Paul A. Ginno**[10], **Vera B. Kaiser**[1], **Elissavet Kentepozidou**[8], **Erika López-Arribillaga**[6], **Núria López-Bigas**[6,12,13,14], **Juliet Luft**[1], **Margus Lukk**[9], **Duncan T. Odom**[9,10], **Oriol Pich**[6], **Tim F. Rayner**[9], **Colin A. Semple**[1], **Inés Sentís**[6], **Vasavi Sundaram**[8], **Lana Talmane**[1], **Martin S. Taylor**[1] & **Jan C. Verburg**[1]

## Methods

### Genomic annotation

The C3H/HeJ mouse strain reference genome assembly C3H_HeJ_v1 (ref. 58) was used for read mapping, annotation and analysis. WGS regions with abnormal read coverage (ARC regions; 12.7% of the genome) were masked from analysis, as previously described[2]. Gene annotation was obtained from Ensembl v.91 (ref. 59).

### Mutation asymmetry

Mutation calling and quality filtering were performed using WGS of 371 DEN-induced liver tumours from $n = 104$ male C3H mice (Supplementary Table 1), as previously reported[2]. All mutation data were derived from sequence data in the European Nucleotide Archive (ENA) under accession PRJEB37808, and processed files directly used as input for this work are publicly available[2].

Genomic segmentation on mutational asymmetry was performed as previously reported[2]. Mutational strand asymmetry was scored for each genomic segment using the relative difference metric $S = (F - R)/(F + R)$ where $F$ is the rate of mutations from T on the forward (plus) strand of the reference genome and $R$ is the rate of mutations from T on the minus strand (mutations from A on the plus strand). A mutational asymmetry score of $S > 0.33$ was used to identify the inheritance of forward strand lesions and $S < -0.33$ as the inheritance of reverse strand lesions. A rare subset of tumours (2.7%) exhibited uniform mutational symmetry (more than 99% of autosomal mutations in genomic segments with abs($S$) < 0.2; these were labelled 'symmetric' tumours.

Except where otherwise stated (within the final results section), analyses were confined to $n = 237$, clonally distinct DEN-induced tumours that met the combined criteria of: (1) not labelled as symmetric, (2) tumour cellularity of more than 50%, and (3) more than 80% of substitution mutations attributed to the DEN1 signature[2] by sigFit (v.2.0)[60].

Relative to the reference genome sequence, a plus ($P$) strand gene was transcribed using the reverse ($R$) strand as a template. So, a $P$ strand gene in a genomic segment with $R$ strand lesions (denoted $RP$ orientation) is expected to be subject to TCR. A minus ($M$) strand gene with forward ($F$) strand lesions ($FM$ orientation) is also expected to be subject to TCR, as the retained lesions are again on the transcription template strand. Conversely, $FP$ and $RM$ orientation combinations will have lesions on the non-template strand for transcription. For DNA replication, we similarly refer to whether the preferential template for the leading strand contains the retained lesions or whether the preferential template for the lagging strand contains the retained lesions.

### Mutation rates and spectra

Mutation rates were calculated as 192 category vectors representing every possible single-nucleotide substitution conditioned on the identity of both the upstream and the downstream nucleotides. Each rate being the observed count of a mutation category divided by the count of the trinucleotide context in the analysed sequence. To report a single aggregate mutation rate, the three rates for each trinucleotide context were summed to give a 64 category vector and the weighted mean of that vector reported as the mutation rate. The vector of weights being the fraction of each trinucleotide in a reference sequence, for example, the composition of the whole genome. Strand-specific mutation rates were calculated with respect to the lesion-containing strand, with both mutation calls and sequence composition reverse complemented for reverse strand lesions. Autosomal chromosomes were considered diploid and the X chromosome haploid (male mice) for the purposes of calculating mutation rates and sequence composition. For the counting of strand-specific mutations, a threshold VAF > 10% was applied to remove mutation calls from contaminating non-clonal cells.

Subtracted spectra plots (Fig. 2c,d) were calculated by subtracting the counts of simulated tumour datasets from those of observed datasets and then scaling as for mutation spectra, so that the absolute area of

the histogram summed to 100. Percent repair efficiency (Extended Data Fig. 7j) was calculated as (observed/expected) × 100, where expected was the corresponding mutation rate for non-expressed genes (stratum 1, see below) averaged between the template and non-template strand. Cosine similarity was used as a relative measure of mutation signature similarity. Mutation signature deconvolution was performed using sigFit (v.2.0), with two component signatures ($K = 2$) chosen based on heuristic goodness-of-fit for integer values of $K$ from 2 to 8, with 2,000 iterations each. Final $K = 2$ deconvolution used 40,000 iterations.

The expected number of mutations at each position of the analysed transcription factor-binding site (Supplementary Table 2) and nucleosome regions was calculated as a sum of genome-wide rates (mutations per base pair) for that particular trinucleotide context from each tumour that had this region classified as either forward or reverse segment. The genome-wide rate for each tumour was calculated by dividing the number of mutations in a particular trinucleotide context (that fall within genomic space phased to have inherited either a forward or a reverse lesion-containing strand) by the total count of that trinucleotide in that genomic space; this was done separately for forward and reverse segments.

Excess mutations per Mb were calculated as $(observed_{i,n} - expected_{i,n}) \times 10^6/(count_i)$, where $i$ is the relative position within the region, $count_i$ represents a total number of regions with non-'$N$' nucleotide at position $i$, and $n$ is the specific mutation context (for example, mutation from A). Mutation enrichment was calculated as $(observed_{i,n} - expected_{i,n})/(observed_{i,n} + expected_{i,n})$. Rolling mean values were plotted using windows of 51 bp and 21 bp for nucleosome-centred and CTCF-centred plots, respectively. On the basis of bootstrap sampling of the analysed regions, 95% confidence intervals were calculated.

### Multiallelic mutation rates

Aligned reads spanning genomic positions of somatic mutations were re-genotyped using SAMtools mpileup (v1.9)[61]. Genotypes supported by 2 or more reads with a nucleotide quality score of 20 or more were reported, considering sites with two alleles as biallelic, those with three or four alleles as multiallelic. For a defined set of mutations, the background composition is the count of mutations in each of the 64 possible trinucleotide contexts. The count of multiallelic mutations in each of those 64 categories was divided by the corresponding background mutation count and the weighted average of those ratios are reported as the multiallelic rate. As for mutation rates, the vector of weights being the fraction of each trinucleotide in a reference sequence, for example, the composition of the whole genome.

### Replication time

We generated early–late Repli-seq as previously described[62] for two mouse hepatocellular carcinoma-derived cell lines (Hep-74.3a and Hepa1-6, obtained from biohippo and the American Type Culture Collection, respectively, and tested for mycoplasma at source), matching for the study cell type[63]. Furthermore, the tumour from which the Hep-74.3a cell line was derived was induced by a single intraperitoneal injection of DEN at P15 into a C3H/He mouse[64], thus closely matching the DEN-induced tumours in our study. For each cell line, two ENCODE-style biological replicates were generated with individual BrdU labelling and fluorescence-activated cell sorting (FACS) into early and late S-phase fractions for Repli-seq Illumina sequencing library preparation[62]. Sequencing was performed on Illumina NextSeq550 using a Mid-Output v2.5 kit generating 75 bp paired-end reads, producing a total of $1.2 \times 10^8$ read pairs (Hep-74.3a), and Illumina NovaSeq with an S1 flowcell generating 50 bp paired-end reads, producing a total of $3.9 \times 10^7$ read pairs (Hepa1-6). Sequencing reads were mapped using Bowtie2 (v2.4.5) to the C3H_HeJ_v1 reference genome. SAMtools (v1.15.1) was used for alignment quality filtering (-bSq 20), matepair annotation (fixmate -m) and deduplication (markdup -r -s). After confirming concordance, replicates were aggregated and read coverage was calculated

for 10 kb consecutive windows with local smoothing: 50 kb windows with a step-length of 10 kb using the central 10 kb window coordinates using bedtools (v2.30.0) multicov. Windowed read counts were normalized to aggregate library size (tags per million, separately for early ($E$) and late ($L$)) and replication time was taken as the relative enrichment $(E - L)/(E + L)$. For replication time analysis, genomic regions were categorized into 21 quantile bins of replication time relative enrichment, and the median value for each bin used in quantile-based visualization and regression analysis. As the Hep-74.3a cell line is better matched for both strain and treatment, these Repli-seq data were used throughout the paper. The results were replicated with matched analyses of the Hepa1-6 Repli-seq data (Extended Data Figs. 2a and 3h–j).

Repli-seq data are available at the ENA at EMBL-EBI under accessions PRJEB72349 (Hep-74.3a) and PRJEB67994 (Hepa1-6).

### Replication strand bias

Replication fork directionality (RFD) is a relative difference metric that scales from 1 to −1. RFD values > 0 indicate a consensus rightward progressing replication fork, whereas RFD < 0 indicates a consensus leftward progressing fork. RFD can be directly measured at 1 kb resolution from Okazaki fragment sequencing (OK-seq)[65], but such data have only been obtained from cultured cells that can be prepared in large quantities with a high fraction in S phase. Alternatively, RFD has been inferred from Repli-seq data, where RFD is calculated as the derivative of the change in replication time along the genome[12,66], but has lower spatial resolution and is dependent on ad hoc filtering. Here we intersected cell-type-matched Repli-seq RFD with higher resolution OK-seq to ensure high-resolution tissue-matched RFD, and removing the need for ad hoc filtering. Replication time was converted to Repli-seq RFD by taking the average of the difference in replication time of the adjacent upstream and downstream windows.

OK-seq data from mouse activated primary splenic B cells[65] were aligned to the C3H_HeJ_v1 reference genome using Bowtie2 (v2.4.5)[67], quantified using bedtools multicov and RFD calculated as the relative enrichment of reverse ($R$) versus forward ($F$) read coverage $(RFD = (R - F)/(R + F))$[68]. This OK-seq RFD (OK-RFD) metric was calculated for 10 kb consecutive windows to match Repli-seq RFD analysis. Both OK-RFD and Repli-seq RFD measures were categorized into 21 quantile bins. Subsequent mutation rate analysis used OK-RFD quantile classification but was restricted to those that differed from the corresponding Repli-seq RFD by less than 19% of the category range (four bins). Other OK-seq and Repli-seq datasets (Supplementary Table 3) were processed as outlined above, aligning to the GRCh37 reference genome in the case of human-derived sequences. For comparisons between Repli-seq RFD and high-resolution OK-RFD (Extended Data Fig. 2), OK-RFD was calculated as above but in 1 kb consecutive windows and smoothed (R loess function), with the span parameter set to encompass 25 windows.

For each DEN-induced tumour, we identified all RFD segments that were completely contained within lesion segregation mutational asymmetry segments (as defined above) with $|S| > 0.33$. For these segments, we resolved the lesion-containing strand to the template of either the leading or lagging replication strand. A forward strand mutation asymmetry (lesions on the forward strand, $S > 0.33$) and rightward progressing replication fork (RFD > 0) was consensus lagging strand replication over the lesions (Fig. 1e). Similarly $S < −0.33$ and RFD < 0 was also lagging strand replication over lesions. Consensus leading strand replication over lesions is indicated by $S > 0.33$, RFD < 0; or $S < −0.33$, RFD > 0. For the purposes of visualization and the aggregation of equivalent data for increased statistical power, a single replication strand bias (RSB) metric was defined by consistently orienting the strandedness of analyses such that the lesion-containing strand is the reverse strand (compare Extended Data Fig. 3d and 3f). Consequently, new replication and transcription will proceed left to right as the forward strand over a damaged template strand in all RSB figures.

### Gene expression

Paired-end, stranded total RNA-seq from unexposed P15 C3H male mouse livers ($n = 4$, matching the developmental time of mutagenesis) were aligned, annotated and quantified previously[2]. All transcriptome data used were derived from sequence data in Array Express under accession E-MTAB-8518 and are publicly available[2].

The transcription strand of RNA-seq reads was resolved using read-end and mapping orientation using SAMtools (v.1.7.0) and read pairs exclusively mapping within annotated exons were identified using bedtools intersect (v2.29.2)[69]. Intronic read pairs were defined as those mapping within a genic span, derived from a sense strand transcript and not in the exonic set.

For genes with multiple annotated transcript isoforms, the sum of transcripts per million (TPM) over the isoforms was taken as the expression measure (mature transcript, steady state), although similar results — with the same conclusions — were obtained if the maximum for any one isoform was used. Nascent transcription was quantified by counting read pairs with a mapping quality of more than 10 overlapping intronic regions (defined as intronic in all annotated transcript isoforms of the gene) using bedtools multicov (v2.29.2). The read count was normalized to reads per kilobase of analysed intron for each gene in each sequence library, and then normalized to TPM for each library. The final nascent transcript expression estimate per gene was taken as the mean of nascent TPM over replicate libraries. Nascent transcription estimates could be generated for 85% ($n = 17,304$) of protein-coding genes.

Gene-based analyses of mutation rates used the genomic extent of the most highly expressed transcript isoform (the primary transcript) based on P15 C3H mouse liver gene expression. Overlapping genes, defined by primary transcript coordinates, were hierarchically excluded from analysis. Starting with the most expressed gene, any overlapping less-expressed genes were excluded. For the plotting of per-gene, per-strand mutation rates (Fig. 3b and Extended Data Fig. 7b–d), only genes spanning more than 2 million nucleotides of strand-resolved tumour genome in aggregate were shown ($n = 3,392$ genes) to minimize stochastic noise from genes with little power individually to accurately estimate mutation rates. Analyses of aggregating rates by expression bin included all genes within the bin.

Genes with similar estimates of nascent expression were aggregated for analysis of TCR. The sigmoidal distribution relating nascent transcription rate to mutation rate (Fig. 3b) was segmented using linear regression models in the R package Segmented (v1.3-3)[70]. This defined $n = 4,649$ genes with zero or low-detected nascent expression (less than 0.287 TPM) in which reduced mutation rates associated with TCR are essentially undetectable; subsequently, stratum 1 genes (light blue in plots). Genes expressed at a greater rate than segmentation threshold (more than 3.73 TPM) do not show a further decrease in mutation rate with increased expression; these $n = 7,176$ highly expressed genes were defined as stratum 6 (bright red in plots). The $n = 4,005$ genes with intermediate expression (0.287–3.73 TPM) exhibited a log-linear relationship between expression and mutation rate. These were quantile split into strata 2–5, containing approximately 1,000 genes in each strata.

### Genomic intersection and bootstrapping

The intersection and subsetting of genomic intervals were performed using bedtools intersect (v2.30.0). For the removal of genic subregions, overlapping genes were merged (bedtools merge), the regions extended 5 kb upstream and downstream (bedtools slop) and removed from pre-defined intervals using bedtools subtract. Genomic window coordinates were defined using bedtools makewindows. Bootstrap analysis, for example, in mutation rate calculations, resampled genomic intervals that met the selection criteria (for example, RFD category 1, non-genic, minus strand lesions) with replacement to the same total count, within the same tumour.

Multivariate regression analysis was performed using the lm function of R. The reference genome was partitioned into consecutive 10 kb windows, and composition-corrected mutation rates were calculated for each window in aggregate across tumours, separately for forward-strand and reverse strand lesions. Windows in a tumour with an unresolved lesion strand or containing lesion strand transitions were excluded. The fraction of nucleotides within a window overlapping genomic extents expressed at more than 1 TPM were separately calculated for template and non-template strand lesions. Replication time and RSB were both annotated for 10 kb windows by overlap with larger-scale replication time and RSB measures described above, taking the consensus measure (most nucleotide span) for the 10 kb window as the value for regression analysis. The fraction of window nucleotides annotated as genic but excluding regions identified as expressed genes was also included as a predictor variable (residual genic). The relative enrichment measures RSB and replication time were bounded $(-1,1)$, whereas other parameters were fractions bounded $(0,1)$. To ensure equal scaling for regression analysis, RSB and replication time were rescaled to the $(0,1)$ range as $f = 1 - (1 - r)/2$, where $r$ is the relative enrichment metric and $f$ is the rescaled fractional range. Regression models were constructed with mutation rate as the outcome variable and other variables as independent predictor variables.

## Substitution mutation clusters

For each nucleotide substitution mutation, the closest adjacent mutation was found. Null expectations of mutation spacing were generated by sampling mutation positions from other tumours without replacement, to generate an identical number of proxy mutations for each tumour. Initial analysis of mutation spacing indicated strong enrichment of mutations spaced less than 11 nt apart and evidence of enrichment to 100 nt spacing. Mutation clusters were defined as chains of mutations within the same tumour spaced less than $X$ nucleotides from adjacent mutations, with $X = 11$, $X = 101$ or $X = 201$ depending on analysis as indicated. Over 97% of $X = 101$ mutation clusters (29,307 of 30,028) contained only two mutations, 721 clusters contained three mutations and no larger clusters were identified. Of $X = 101$ clusters from proxy-tumour mutations, 100% contained only two mutations.

For each mutation cluster, if it was located within a lesion segregation mutation asymmetry segment, we annotated the mutations within the cluster with respect to the inferred lesion-containing strand. For a genomic segment containing reverse strand lesions, the leftmost mutation site would be the first used as a template for an extending DNA polymerase (as DNA synthesis extends 5′→3′), and the rightmost mutation site replicated over subsequently. These orientations are reversed for a genomic segment containing forward strand lesions. The first replicated-over mutation site for each cluster was annotated distinctly from subsequent sites in the cluster.

Pairs of mutations were phased to the same chromosome by co-occurrence in the same sequencing read. Sequencing reads were extracted from genomic alignments using SAMtools mpileup (v1.7) where they overlapped both genomic positions of a pair of mutations called from the same tumour and separated by 75 nt or fewer. Any sequencing read supporting the called mutant allele with a phred-scaled quality score $\geq 20$ at both mutation positions was taken as support for those mutations occurring on the same chromosome.

Mutation clusters were resolved to preferential leading or lagging strand replication-based RSB measures as defined above. Only the more extreme RSB windows (quantiles 1, 2, 20 and 21; $|RSB| > 0.51$) were considered for comparisons of leading versus lagging strand asymmetry, so that any strand differences were not swamped by regions with low levels of replicative asymmetry. Clusters were defined with $X = 101$ as above, resulting in $n = 2{,}791$ leading strand and $n = 3{,}289$ lagging strand clusters, the difference in count attributable to TCR correlating with leading strand replication (Fig. 1f). Cluster length distributions were compared using a two-sample, two-sided Kolmogorov–Smirnov test

(ks.test function in R). To estimate statistical power for detecting differences in cluster size distribution between leading and lagging strands, we simulated distorted length distributions. The lagging strand length distribution vector was partitioned into clusters of length of 10 or less (short) or more than 10 (long) and randomly sampled with replacement to produce a vector of length matching the leading strand vector. Bias sampling between the short and long cluster bins was controlled by parameter $d$. An undistorted sample of the original distribution would be $d = 0$; whereas 10% of short clusters sampled from the long bin instead of the short bin would be $d = 0.1$. Two-sample, two-sided Kolmogorov–Smirnov tests comparing the original to the distorted sample distribution were applied to 100 bootstraps for each tested value of $d$ (0–0.1 in increments of 0.0005), recording nominal significant difference at $P < 0.05$. The percent of bootstraps supporting nominal significance is the power to detect significance at the tested value of $d$.

## Indel–substitution mutation clusters

Insertion and deletion (indel) mutations were filtered as previously described for base substitutions[2]. For clustering analysis, we only considered indel mutations in lesion strand-resolved autosomal regions where at least three reads support precisely the called mutation. We identified the closest upstream or downstream substitution to each insertion or deletion, called within the same tumour. Null expectation datasets were generated by sampling substitution mutations between tumours as described for substitution mutation clustering above; 100 of these permuted datasets were generated for each tumour. Enrichment of clustering was evaluated by two-sided Fisher's exact test (fisher.test function in R) considering the observed count of indels with a substitution within 100 bp versus the count of indels without a substitution within 100 bp, as compared with the same values estimated from the average of permuted datasets.

For a pair of sequences that differ by a single substitution and a single indel, there can be multiple equally optimal alignments. We identified all cases where there was a substitution mutation within 100 nt of the indel. For each of these, the ancestral and derived sequences were constructed by editing the mutations into the reference genome sequence, and they were oriented to represent the forward strand being newly synthesized over a lesion-containing template (that is, reverse complemented if the reference genome forward strand was the lesion-containing strand). We considered all possible gap placements within those more than 200 bp (2 × 100 flanks + indel length) alignments between ancestral and derived sequence. All alignments that had a single indel-length gap and one substitution were kept, but multiple solutions fractionally weighted, for example, four equally scoring alignment solutions would each be scored $1/4 = 0.25$, whereas an alignment with just one solution would score $1/1 = 1$. For the distance between indel and substitution, and the identity of the substituted, inserted or deleted bases were recorded for each weighted solution. Observed indel–substitution clusters were further filtered to ensure at least two sequence reads supported the existence of both the indel and the substitution in the same read (SAMtools v1.7.0 mpileup), confirming that the mutations occur on the same copy of the same chromosome. This filtering was not possible for the permuted data and thus makes our estimate of mutation clustering in the observed data conservative.

To consider whether substitutions were preferentially located upstream or downstream of the indel with respect to synthesis over the lesion strand, we considered both the full set of indel–substitution mutation clusters and additionally the subset where all equally scoring alignments placed the substitution on a single side of the indel. To generate a null expectation, for each of these datasets, the annotation of the lesion strand was randomly permuted, the distribution of biases from 10,000 permuted datasets were used to derive an empirical $P$ value for each considered set of indel–substitution clusters.

## Transcription-coupled repair

Annotated genes (Ensembl v91) were partitioned into six expression strata based on P15 liver RNA-seq (see above). For each tumour, genes were identified that were wholly contained within a mutation asymmetry segment. Using the annotated transcriptional orientation of the gene and mutational asymmetry of the tumour, each of these genes was categorized as either template strand lesion or non-template strand lesion.

## Mouse colony management

Animal experimentation was carried out in accordance with the Animals (Scientific Procedures) Act 1986 (UK) and with the approval of the Cancer Research UK Cambridge Institute Animal Welfare and Ethical Review Body (AWERB). Animals were maintained using standard husbandry: mice were group housed in Tecniplast GM500 IVC cages with a 12–12-h light–dark cycle and ad libitum access to water, food (LabDiet 5058) and environmental enrichments. Ethical approval, tumour size limits, sample size choice, randomization and blinding for the tumour samples have been previously reported[2]. At least three biological replicates were included for ATAC-seq and ChIP–seq experiments.

## ATAC-seq

Liver samples from P15 mice (matching the developmental time of mutagenesis) were isolated and flash frozen. ATAC-seq was performed as previously described[71], with minor modifications to the nuclear isolation steps (in step 1, 1 ml of 1× homogenizer buffer was used instead of 2 ml; in step 4, douncing was performed with 30 strokes instead of 20). Pooled libraries were sequenced on a NovaSeq6000 (Illumina) to produce paired-end 50 bp reads, according to the manufacturer's instructions. Experiments were performed with three biological replicates.

## ATAC-seq data processing and analysis

ATAC-seq data processing was performed using a Snakemake pipeline (v6.1.1)[72]. Adaptor sequences were removed using cutadapt (v2.6)[73]. Reads were aligned to the reference genome (Ensembl v91: C3H_HeJ_v1 (ref. 59)) using BWA (v0.7.17)[74]. Data from multiple lanes were merged before deduplication; duplicates were marked using Picard (v2.23.8)[75]. Reads overlapping ARC regions were removed using SAMtools (v1.9). Reads aligning to mitochondrial DNA were excluded from further analysis. Read positions aligning to forward and reverse strands were offset by +5 bp and −4 bp, respectively, to represent the middle of the transposition event, as previously described[76]. ATAC-seq peaks were called using MACS2 (v2.1.2)[77] on pooled data containing all replicates. Single-nucleotide-resolution chromatin accessibility was measured and plotted as coverage of ATAC-seq 'tags' (Tn5 insertion sites, adjusted to represent the middle of the transposition event, as described above).

ATAC-seq data are available from Array Express at EMBL-EBI under accession E-MTAB-11780.

## Nucleosome positioning analysis

We used nucleosome positions determined through chemical profiling of mouse embryonic stem cells[39] using a nucleosome centre positioning score to signify the prevalence of nucleosome dyads for a given genomic position. We transferred genome coordinates from mm9 to mm10 using UCSC liftover[78], before using halLiftover (v2.1) to derive expanded C3H-specific coordinates, considering only unique non-overlapping and syntenic positions. The top 4 million dyad positions were selected based on the nucleosome centre positioning score.

The positions and span of the major groove (either facing out or into the histones relative to the dyad) were calculated with the centre of the major groove facing inwards, repeating every ±10.3 bp away from the dyad position and spanning 5.15 bp (ref. 10).

## CTCF ChIP–seq

Livers from P15 mice (matching the developmental time of mutagenesis) were perfused in situ with PBS and then dissected, minced, cross-linked using 1% formaldehyde solution for 20 min, quenched for 10 min with 250 mM glycine, washed twice with ice-cold PBS and then stored as tissue pellets at −80 °C. Tissues were homogenized using a dounce tissue grinder, washed twice with PBS and lysed according to published protocols[79]. Chromatin was sonicated to an average fragment length of 300 bp using a Misonix tip sonicator 3000. To negate batch effects and allow multiple ChIP experiments to be performed using the same tissue, we pooled ten livers for each experiment; 0.5 g of washed homogenized tissue was used for each ChIP, using 20 µg CTCF antibody (rabbit polyclonal; 07-729, lot 2517762, Merck Millipore). Library preparation was performed using immunoprecipitated DNA or input DNA (maximum 50 ng) as previously described[80] with the ThruPLEX DNA-Seq library preparation protocol (Rubicon Genomics). Libraries were quantified by qPCR (Kapa Biosystems), and fragment size was determined using a 2100 Bioanalyzer (Agilent). Pooled libraries were initially sequenced on a MiSeq (Illumina) to ensure balanced pooling, followed by deeper sequencing on a HiSeq4000 (Illumina) to produce paired-end 150 bp reads, according to the manufacturer's instructions; only HiSeq libraries were used for downstream analyses. Experiments were performed with five biological replicates.

To identify ChIP–seq-positive regions, we trimmed the HiSeq sequencing reads to 50 bp and then aligned them using BWA (v0.7.17) using default parameters. Uniquely mapping reads were selected for further analysis. Peaks were identified for each ChIP library and input control using MACS2 (v2.1.2) callpeak with default parameters, and all peaks with a $q > 0.05$ were included in downstream analyses. Input libraries were used to filter spurious peaks associated with a high-input signal using the GreyListChIP R package[81]. Biologically reproducible peaks were identified by merging ChIP–seq peaks defined as above from individual replicates and selecting those that overlapped with two or more individual replicate peaks.

ChIP–seq data are available from Array Express at EMBL-EBI under accession E-MTAB-11959.

## Transcription factor binding site identification and analysis

ChIP–seq data for transcription factors, apart from CTCF (see above), were obtained from Life Science Database Archive (https://dbarchive. biosciencedbc.jp/datameta-list-e.html) with genomic coordinates for the mm9 reference assembly. Liver-specific ChIP–seq was used whenever possible, otherwise files marked with 'All cell types' were used instead (Supplementary Table 2). Genomic coordinates were lifted to mm10 using liftOver, and then lifted to the C3H genome assembly using halLiftover (as above). Overlapping ChIP–seq regions were merged, using the outermost coordinates as the new start/end of regions. FASTA sequences of the regions were extracted using bedtools getfasta (v2.27.1) and used together with non-redundant vertebrate position weight matrices from JASPAR[82] to run FIMO (MEME suite)[83] with default parameters to detect motifs within ChIP–seq peaks. Those motifs were then filtered based on an overlap with ATAC-seq peaks (defined above) to ensure that the analysed set was within open chromatin regions of P15 C3H mouse livers. For CTCF-binding site analysis, in-house generated ChIP–seq data (described above) was used. For wider flank (1 kb) analysis, all motifs (JASPAR matrix profile MA0139.1) within the peaks were retained regardless of ATAC-seq intersection, allowing multiple motifs per ChIP–seq peak.

For high-resolution CTCF and transcription factor-binding site analysis (Extended Data Fig. 8), only one highest-scoring motif per ChIP–seq peak was retained. Similarly, for aggregate transcription factor analysis, only one highest-scoring motif per ChIP–seq peak was retained if it overlapped with an ATAC-seq peak. A total of 129 transcription factors were analysed based on ChIP–seq and position weight matrix

availability, RNA-seq support for transcription factor expression (1 TPM or more) in the P15 mouse liver[2]. In all the analyses, 'bit score' refers to the information content of the whole position. Within the motif, only mutations with the reference nucleotide matching the consensus nucleotide from position weight matrix were retained. In the flanks, mutations from all reference nucleotides were used.

## CTCF structural analysis

High-resolution crystal structures for CTCF zinc fingers complexed with binding site DNA were obtained from the Protein Data Bank (PDB; 5YEL, 5T0U and 5UND)[84,85]. As no single structure contains all 11 CTCF zinc fingers, a composite structure was compiled through alignment using PyMOL (v2.5.2)[86] align function. The PDB 5UND A chain 406–556 was aligned to the PDB 5T0U A chain (root mean square deviation of 1.06 Å); then the PDB 5YEL A chain was aligned to the PDB 5UND chain A (root mean square deviation of 1.3 Å). The composite image (Extended Data Fig. 8d) then shows the PDB 5T0U A chain 289–405, PDB 5UND A chain 406–488 and PDB 5YEL A chain 489–556, which collectively spans CTCF zinc fingers 2–11 inclusive. The bound DNA strands comprise the PDB 5YEL F chain 1–24, PDB 5T0U C chain 7–23, PDB 5T0U B chain 1–18 and PDB 5YEL E chain 5–26.

Protein–DNA contact distance measurements were performed using the Protein Contacts Atlas[87]. Non-covalent interatomic contacts of 3 Å or less between CTCF protein and DNA were considered close contacts. Close contacts of atoms within phosphate groups or deoxyribose were considered backbone, and other DNA contacts were annotated as base contacts. Close base contacts involving atoms expected to acquire DEN-induced mutagenic adducts[23] or structurally equivalent positions in other bases (purines: N6 and O6; pyrimidines: O4, N4 and O2) were annotated as lesion site contacts. Distance measurements were taken separately for each structure (rather than from the composite) and excluded PDB 5T0U nucleotide contacts upstream of binding motif position +1 where this structure substantially deviates from PDB 5YEL. PDB 5T0U is truncated at zinc finger 7, whereas PDB 5YEL extends to zinc finger 11 and makes additional base-specific contacts absent from PDB 5T0U. Close backbone, base and lesion site contacts were reported if the distance threshold criteria were met in any of the three considered structures, although concordance was high in the overlapping regions.

## Histology and image analysis

Digitized histology images of DEN-induced tumours[2] were obtained from Biostudies (accession S-BSST383).

Whole-slide images of tumours that met inclusion criteria (cellularity of more than 50% and DEN1 signature of more than 80%) were annotated in QuPath (v0.2.2)[88] using the polygon tool to include neoplastic tissue and excluded adjacent parenchyma, cyst cavities, processing artefacts and white space. For tumours with multiple transections, only a single whole-slide image was used. Annotations were reviewed for quality by a histopathologist (S.J.A.). Using Groovy in QuPath, annotated regions were tessellated into fixed size, non-overlapping 256 × 256 µm tiles. For segmentation of epithelioid nuclei, a pre-trained StarDist[89] model (he_heavy_augment.zip) was downloaded from https://github.com/stardist/stardist-imagej/tree/master/src/main/resources/models/2D, and an inference instance was deployed using Groovy across the tiles in QuPath, built from source with Tensorflow[90], with a minimum detection threshold of 0.5. Python (v3.9.7) was used for downstream analyses. Data were filtered to exclude extreme outliers: tiles with 43 nuclei per tile or fewer; nuclei with an area of 227.18386 µm or more, circularity of 0.4841 or less, or non-computable circularity were excluded. From the 245 whole-slide images ($n = 237$ mutationally asymmetric tumours and $n = 8$ symmetric tumours), 70,414 tiles were generated, and 9,999,783 nuclei were segmented (post-filtering). To compute inter-nuclear distance, for each nucleus in a tile represented by its $x$–$y$ centroid coordinates, nearest neighbours were identified using the $k$-dimensional tree function from the spatial module of SciPy (v1.7.1)[91]. The Euclidean distance for each nearest neighbour pair was computed using the paired distances function from the metrics module of SciKit-Learn (v1.0.2)[92]. The median nuclear area, median nuclei per tile and median inter-nuclear distances were compared between asymmetric and symmetric tumours using a two-tailed Wilcoxon rank-sum test.

## Symmetric versus asymmetric tumour comparison

Mutationally symmetric tumours (defined above; more than 99% of autosomal mutations in genomic segments with abs(S) < 0.2) were filtered to the subset that met the same inclusion criteria as the other $n = 237$ tumours analysed in this study (more than 50% cellularity (after adjusting for the presence of two genomes) and more than 80% substitution mutations attributed to the DEN1 signature). Eight tumours met this criteria. We subsequently show that these tumours are not whole-genome duplicated, but that they contain both daughter lineages of an originally mutagenized cell (Extended Data Fig. 10b). For each autosomal variant in a tumour, we calculated its VAF quantile position among point mutations in that tumour, using the R ecdf function[93]. The quantile positions (range 0–1) were grouped into consecutive bins of 0.005 unit span, that is, the 0.995–1.0 was the rightmost bin representing the top 0.5% of VAF values for mutations in a tumour. The mutations within a VAF quantile bin were classified as either overlapping or not overlapping with the genomic span of the most highly expressed genes (stratum 6) using the R data.table foverlaps function[94]. The counts of overlapping and non-overlapping mutations from the focal tumour were compared as a two-tailed Fisher's exact test to the equivalent counts aggregated from all asymmetric tumours (excluding the focal tumour in the case of asymmetric focal tumours for the calculation of background expectation). The same analysis was performed in aggregate for all symmetric tumours ($n = 8$) compared with all asymmetric tumours ($n = 237$). The calculations were repeated for each of the 200 consecutive bins to demonstrate the VAF range over which high VAF mutations are preferentially enriched in highly expressed genes specifically in symmetric tumours, as predicted under NER-TRIM.

## Computational analysis environment

Except where otherwise noted, analysis was performed in Conda environments and choreographed with Snakemake[72] running in an LSF 965 or Univa Grid Engine batch control system (Supplementary Table 3). Statistical tests were performed in R (v4.0.5) using fisher.test, ks.test, cor.test and wilcox.test functions for Fisher's exact, Kolmogorov–Smirnov, Pearson's and Spearman's correlation and Wilcoxon tests, respectively. Graphics were generated using R.

## Reporting summary

Further information on research design is available in the Nature Portfolio Reporting Summary linked to this article.

## Data availability

Raw data files for all new datasets are available from Array Express and the ENA at the EMBL-EBI. Early–late Repli-seq accession numbers from the ENA: PRJEB72349 and PRJEB67994. ATAC-seq accession number from Array Express: E-MTAB-11780. ChIP–seq accession number from Array Express: E-MTAB-11959.

## Code availability

The analysis pipeline including Conda and Snakemake configuration files can be obtained without restriction from the repository https://git.ecdf.ed.ac.uk/taylor-lab/lceStrandInteractions.

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

**Acknowledgements** We thank P. Flicek for LCE Consortium management, computational resources and ATAC-seq support; P. Bankhead for supervision of image processing; T. Deegan, V. Seplyarskiy and M. Andrianova for informative discussions; N. Hastie, C. Ponting and W. Bickmore for comments on the manuscript; M. Roller for assistance with data curation; the CRUK Cambridge Institute Core facilities for their valuable contribution: CRUK Biological Resources (A. Mowbray), Genomics (P. Coupland) and Bioinformatics (G. Brown and M. Eldridge); Edinburgh Genomics, The University of Edinburgh for provision of sequencing services; and the European Molecular Biology Laboratory for access to computational resources. This work was supported by the MRC Human Genetics Unit core funding programme grants (MC_UU_00007/11, MC_UU_00007/16 and MC_UU_00035/2), MRC Toxicology Unit core funding (RG94521), Cancer Research UK Cambridge Institute funding (20412 and 22398) and European Molecular Biology Laboratory core funding. Support was also provided from specific research grants: PID2021-126568OB-I00 (CHEMOHEALTH) project, funded by the Spanish Ministry of Science (MCIN, AEI/10.13039/501100011033/); funded by the Wellcome Trust (WT202878/B/16/Z); the European Research Council (615584 and 788937); Helmholtz NCT (DKFZ abteiling B270); the US NIH (R01GM083337); and the MRC equipment award (MC_PC_MR/X013677/1). Edinburgh Genomics is partly supported through core grants from the NERC (R8/H10/56), the MRC (MR/K001744/1) and the BBSRC (BB/J004243/1). J.C. was supported by a Wellcome Trust PhD Training Fellowship for Clinicians (WT223088/Z/21/Z) as part of the Edinburgh Clinical Academic Track (ECAT) programme. M.D.N. is a cross-disciplinary post-doctoral fellow supported by funding from the CRUK Brain Tumour Centre of Excellence Award (C157/A27589). O.P. was funded by a BIST PhD fellowship supported by the Secretariat for Universities and Research of the Ministry of Business and Knowledge of the Government of Catalonia and the Barcelona Institute of Science and Technology. V.S. was supported by an EMBL Interdisciplinary Postdoc (EIPOD) fellowship under Marie Skłodowska Curie actions COFUND (664726). P.-C.W. is supported by the ERC Starting Grant (BrainBreaks 949990) and a Helmholtz Young Investigator grant. S.J.A. received a Wellcome Trust PhD Training Fellowship for Clinicians (WT106563/Z/14/Z), an National Institute for Health and Care Research (NIHR) Clinical Lectureship and a CRUK Clinician Scientist Fellowship (RCCCSF-May23/100001).

**Author contributions** M.S.T. conceived the project and designed the analyses. F.C. and S.J.A. performed the mouse experiments and collected tissue samples. S.J.A. performed the ChIP–seq experiments. S.C. performed the ATAC-seq experiments. M.G., P.-C.W., T.S. and D.M.G. performed and supervised the Repli-seq experiments. C.J.A., L.T., J.L., M.D.N., J.C.V. and M.S.T. designed and performed the computational analysis of genomic data. O.P., V.S. and P.A.G. provided supporting genomic analyses. J.C. and S.J.A. annotated and analysed the histology images. J.C. performed the computational image analysis. N.L.-B., C.A.S., D.T.O., S.J.A. and M.S.T. led the Liver Cancer Evolution Consortium and supervised the work. Research support to D.T.O., S.J.A. and M.S.T. funded the work. D.T.O., S.J.A. and M.S.T. wrote the manuscript, with contributions from C.J.A., L.T., J.L., J.C., M.D.N. and J.C.V. All authors had the opportunity to edit the manuscript. All authors approved the final manuscript.

**Competing interests** The authors declare no competing interests.

**Additional information**
**Correspondence and requests for materials** should be addressed to Duncan T. Odom, Sarah J. Aitken or Martin S. Taylor.

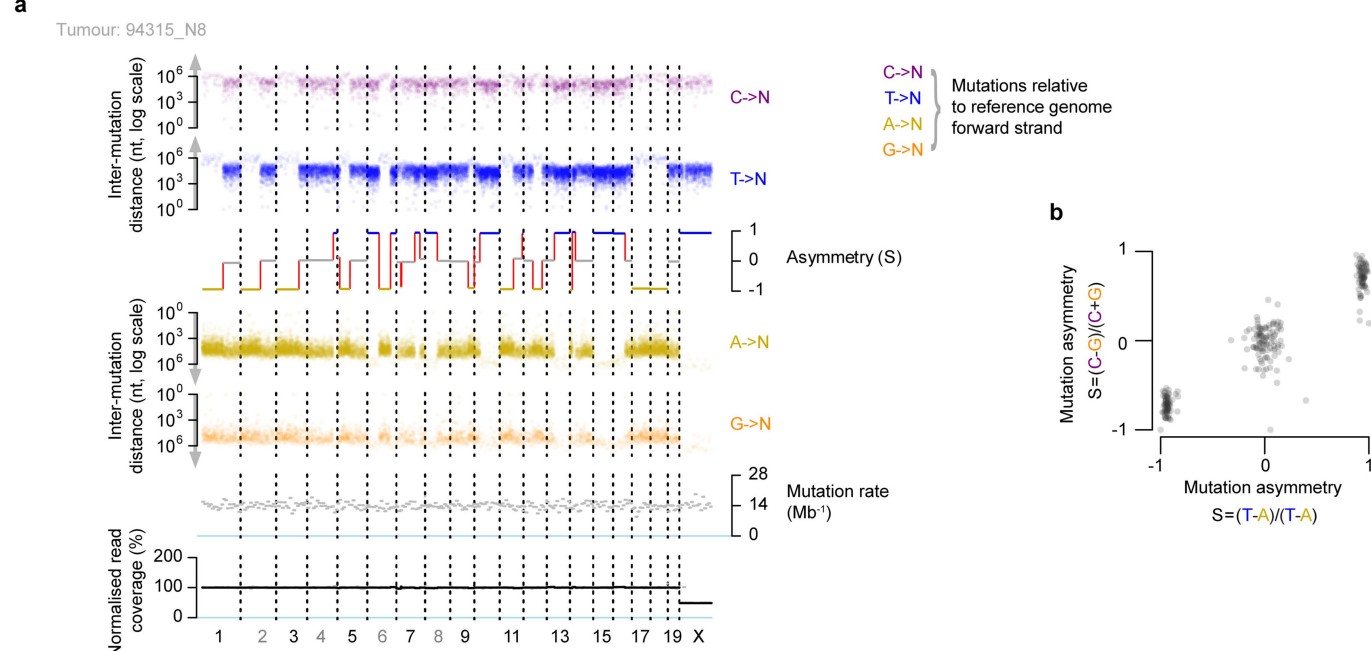

**Extended Data Fig. 1 | Exemplar tumour genome demonstrating mutation asymmetry from lesion segregation. a**, Mutational summary of one DEN induced tumour; the tumour genome represented by the shared x-axis and chromosome boundaries marked with dashed vertical lines. Mutations are called relative to the forward strand of the reference genome and shown as coloured points stratified type (C → N, T → N, A → N, G → N). Y-axis positions show the genomic distance to the next mutation of the same type and plotted on a $\log_{10}$ scale. Mutations of type T → N and A → N are complements of each other and plotted on opposite sides of the asymmetry segmentation track with inverted y-axis orientations (y-axis arrows). The same for C → N versus G → N mutations. Genomic segmentation by T → N/A → N mutation asymmetry is plotted showing genomic segments where mutations have arisen from forward strand lesions (blue), reverse strand lesions (gold), or where one chromosome

has forward and the other reverse strand lesions meaning that they cancel each other out (grey). Hemizygous X chromosomes are always mutationally asymmetric. The asymmetry score is calculated as S = (forward-reverse)/(forward+reverse) where forward and reverse are the sequence composition adjusted rates of T → N and A → N mutations. Both average total mutation rate and read coverage are typically uniform across the autosomal portion of the tumour genomes. **b**, The mutational asymmetry calculated from T → N/A → N mutations (x-axis) and C → N/G → N mutations (y-axis) in 5 Mb windows over the genome is closely correlated, consistent with the interpretation that most mutagenic adducts in these tumours are on T and C nucleotides[2] and supported by reduced mutation rates when T and C are on the transcriptional template strand (Extended Data Fig. 7).

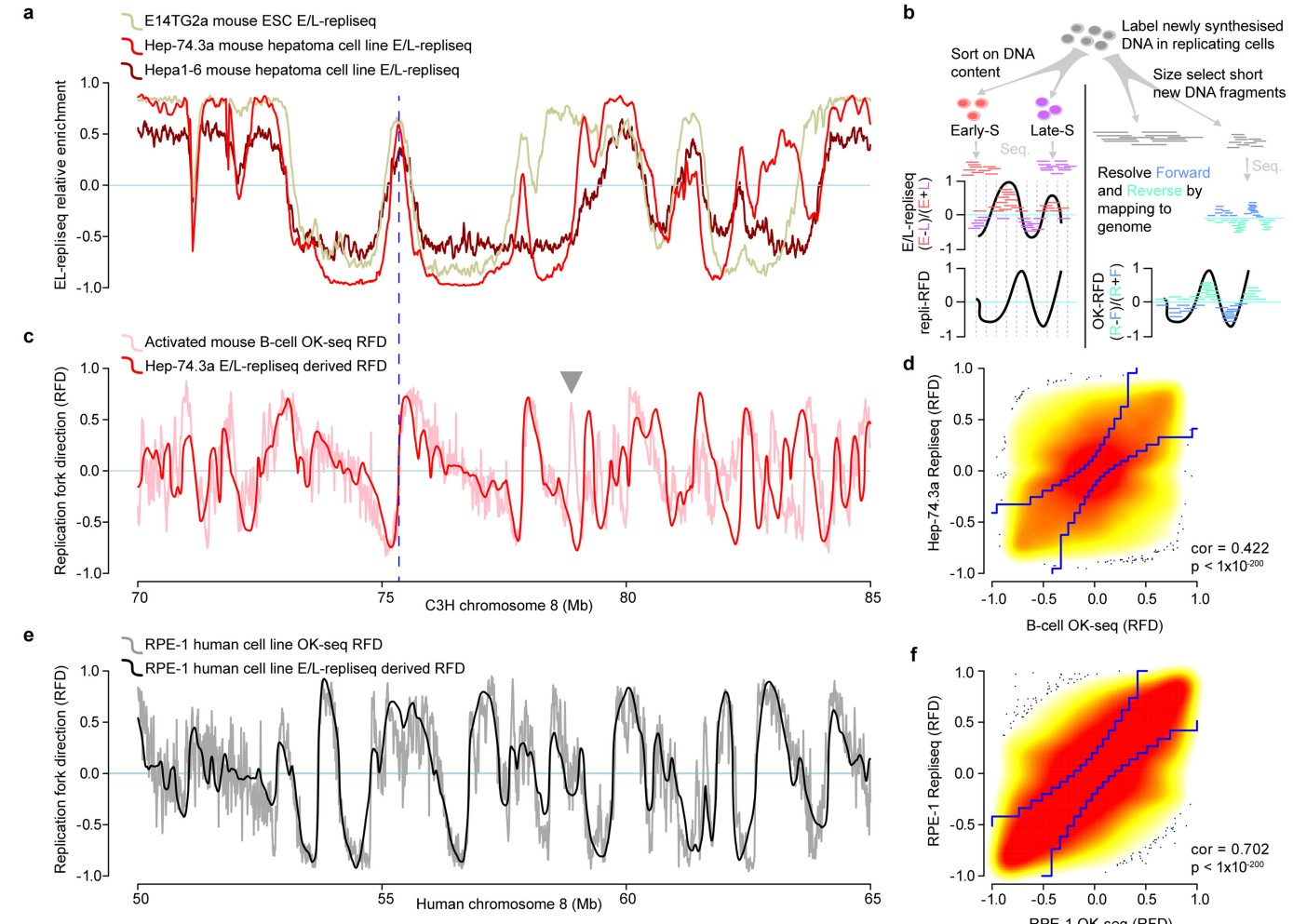

**Extended Data Fig. 2 | Quantifying replication fork directionality.**
**a**, Replication time profile of an example 15 Mb of C3H genome chromosome 8
(x-axis, shared with panel **c**). Curves show early/late (EL) replication relative
enrichment (E and L read counts normalised to their respective library read
depth, then relative enrichment, RE = (E − L)/(E + L)) where more positive values
indicate earlier replication and more negative values indicate later replication.
Replication profiles shown for a mouse embryonic stem cell line (E14TG2a, tan)
and mouse hepatocyte derived cell lines (Hep-74.3a, red; Hepa1-6, brown). Blue
dash line indicates the centre of a strong replication origin region (schematic)
and is projected into panel **c** for comparison. **b**, Schematic illustrating two
alternate strategies to generate replication fork directionality measures (RFD).
Left side, E/L-Repli-seq (top) can be used to derive Repli-seq based replication
fork RFD (repli-RFD; bottom). On the right side, Okazaki fragment sequencing
based RFD (OK-RFD). **c**, Smoothed derivatives of Hep-74.3a E/L-Repli-seq data
(red, panel **a**) provides an RFD estimate. Comparison to OK-seq data from

another differentiated cell type (pink, activated B-cells) shows overall good
concordance but captures some replication profile differences between
cells (grey triangle). **d**, Kernel density plot summarising the genome-wide
correlation of B-cell derived OK-RFD (x-axis) and Hep-74.3a derived repli-RFD
(y-axis), both at 10 kb resolution. Only high-concordance genomic intervals
between blue stepped lines (21 quantile boundaries) were used for RFD based
measures of liver tumour mutation rate. **e**, Validation of the E/L-Repli-seq to
RFD measure in human RPE-1 cells where both OK-seq (grey) and E/L-Repli-seq
(black) has been generated and used to calculate RFD. The curves are shown
over a 15 Mb interval of human chromosome 8 and illustrate a high concordance
of RFD profile. Although both traces are plotted at 10 kb resolution, the
smoothing and processing required to calculate RFD from E/L-Repli-seq
averages out some of the fine grained structure evident in the OK-seq derived
profile. **f**, Kernel density plot summarising the OK-seq (x-axis) and E/L-Repli-seq
(y-axis) RFD estimates for RPE-1 cells, as for panel **d**.

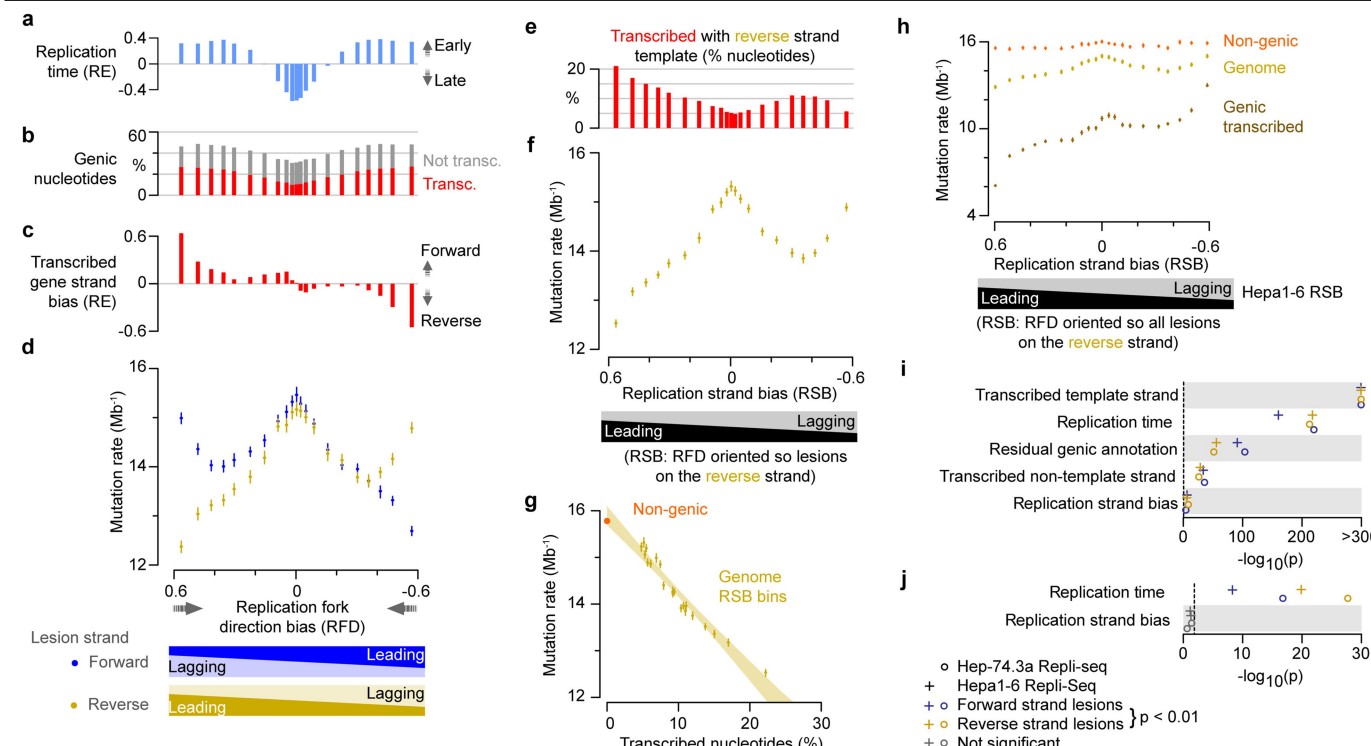

**Extended Data Fig. 3 | Transcription and replication time influence DNA damage induced mutation rate but replication strand bias has negligible impact. a**, Relative enrichment (RE) of early versus late replication time for 21 quantile bins of replication fork direction bias (RFD, x-axis shared with **b-d**). Relative enrichment calculated as RE = (early−late)/(early+late) using the number of nucleotides annotated as early or late replicating in each of the RFD bins. **b**, Percent of genic nucleotides in each quantile bin, stratified as transcribed (red, >1 transcript per million (TPM) in P15 mouse liver) or non-transcribed (grey). **c**, Relative enrichment of strand-biassed transcription across RFD bins (RE = (forward-reverse)/(forward+reverse)) calculated using the number of nucleotides contained within the transcription strand resolved genomic span of expressed genes (panel **b**). **d**, Mutation rate (nucleotide composition normalised) for RFD bins calculated separately for forward strand and reverse strand lesions, 95% C.I. (whiskers) from bootstrap sampling. **e**, Percentage of nucleotides that are transcribed (>1 TPM, P15 mouse liver) in each of the 21 quantile bins of replication strand bias (RSB, x-axis shared with **f**). RSB is the RFD metric but all data oriented so that lesions would be on the reverse strand. **f**, Mutation rates for the 21 RSB bins. **g**, Mutation rates (y-axis) points and RSB bins identical to panel **f**, but x-axis shows the percent of nucleotides with transcription over a lesion strand template, illustrating that transcription using a lesion containing strand is the main determinant of mutation rate. Linear modelling (shaded area 95% C.I.) and extrapolation of this correlation accurately predicts the observed mutation rate in non-genic

regions (orange point). **h**, Mutation rates (y-axis) for the whole genome (gold) stratified into 21 quantile bins of RSB (x-axis). Equivalent analysis is shown for fractions of the genome contained within expressed genes (tan) and non-genic regions (orange). This is a repeat of the analysis shown in Fig. 1f confirming the results using Repli-seq data from a second independent hepatocyte cell line (Hepa1-6 (**h**), rather than Hep-74.3a (Fig. 1f) that is used except where otherwise stated). **i**, Multivariate regression modelling based on 10 kb consecutive genomic windows finds all five tested parameters make nominally significant (right of the dashed line), independent contributions to variation in mutation rate (calculated separately for forward strand and reverse strand lesions, blue and gold, respectively). The predominant contributions are transcription over a lesion containing template strand and to a lesser extent replication time. Residual genomic annotation (annotated genes not meeting the >1 TPM threshold for expression) is notably significant, indicating sub-threshold expression contributes to reducing the mutation rate. The results are highly reproducible, independently using either Hep-74.3a and Hepa1-6 Repli-seq measures (circles and crosses, respectively). **j**, Multi-regression analysis considering only 10 kb segments that are >5 kb from annotated genes, demonstrates significant replication time influences on mutation rate but that replication strand bias does not significantly influence the mutation rate. Forward strand lesions (blue) and reverse strand lesions (gold) calculated separately.

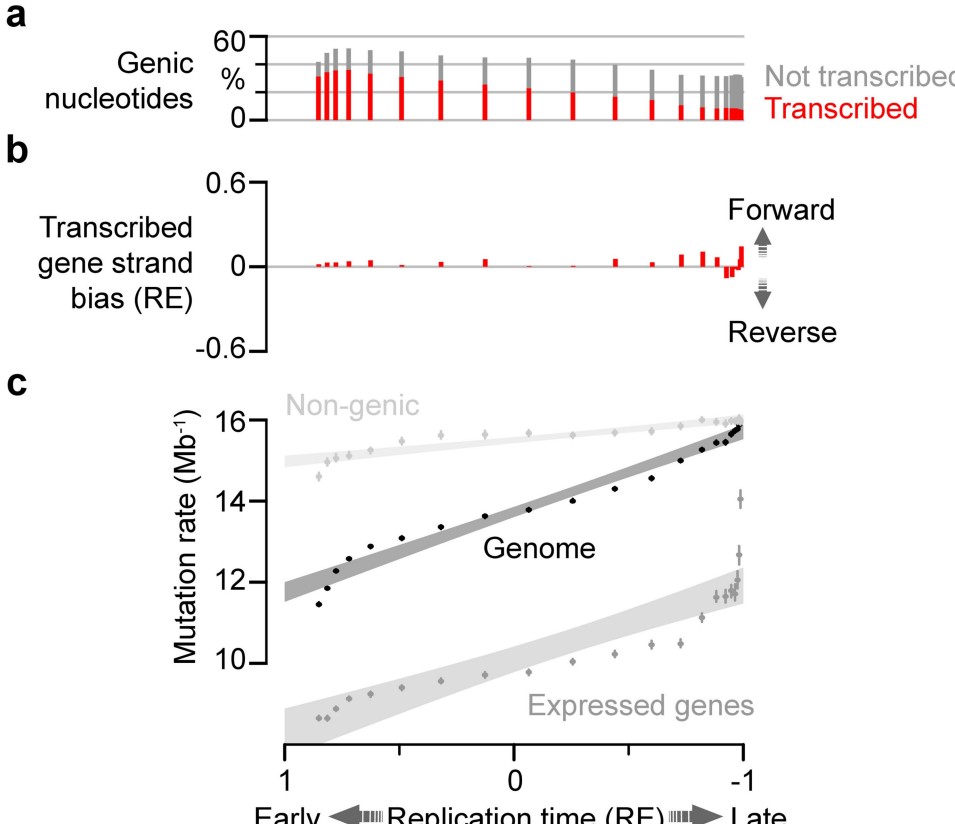

**Extended Data Fig. 4 | Replication time correlates with mutation rate partly independent of transcription. a-c**, The genome was partitioned into 21 quantile bins of replication time, relative enrichment (shared x-axis, RE = (early−late)/(early+late)) **a**, Percent of genic nucleotides in each quantile bin, stratified as transcribed (red, >1 transcript per million (TPM) in P15 mouse liver) or non-transcribed (grey). **b**, Relative enrichment of strand-biassed transcription across replication time bins (RE = (forward−reverse)/ (forward+reverse)) calculated using the number of nucleotides contained within the transcription strand resolved genomic span of expressed genes (panel **a**). **c**, Mutation rates (y-axis) for the whole genome (black, 95% C.I. whiskers). A linear regression 95% C.I. shown as a corresponding shaded area. Equivalent analysis is also shown, restricted to only expressed genes (mid-grey) and non-genic regions (light-grey).

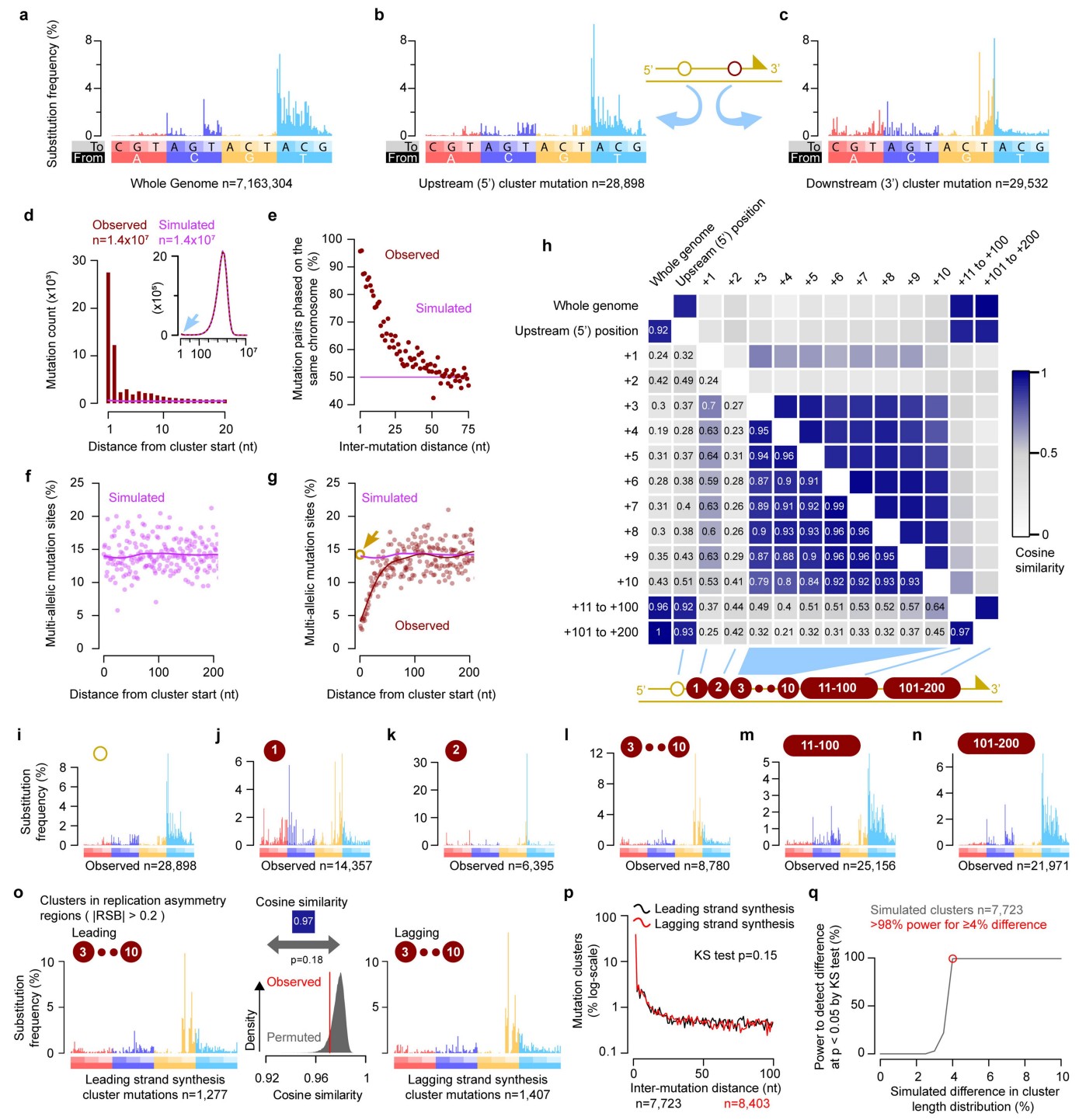

**Extended Data Fig. 5 |** See next page for caption.

**Extended Data Fig. 5 | Tracts of low-fidelity replication downstream of lesion induced mutations. a**, Genome-wide mutation signature of DEN induced tumours. **b**, Signature of mutation cluster upstream (5′) position mutations, oriented so the lesion containing strand is the replication template. **c**, Signature of downstream mutations in the cluster (2.2% of clusters have two downstream mutations). **d**, Frequency distribution of the spacing between adjacent observed (dark-red) and simulated (pink) mutations for all tumours (n = 237). The simulated data were generated by sampling mutations across all other tumours to create proxy tumour datasets with identical mutation counts (see Methods). Main histogram shows only closest spaced mutations, inset graph shows full distribution of both observed and simulated, blue arrow indicates x-axis area expanded in main histogram. Excess clustering of observed mutations (blue arrow) accounts for only 0.8% of the total mutation burden. **e**, Clustered mutation pairs co-occur in the same sequencing read, confirming they are on the same DNA duplex. Expected (pink) is analogous to two heads or two tails from consecutive flips of a fair coin. **f**, Multiallelism is a hallmark of lesion templated mutations[2]. The multiallelic rate (y-axis, fraction of mutation sites with multiallelic variation) for simulated data (pink spots). Curve shows best-fit spline (25 degrees of freedom) for the downstream mutations. **g**, As for (**f**) but showing observed data (red), demonstrating a pronounced and specific depletion of multiallelic variation immediately downstream of the cluster 5′ mutation (yellow circle and arrow). **h**, Heatmap summarising cosine similarity between mutation clusters with different inter-mutation spacing (schematic in lower panel). Upstream (5′) cluster mutations closely match the genome wide mutation spectrum. Mutations 3 to 10 nt downstream of the 5′ mutation share a common signature. **i-n**, Mutation signature profiles for clustered mutations; distance from the upstream mutation (number in brown circle) relate to schematic in **h**. Mutation counts in each category indicated below the plot. **o**, The mutation spectrum of downstream mutations closely matches between leading and lagging strand replication (strongly RSB regions, absolute RSB > 0.2). The observed cosine similarity between mutation spectra is robustly within the range expected by random permutation of mutations between leading and lagging strands (n = $10^5$ permutations, two tailed empirical p = 0.18). **p**, The distribution of mutation cluster length also matches between leading (black) and lagging (red) strands (no significant difference; two sided Kolmogorov-Smirnov test p = 0.15). **q**, Simulations show >98% power to detect a ≥ 4% difference in the distribution of cluster lengths for strongly RSB regions of the genome.

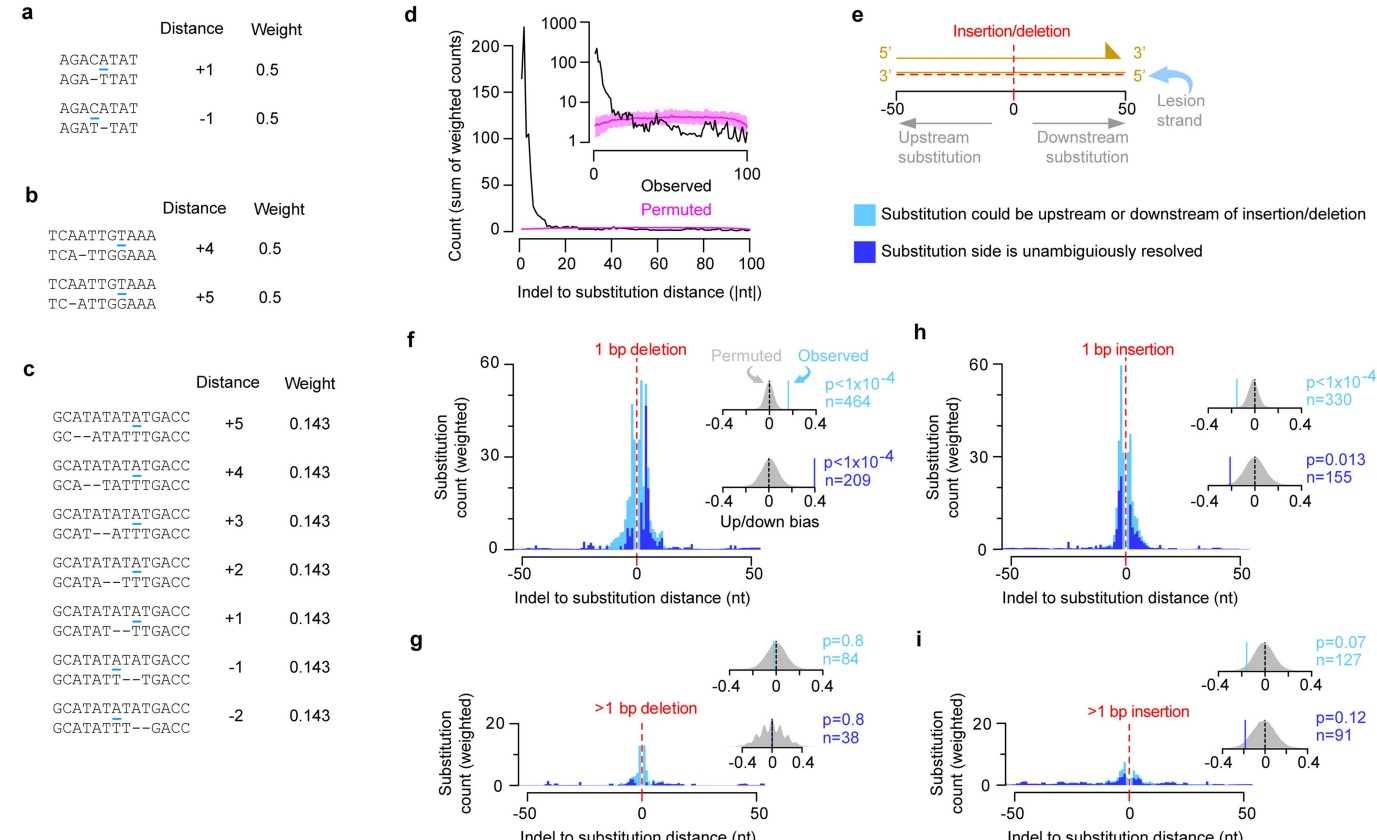

**Extended Data Fig. 6 | DNA damage induces deletion mutations at damaged bases and collateral insertion mutagenesis. a**, A deletion or insertion mutation with a proximal substitution can often be explained by multiple equally scoring alignments. Two example sequences can be aligned with a single gap (dash) and substitution (blue line), in this case with two possible solutions. To avoid systematic biases in gap placement by alignment and mutation calling software, all equally optimal alignments are calculated, the distance between gap and substitution measured for each and count value distributed equally between possible solutions (weight). **b**, As (**a**) but gap and substitution position are not immediately adjacent. **c**, As (**a**) but demonstrating an example with seven equally scoring solutions where the substitution could be assigned to either upstream or downstream of the insertion/deletion. **d**, Frequency distribution of the distance between insertion or deletion (indel) mutations and their closest proximal substitution mutation (black curve), demonstrating a high degree of spatial clustering within 10 bp. The permuted expectation (pink) was calculated by measuring the distance to the nearest substitution in a permuted set of substitutions sampled from other tumours (Methods). Confidence intervals (95%, light pink) on the permuted set were calculated from 100 permuted sets of substitutions. Inset graph shows the

same data plotted with the y-axis on a $\log_{10}$ scale. Counts for both observed and permuted are the sums of the weighted counts for each distance as illustrated in (**a-c**). **e**, Schematic to show how indel and substitution mutation clusters are oriented by the lesion containing strand in subsequent plots, and that the position of the insertion or deletion is set as x = 0. The subsequent plots (**f-i**) also show cases where all optimal alignments agree on the upstream/downstream placement of the substitution relative to the indel (dark blue, e.g. panel **b**) as distinct from where that assignment is ambiguous (light blue, e.g. panel **c**). **f**, Substitutions are strongly clustered around 1 bp deletions and biased towards a downstream location. Inset shows the density plot for 10,000 permutations of the observed data where the assignment of the lesion strand was randomly permuted (grey) compared with the observed level of upstream/downstream bias (calculated as bias = (down−up)/(down+up)). Two-sided p-values were empirically derived from the permutations. **g**, Deletions >1 bp are rarely clustered with substitutions and do not show a significant upstream/downstream bias. **h**, Single base insertions are clustered with substitutions and are significantly biassed to upstream of the insertion. **i**, Longer insertions show similar clustering trends to 1 bp insertions but do not reach statistical significance.

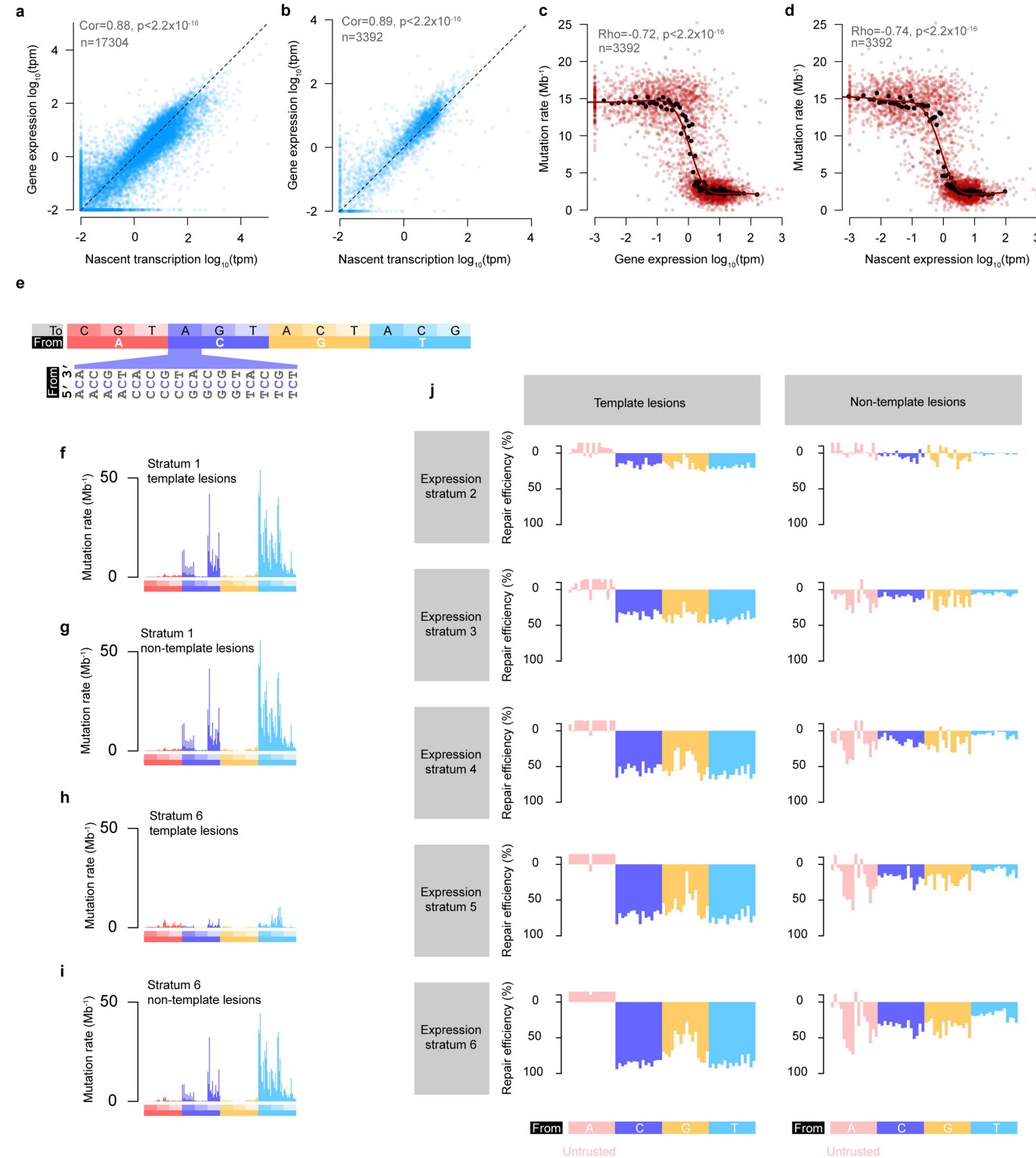

**Extended Data Fig. 7** | See next page for caption.

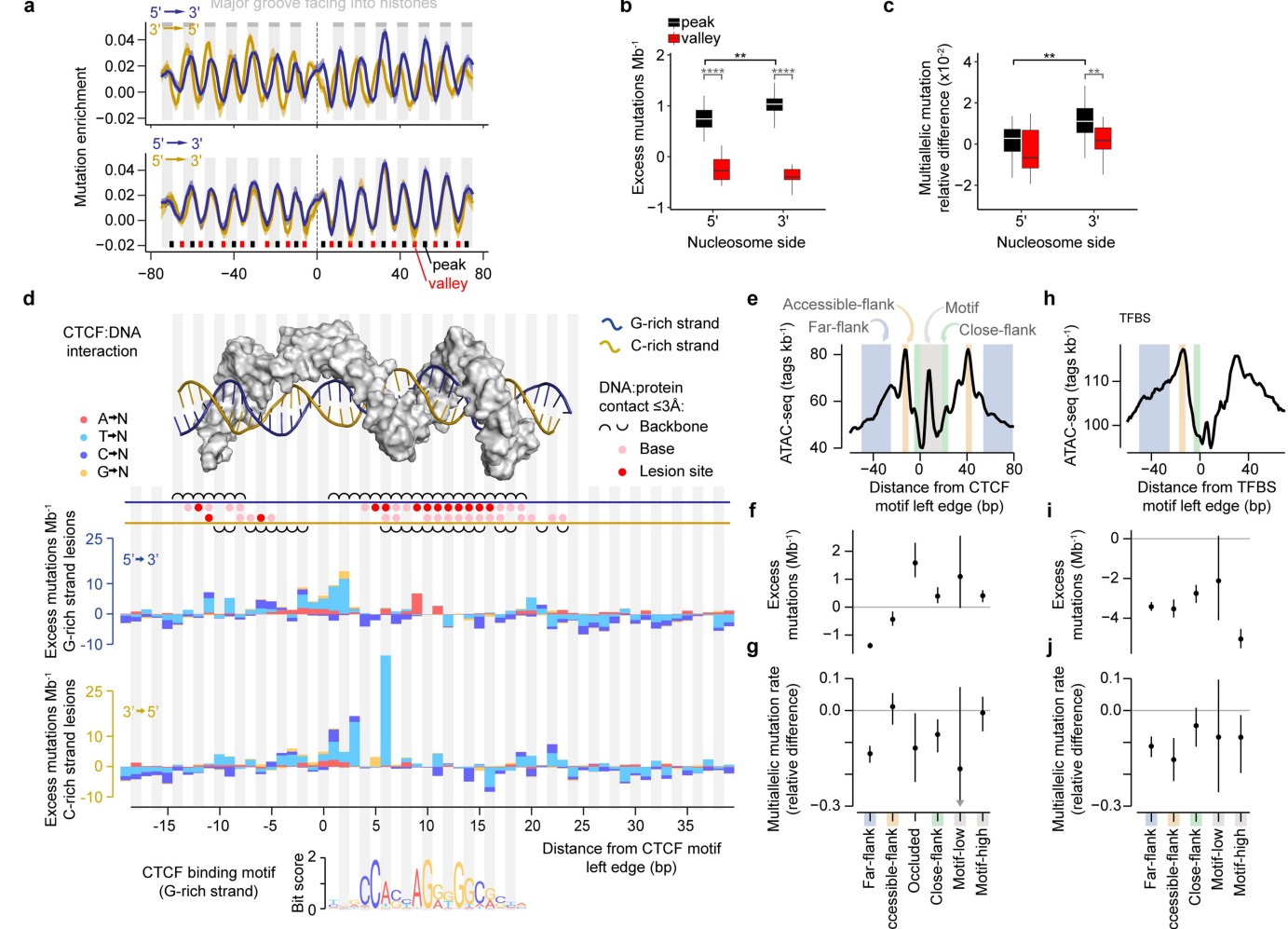

**Extended Data Fig. 8 | Mutation enrichment and depletion at transcription factor binding sites (TFBS). a**, The compositionally corrected mutation rate shows helical (10 bp) periodicity over nucleosomes. Separating the mutation rates by the lesion containing strand (blue, forward; gold, reverse) reveals two partially offset periodic profiles (top panel). Orientating both strands 5′ → 3′ demonstrates that the profiles are mirror images (bottom panel). Mutation rate peaks (black) correspond to regions where the DNA major groove faces into the histones, and valleys (red) where the major groove faces outward. Mutation enrichment is shown with shaded 95% bootstrap confidence intervals (blue, gold). **b**, For the lesion containing strand, mutation rates are significantly higher for the peaks on the 3′ side of the nucleosome dyad than on the 5′ side (significant p-values shown, two tailed Wilcoxon tests). **c**, Comparing the compositionally corrected multiallelic rates shows significantly increased multiallelic variation for the 3′ peaks (significant p-values shown, two tailed Wilcoxon test), indicating the increased mutation rate results from slower repair on the 3′ side of the dyad. **d**, The molecular structure of the CTCF:DNA interface (top) reflects the strand specific mutation profiles of CTCF binding sites (histograms, composition corrected). A composite crystal structure of CTCF zinc fingers 2-11 (grey surface) is shown binding DNA (blue & gold strands) and close protein:DNA contacts (≤3 Å) illustrated below the structure. At nucleotide positions with close contact between CTCF and atoms thought to acquire mutagenic lesions (red circles), the corresponding strand specific

mutation rates are generally lower than genome-wide expectation (y ≤ 0; excepting apparent A → N mutations considered later). Mutation rates are high (y > 0) for nucleotide positions with backbone-only contacts or no close contacts but still occluded by CTCF. CTCF motif position 6 exhibits an exceptionally high T → N mutation rate that cannot be readily reconciled with the structure, but the strand specificity demonstrates it is a consequence of DEN exposure. **e**, The profile of DNA accessibility around CTCF binding sites, defines categories of sequence (shaded areas) considered subsequently. **f**, Mutation rates are higher than genome-wide expectation (y = 0) for CTCF binding motif nucleotides and their close flanks. **g**, This is not reflected in increased rates of multiallelic variation. CTCF occluded positions (positions -5 to 3 of the CTCF motif) show the greatest elevation of mutation rate but evidence of decreased multiallelic variation. Both high information content (motif-high, bit score>0.2) and low information content (motif-low, bit-score ≤0.2) motif positions have high mutation rates. **h**, DNA accessibility around non-CTCF transcription factor binding sites (TFBS) as in **e. i,j**, In contrast to the situation for CTCF, all TFBS categories of sites have suppressed mutation rate compared to genome-wide expectation, y = 0 (**i**), and suppression of multiallelic variation (**j**) indicates enhanced repair. However, high information content motif sites (motif-high) have exceptionally reduced mutation rate not similarly reflected by multiallelic variation, suggesting there may be reduced damage in addition to efficient repair at these sites.

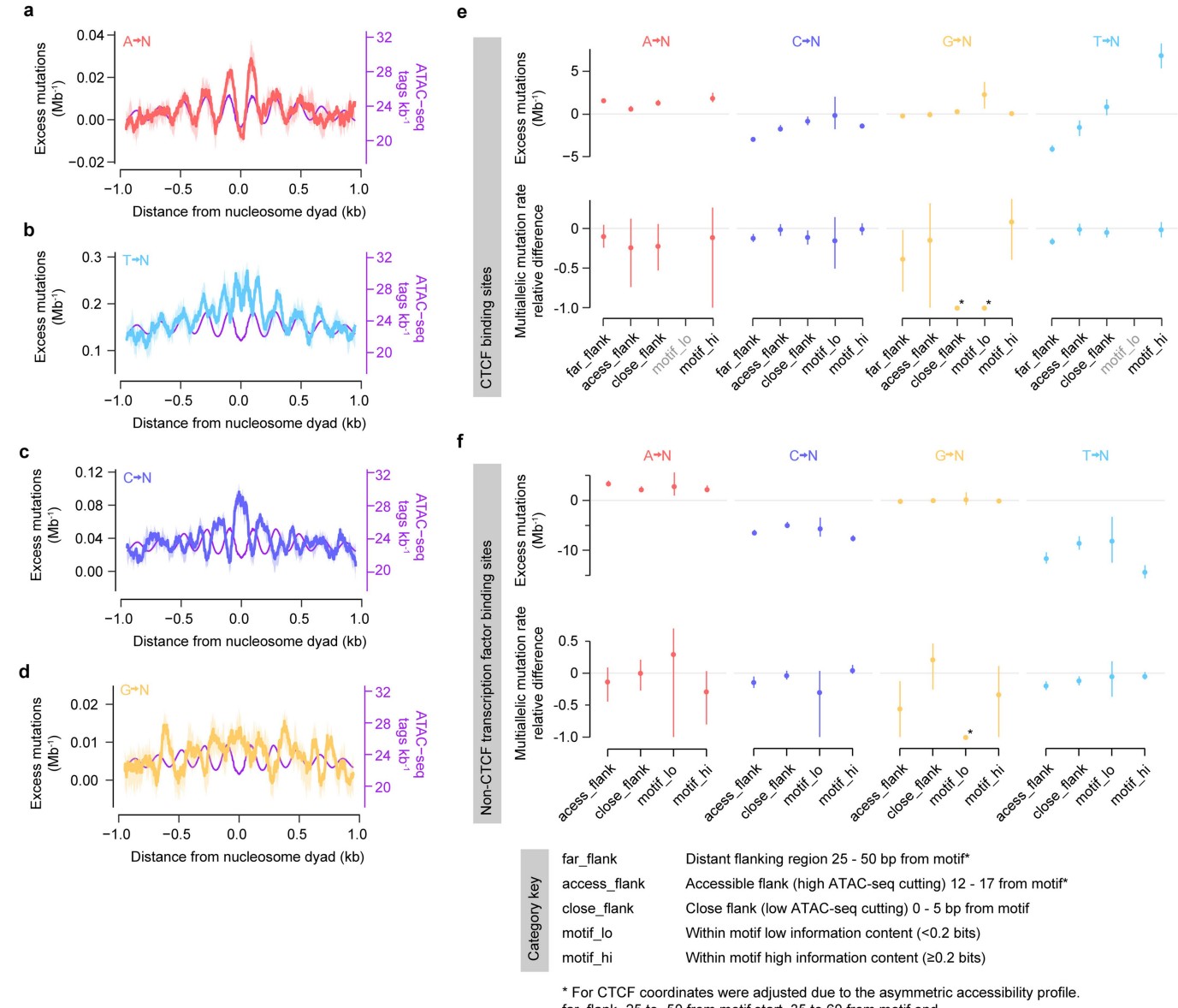

**Extended Data Fig. 9 | Lesion induced mutation patterns at DNA:protein interaction sites. a**, Excess mutations resulting from A lesions in accessible DNA (relative to the genome-wide trinucleotide mutation rate) centred on the nucleosome dyad. DNA accessibility as measured by ATAC-seq (purple; higher values mean more accessible chromatin). Excess mutations are shown with shaded 95% bootstrap confidence intervals. **b-d**, Relative mutation rates as **a**, for apparent T lesions (**b**), C lesions (**c**), and G lesions (**d**); in each case, except A → N mutations, the mutation rate is lower in accessible DNA and higher in less-accessible DNA. **e**, Mutation rates and multiallelic rates for sequence categories (Methods) within, and adjacent to, CTCF binding sites, stratified by

the identity of the inferred lesion containing nucleotide. Point estimate (circles) and bootstrap 95% confidence intervals (whiskers) are shown for the rate difference relative to genome-wide expectation (y = 0, mutations Mb⁻¹ for mutation rates, relative difference metric for multiallelic variation). All rates are adjusted for trinucleotide composition. Instances where the motif_lo category has too few observed or expected mutations to calculate estimates (x-axis label grey) have no data point. Where the observed level of multiallelic variation is zero (asterisk) bootstrap confidence intervals cannot be calculated. **f**, Mutation rates and multiallelic variation for P15 liver expressed transcription factors; plots as in (**e**).

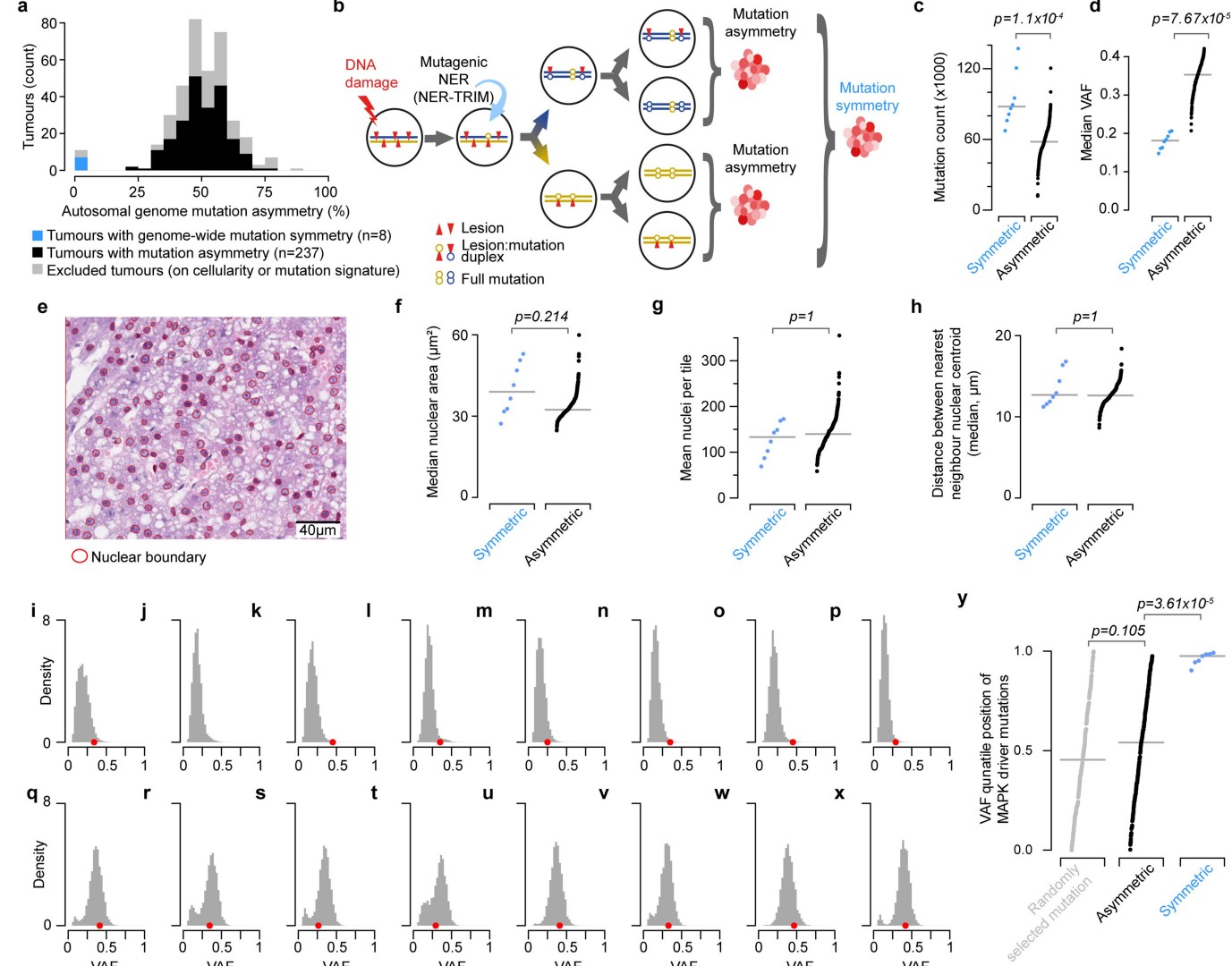

**Extended Data Fig. 10 | Mutagenic nucleotide excision repair. a**, Most DEN induced tumours show pronounced mutation asymmetry across approximately 50% of their genome. Asymmetric tumours meeting inclusion criteria (mutation signature and cellularity thresholds; black) are included in the preceding analyses of this study. In addition, here we include a subset of tumours that were excluded due to the absence of mutation asymmetry (n = 8, blue). **b**, The mutational symmetry of these tumours could be explained if both daughters of the originally mutagenised cell persist (schematic). Mutagenic NER in the first generation of the mutagenised cell could produce mutations at the same base pair in both daughter lineages; such mutations would have approximately double the variant allele frequency (VAF) of mutations confined to one daughter lineage. Whole genome duplication in the first generation of the mutagenised cell could also produce symmetric tumours. **c**, Tumours with symmetric mutation patterns have a significantly higher mutation load than those with asymmetric mutations, consistent with mutations from both mutagenised strands contributing to the tumour. Statistical analysis (p = 1.1 × 10⁻⁴) by two tailed Wilcoxon rank sum test. In panels **c,d,f,g,h** points are individual tumours, bar is median, statistical tests are based on n = 8 symmetric and n = 237 asymmetric tumours, all reported p-values are Bonferroni corrected (n = 5 tests). **d**, The median VAF for mutations in symmetric tumours is approximately half that of asymmetric tumours. Statistical analysis (p = 7.67 × 10⁻⁶) by two tailed Wilcoxon rank sum test. **e**, Automated nuclear detection (red circles) and quantification in an exemplar hematoxylin and eosin stained tumour section (93131_N2). Original digitised magnification x200; scale bar indicated. **f**, Nuclear area is not significantly different between

symmetric and asymmetric tumours (p = 0.215, two tailed Wilcoxon rank sum test), indicating similar DNA content and arguing against mononuclear whole-genome duplication. **g**, The density of nuclei is not significantly different between symmetric and asymmetric tumours (p = 1, two tailed Wilcoxon rank sum test), arguing against both mononuclear and possibly multi-nuclear whole genome duplication. **h**, Internuclear distance is not significantly different between symmetric and asymmetric tumours (p = 1, two tailed Wilcoxon rank sum test), arguing against multi-nuclear whole genome duplication. **i-p**, VAF frequency distributions for symmetric tumours, indicating the VAF of MAPK pathway driver mutations (red points, also in **q-x**). For symmetric tumours, the driver VAFs are strongly right-biased (i.e. high VAF). This is consistent with mutagenic NER copying the same driver mutation site into both daughter genomes of the mutagenised cell, and in turn both daughter lineages (containing either the same driver mutation, or multiallelic driver mutations at the same site) contributing to the resultant tumour. **q-x**, VAF frequency distributions for example asymmetric tumours. **y**, MAPK pathway driver mutations are biassed to the highest VAF values in symmetric tumours but not in asymmetric tumours (p = 3.61 × 10⁻⁵ two tailed Wilcoxon rank sum test, Bonferroni corrected). VAF quantile position (y-axis) indicates the fraction of mutations in a tumour that have lower VAF than the driver mutation (quantile of 1.0 indicates all other mutations in that tumour have a lower VAF). Horizontal bars indicate median VAF quantile position of the focal driver mutations. As a null expectation for comparison, one mutation was randomly selected from each of the asymmetric tumours (grey points).

                            Duncan Odom
                            Sarah Aitken

# Reporting Summary

## Statistics

For all statistical analyses, confirm that the following items are present in the figure legend, table legend, main text, or Methods section.

| n/a | Confirmed | |
|---|---|---|
| ☐ | ☒ | The exact sample size (*n*) for each experimental group/condition, given as a discrete number and unit of measurement |
| ☐ | ☒ | A statement on whether measurements were taken from distinct samples or whether the same sample was measured repeatedly |
| ☐ | ☒ | The statistical test(s) used AND whether they are one- or two-sided *Only common tests should be described solely by name; describe more complex techniques in the Methods section.* |
| ☐ | ☒ | A description of all covariates tested |
| ☐ | ☒ | A description of any assumptions or corrections, such as tests of normality and adjustment for multiple comparisons |
| ☐ | ☒ | A full description of the statistical parameters including central tendency (e.g. means) or other basic estimates (e.g. regression coefficient) AND variation (e.g. standard deviation) or associated estimates of uncertainty (e.g. confidence intervals) |
| ☐ | ☒ | For null hypothesis testing, the test statistic (e.g. *F*, *t*, *r*) with confidence intervals, effect sizes, degrees of freedom and *P* value noted *Give P values as exact values whenever suitable.* |
| ☒ | ☐ | For Bayesian analysis, information on the choice of priors and Markov chain Monte Carlo settings |
| ☐ | ☒ | For hierarchical and complex designs, identification of the appropriate level for tests and full reporting of outcomes |
| ☐ | ☒ | Estimates of effect sizes (e.g. Cohen's *d*, Pearson's *r*), indicating how they were calculated |

*Our web collection on statistics for biologists contains articles on many of the points above.*

## Software and code

Policy information about availability of computer code

| Data collection | Illumina Software Control (ICS) v.3.3.76 |
|---|---|
| Data analysis | Software and versions used in the analysis (fully detailed in Supplemental Table 3): |
| | bwa mem 0.7.17 Li and Durbin, 2009 |
| | Bedtools intersect 2.29.2 Quinlan and Hall, 2010 |
| | cutadapt 2.6 Martin 2011 |
| | FIMO (MEME suite) 5.0.5 Grant et al, 2011 |
| | MACS2 2.1.2 Zhang et al., 2008 |
| | picard 2.23.8 Broad Institute, 2019 |
| | Python 3.9.7 |
| | QuPath 0.2.2 Bankhead et al., 2017 |
| | R 3.6.3 and 4.0.5 R Core Team, 2017 |
| | r-data.table 1.12.8 R Core Team, 2017 |
| | r-GreyListChIP unversionned Brown, 2021 |
| | r-segmented 1.3-3 Muggeo, 2003 |
| | r-sigFit 2.0 Gori and Baez-Ortega, 2018 |
| | Samtools 1.7 and 1.9 Li et al., 2009 |
| | SciKit-Learn 1.0.2 Pedregosa et al., 2011 |
| | SciPy 1.7.1 Virtanen et al., 2020 |
| | snakemake 6.1.1 Mölder et al., 2021 |

StarDist unversionned Schmidt et al., 2018
UCSC liftOver 377 Hinrichs et al., 2006

For manuscripts utilizing custom algorithms or software that are central to the research but not yet described in published literature, software must be made available to editors and reviewers. We strongly encourage code deposition in a community repository (e.g. GitHub). See the Nature Portfolio guidelines for submitting code & software for further information.

## Data

Policy information about availability of data

All manuscripts must include a data availability statement. This statement should provide the following information, where applicable:
- Accession codes, unique identifiers, or web links for publicly available datasets
- A description of any restrictions on data availability
- For clinical datasets or third party data, please ensure that the statement adheres to our policy

Raw data files for all new datasets are available from Array Express (AE) and the European Nucleotide Archive (ENA) at EMBL-EBI. E/L Repli-seq accession numbers ENA: PRJEB67994, PRJEB72349; ATAC-seq accession number AE: E-MTAB-11780; ChIP-seq accession number AE: E-MTAB-11959.

## Research involving human participants, their data, or biological material

Policy information about studies with human participants or human data. See also policy information about sex, gender (identity/presentation), and sexual orientation and race, ethnicity and racism.

| | |
|---|---|
| Reporting on sex and gender | Not applicable |
| Reporting on race, ethnicity, or other socially relevant groupings | Not applicable |
| Population characteristics | Not applicable |
| Recruitment | Not applicable |
| Ethics oversight | Not applicable |

Note that full information on the approval of the study protocol must also be provided in the manuscript.

# Field-specific reporting

Please select the one below that is the best fit for your research. If you are not sure, read the appropriate sections before making your selection.

☒ Life sciences   ☐ Behavioural & social sciences   ☐ Ecological, evolutionary & environmental sciences

For a reference copy of the document with all sections, see nature.com/documents/nr-reporting-summary-flat.pdf

# Life sciences study design

All studies must disclose on these points even when the disclosure is negative.

| | |
|---|---|
| Sample size | Primary data for the study was previously generated and reported whole genome sequence for mouse DEN induced tumours, the full dataset comprised 371 biological replicates (Aitken et al, Nature 2020). ATAC-seq was performed using 3 biological replicates. ChIP-seq was performed using 5 biological replicates. |
| Data exclusions | Except where otherwise stated (within the final results section), analyses were confined to n=237, clonally distinct DEN induced tumours that met the combined criteria of: (i) not labelled as symmetric, (ii) tumour cellularity >50%, and (iii) >80% of substitution mutations attributed to the DEN1 signature by sigFit (v.2.0). Data exclusions were pre-defined based in previously reported frequency distributions of mutation spectra and variant allele frequency in the primary data (Aitken et al., Nature 2020). |
| Replication | Replication through the orthogonal validation of conclusions. Mutation rates and mutation spectra (successful). Mutation rates and multiallelic variation (successful). Symmetric tumour validation of NER-TRIM (successful). DNA accessibility findings replicated at nucleosomes, CTCF binding sites and transcription factor binding sites. Repli-Seq analysis was replicated with two independent cell lines. |
| Randomization | This study did not involve a randomised case:control study design. Randomisation was used to permute genes and tumour classifications for permutation based analysis (using the R sample function without replacement). Randomisation was used for bootstrap analyses (R sample function with replacement). Analysis code provided uses a defined seed for random number generation for reproducibility but equivalent results in each case were obtained using pseudo-randomly generated seeds. |
| Blinding | Researchers were blinded during histological scoring of tumours in prior work that generated the primary data for this study (Aitken et al., Nature 2020). Blinding was not relevant for genomic analyses as all processing was automated and all samples/genomic-loci meeting pre- |

defined inclusion criteria were processed.

# Reporting for specific materials, systems and methods

We require information from authors about some types of materials, experimental systems and methods used in many studies. Here, indicate whether each material, system or method listed is relevant to your study. If you are not sure if a list item applies to your research, read the appropriate section before selecting a response.

## Materials & experimental systems

| n/a | Involved in the study |
|---|---|
| ☐ | ☒ Antibodies |
| ☐ | ☒ Eukaryotic cell lines |
| ☒ | ☐ Palaeontology and archaeology |
| ☐ | ☒ Animals and other organisms |
| ☒ | ☐ Clinical data |
| ☒ | ☐ Dual use research of concern |
| ☒ | ☐ Plants |

## Methods

| n/a | Involved in the study |
|---|---|
| ☐ | ☒ ChIP-seq |
| ☒ | ☐ Flow cytometry |
| ☒ | ☐ MRI-based neuroimaging |

## Antibodies

| | |
|---|---|
| Antibodies used | CTCF antibody (rabbit polyclonal, Merck Millipore 07-729, lot 2517762). |
| Validation | From Merck Millipore : https://www.merckmillipore.com/GB/en/product/Anti-CTCF-Antibody,MM_NF-07-729 <br> Routinely evaluated by Western Blot. <br><br> Western Blot Analysis: <br> A 1:1000–1:5000 dilution of this lot detected CTCF in HeLa nuclear extract. A previous lot detected CTCF in K562 nuclear extract (data not shown). <br> Application - Use Anti-CTCF Antibody (Rabbit Polyclonal Antibody) validated in ChIP, WB, ChIP-seq to detect CTCF also known as 11-zinc finger protein, CCCTC-binding factor. <br> The same antibody has also been used and validated in previous publications, for example https://doi.org/10.1186/s13059-018-1484-3 |

## Eukaryotic cell lines

Policy information about cell lines and Sex and Gender in Research

| | |
|---|---|
| Cell line source(s) | The Hepa1-6 cell line was obtained from ATCC (accession: CRL-1830). The Hep-74.3a cell line was obtained from biohippo (accession BHC18001713). |
| Authentication | Genome sequencing confirmed expected mouse strains for both cell lines. |
| Mycoplasma contamination | Cell lines were newly acquired from sources for this study, tested and found negative for mycoplasma at source. |
| Commonly misidentified lines (See ICLAC register) | None |

## Animals and other research organisms

Policy information about studies involving animals; ARRIVE guidelines recommended for reporting animal research, and Sex and Gender in Research

| | |
|---|---|
| Laboratory animals | Mus musculus C3H/HeOuJ strain postpartum 15 days (P15), male. |
| Wild animals | This study did not involve wild animals. |
| Reporting on sex | As the study built on a large pre-existing dataset of tumours induced specifically in male mice (Aitken et al., Nature 2020), matched animals were use for the new sequence data reported in this study. Tissue samples for ATAC-seq and ChIP-seq were collected contemporaneously from the same colony as the original study. |
| Field-collected samples | This study did not involve field-collected samples. |
| Ethics oversight | Animal experimentation was carried out in accordance with the Animals (Scientific Procedures) Act 1986 (United Kingdom) and with the approval of the Cancer Research UK Cambridge Institute Animal Welfare and Ethical Review Body (AWERB). |

Note that full information on the approval of the study protocol must also be provided in the manuscript.

# Plants

| Seed stocks | Not applicable |
|---|---|

| Novel plant genotypes | Not applicable |
|---|---|

| Authentication | Not applicable |
|---|---|

# ChIP-seq

## Data deposition

☒ Confirm that both raw and final processed data have been deposited in a public database such as GEO.

☐ Confirm that you have deposited or provided access to graph files (e.g. BED files) for the called peaks.

| Data access links
*May remain private before publication.* | ChIP-seq data submitted to Array Express, accession E-MTAB-11959. ttps://www.ebi.ac.uk/arrayexpress/experiments/E-MTAB-11959/
ATAC-seq data are available from Array Express at EMBL-EBI under accession E-MTAB-11780. https://www.ebi.ac.uk/arrayexpress/experiments/E-MTAB-11780/ |
|---|---|
| Files in database submission | *Provide a list of all files available in the database submission.* |
| Genome browser session
(e.g. UCSC) | *Provide a link to an anonymized genome browser session for "Initial submission" and "Revised version" documents only, to enable peer review. Write "no longer applicable" for "Final submission" documents.* |

## Methodology

| Replicates | 5 biological replicates. |
|---|---|
| Sequencing depth | ChIP-seq was 150bp paired end reads. Semi-colon delimited fields for CTCF ChIP-seq libraries are:
library_identifier; total_read_count; uniquely_mapped_read_count
do17757; 40,870,808; 33,468,454
do17797; 51,230,864; 41,989,740
do17839; 22,059,191; 18,232,544
do18187 44,945,921 36,979,824
do18326 51,904,233 43,086,293 |
| Antibodies | CTCF antibody (rabbit polyclonal, Merck Millipore 07-729, lot 2517762) |
| Peak calling parameters | To identify ChIP-seq positive regions, we trimmed the HiSeq sequencing reads to 50 bp and then aligned them using BWA (v.0.7.17) using default parameters. Uniquely mapping reads were selected for further analysis. Peaks were identified for each ChIP library and input control using MACS2 (v.2.1.2) callpeak with default parameters, and all peaks with a q-value >0.05 were included in downstream analyses. Input libraries were used to filter spurious peaks associated with a high input signal using the GreyListChIP R package (Brown 2021). |
| Data quality | Biologically-reproducible peaks were identified by merging ChIP-seq peaks defined as above (with an FDR q-value >0.05 ) from individual replicates and selecting those that overlapped ≥2 individual replicate peaks. |
| Software | MACS2 (v.2.1.2) callpeak with default parameters |

