## [Peer Review file · Nature]

Manuscript Title: Strand-resolved mutagenicity of DNA damage and repair

Reviewer Comments & Author Rebuttals

Reviewer Reports on the Initial Version:

Referees' comments:

Referee #1 (Remarks to the Author):

In the manuscript "Strand-resolved mutagenicity of DNA damage and repair" Martin Taylor and colleagues describe a follow-up to their 2020 Nature paper "Pervasive lesion segregation shapes cancer genome evolution", in which they established sequencing of liver tumors caused by a single onslaught of nitrosamine induced DNA damage. The key strength of this system is that repair of these DNA lesions will occur only over several cell cycles, during which they will give rise to mutations on the opposite strand, which will then be seen in the tumor progeny. In this paper, the authors use intricate bioinformatic analysis to investigate DNA replication and repair and in how far these processes lead to mutagenesis in different locations of the genome. They find that leading and lagging strand replication cause similar rates of mutagenesis and that an apparent bias is actually caused by the co-orientation of transcription and transcription-coupled repair in the genome. Notably, within open chromatin repair is upregulated, not only through TCR on transcribed strands, but also generally, suggesting a general regulation through chromatin. Investigating clustered lesions/mutations the authors show that TCR is apparently stochastic suggesting that DEN lesions do not form an efficient block to transcription. With regard to genome architecture the authors find that genome accessibility primarily regulates repair, likely by regulating accessibility of the repair machinery (histones, but seemingly not TFs in general). An exception appears to be the binding of CTCF, which increases the initial damage at specific sites within the CTCF footprint. Lastly, the authors find a novel mutation signature that specifically occurs at highly repaired regions suggesting that specifically during the first division repair may be mutagenic as both DNA strands contain (clustered) DNA lesions.

This is a fantastic study, which highlights the power of genomics if applied to the right system. It significantly extends the previous paper giving detailed insights into the mechanism of replication, damage tolerance and repair and how those processes are associated with mutagenesis. I therefore only have limited experimental points, which I suggest to be addressed before publication. My main criticism concerns the presentation of the manuscript, which in my eyes needs improvement.

Technical points:

Fig. 1f – Why not show this analysis for both forward and reverse strand lesions (similar to Fig. 1 e, can be colour-coded)? Does “oriented so lesions on the reverse strand” mean forward lesions are included as well? If not, they should be.

Fig. 2 – Should the ordering of upstream and downstream lesions not be done only for those regions

that have a strong bias in fork direction? If this was done, I did not find the according information.

Regarding the manuscript:

Personally, it took me a long time to get through the manuscript and to understand data and conclusions. While undeniably there is a level of complexity in this study that cannot be changed – I think the authors need to improve presentation for the broad audience of Nature.

This includes (i) clarifying the experimental systems (e.g. DEN was not explained throughout the MS), (ii) clarifying the analysis (the MS sometimes contains phrases such as “after accounting for TCR” which are ambiguous; or mutation rates given as “nascent transcripts per million”), (iii) including whenever possible graphical illustrations (those included are very useful, but more would be desired).

In contrast to the previous study, this study is not about how DNA damage by nitrosamines causes liver cancer, but essentially it is about how cells deal with and repair those lesions at different places in their genome. Indeed, this may be the most powerful model in the mammalian system. While I generally find the authors conclusions convincing, they are very much based on deduction. Therefore, I think the authors need to put their story better in the context of the published literature including biochemical studies and experiments in model organisms. In this sense I would consider re-writing the discussion chapter. While I enjoyed reading this chapter and found that it nicely highlighted the brilliant “detective work” done in this study, I would have wished for a bit broader discussion of the key findings (e.g. a key result is mutagenic NER – it should therefore be discussed in more detail what others evidence exist about NER being mutagenic).

Referee #2 (Remarks to the Author):

In this manuscript, Anderson et al. provide a natural continuation of their prior work (Aitken et al., Nature, 2020) on chromosome-scale phasing of somatic mutations. In the current paper, the authors leverage strand-phased mutation patterns and multiallelic variation to better understand the strand asymmetry of replication and transcription. Indeed, the analyses reveal several unexpected and surprising findings, including the undistinguishable DNA repair fidelity of the leading and lagging replication strands as well as the high DNA damage tolerance observed during transcription.

I am torn about this manuscript. On one hand, the results (if correct), are potentially quite interesting. On the other hand, I do find the paper to be overstated and I have a number of concerns related to the biological interpretation and to the performed computational analyses.

First, biologically, the manuscript is written as finding a general result related to DNA damage and repair of all types of small DNA adducts. However, the authors are using a N-nitrosodiethylamine (DEN) mouse model which is a very specific mutagen. Perhaps, the presented results are correct for DEN (and maybe for many nitrosamines) but I am not sure whether this is the case for other small DNA adducts. For example, I am not sure whether their results will generalize for small alkylated

bases such as 7-methylguanine. There is just not enough data to support the results as generalizable for other small DNA adducts. If the authors want to make this point, they should explore at least two other distinct classes of small DNA adducts. Otherwise, their results should be reported as specific for DEN.

Second, some of the findings reported as novel have been previously shown and parts of the figures are quite similar to ones from the authors' previous manuscripts. For example, figure 1a-b in the current manuscript is almost identical to figures from their previous manuscript (Aitken et al., Nature, 2020). Similarly, the finding that mutations are largely shaped by the influence of DNA accessibility has been previously reported and some of the authors on this study have reviewed it (Gonzalez-Perez et al., Cell, 2019).

Third, the manuscript makes a number of mechanistic claims based on the computational analyses. For example, the authors speculate that during the repair of small DNA adducts, the same translesion polymerases are recruited by both the leading and lagging strands replication machineries. Indeed, this is reasonable to infer from the performed analysis, but it is never experimentally validated. Further, this is not the only explanation of the observed data. For example, different translesion polymerases can be recruited to the leading and lagging strands provided that they have similar fidelity of repair on these strands. As such, parts of the manuscript report interesting biological models but they do lack any mechanistic or experimental validation.

Lastly, in regard to the performed computational analyses, there are a number of potential concerns that might be affecting the reported results:

-- The somatic mutations are taken from mouse liver cancers. However, replication fork directionality was determined based on mouse primary splenic B cells. Also, the data for B cell was generated using a different reference genome which required the authors to lift it over to the C3H_HeJ_v1 reference assembly. The authors should generate Repli-Seq for some of the tumors (if material is available) or, at least, for mouse hepatocytes as replication fork directionality in hepatocytes could differ from the one observed in B cells.

-- In parts of the manuscript, the authors compared the observed mutation patterns to the expected ones. However, in most cases, it is not clear how these expected patterns have been generated. For example, figure 2a-b has an expected straight line without any confidence intervals. I am unsure how this expected value was derived. In contrast, figure 5a demonstrates extreme local distortions but it is unclear whether these are expected by chance due to the DNA strand sequence in accessible DNA regions or due to the actual repair of this sequence.

-- One of the more provocative findings in the paper was that, after accounting for transcription-coupled repair, there is no residual effect of replication strand on mutation rate. Again, it is quite unclear how the authors are accounting for transcription-coupled repair in regard to replication strand especially considering data from B cells was used to determine replication fork directionality. The authors should use better replication reference and they ought to focus on regions of the genome that are not transcribed to evaluate whether there is replication strand bias different than the one expected by chance. This result should be contrasted to regions of the genome that are

commonly transcribed.

Referee #3 (Remarks to the Author):

Anderson et al.

Strand-resolved mutagenicity of DNA damage and repair

The present manuscript extends the work of this team examining mutagenesis induced by a specific hepatic mutagen diethyl nitrosamine (DEN), leveraging a large sequencing dataset of tumours induced in mice. Their previous work (Aitken et al Nature 2020, ref 2) very nicely demonstrated that DNA lesions can persist through multiple rounds of replication leading to recurrent sites of mutation that in turn leads to strand asymmetric allelic variation. Here the authors use this approach coupled with the properties of DEN as a mutagen to infer which strand was initially damaged. From this analysis they draw several conclusions:

- the fidelity of DNA damage tolerance is the same on both strands, despite the known asymmetry of replication fidelity.
- that there is enhanced repair of both strands in transcribed regions, not just the template strand, likely due to the generally more accessible nature of active genes.
- that there is a gradient of transcription-coupled repair down a gene, which is likely due to the sequential repair of lesions from the 5' end as RNAPII encounters them
- that there is mutagenesis associated with the NER reaction itself, at least in this context in which the mutagenesis being studied was induced by a single very large dose of mutagen.

These are findings of potential broad interest. However, there are some significant limitations to the extent to which the data presented fully supports the generalisability of the conclusions. In addition, there are some statements that I think go beyond what is known or presented and some points that would need to be presented more clearly for a general reader.

Major points

My first point concerns the generalisability of the findings. The study is limited to a single DNA damaging agent, DEN. While I appreciate that the generation of this dataset represents an enormous investment, I believe that it is crucial to know to what extent the findings stand up when similar datasets generated with other mutagens are analysed e.g. those presented in Riva et al 202 (PMID 32989322). This is important as there are significant differences in the conclusions drawn here with regard strand specificity of lesion bypass-dependent mutagenesis to those in Ref 5 (Seplyarskiy et al. Nat. Gen. 2019), which focusses on UV-induced damage rather than an adduct. There is clearly diversity in the mutagenic responses, both in terms of enzymology and outcome, that are largely dictated by the mutagen and lesions it creates. Thus, drawing the sweeping conclusions that are made here from a study of DEN, especially when the results with UV appear quite distinct, seems unsafe.

The biology of DEN and the excision and bypass of its adducts is relatively poorly understood. It

generates a very wide range of damage to both the phosphate backbone and bases. The authors argue that that 75% of DEN lesions occur at T, but DEN is also able to cause significant lesions at A (e.g. O4-ethylidA) and G (e.g. N7-ethylidG and O6-ethylidG) and C (e.g. O2-ethylidC and N3-ethylidC). Why is it only that T bases are persistent and able to cause persistent multiallelic variation?

The broad spectrum of base lesions caused by DEN is also relevant in thinking about mutations that appear downstream of what is inferred to be the reference lesion. These appear to largely be due to incorporation of A opposite template G, which is inferred to represent naturally unforced errors by a TLS polymerase, but could also represent A-rule bypass of e.g. an abasic site created by excision of a G lesion. Is so little mutation seen at G because it is so very efficiently excised? If so, what is the evidence for this? Such considerations are particularly important given the way in which the experiment was set up – induction of hepatic carcinoma with a single very high dose of DEN, which would be expected to produce a relatively dense distribution of lesions in the genome, increasing the chances of clustered mutations, which are indeed invoked by the authors as an explanation for mutagenesis associated with NER.

The idea that some mutagenesis occurs during NER. This is perfectly plausible and is supported by their data. However, it is unclear whether it is due to clustered damage or simply that some TLS polymerases e.g. pol kappa are clearly able to carry out the gap filling reaction, which may provide an advantage in the low dNTP environment of G1 (Ogi & Lehmann Nat Cell Biol 2006). This should be discussed.

Related to this, I was surprised that there was no analysis of indels in these datasets. This is relevant given the propensity for DEN to damage the phosphate backbone (see e.g. Li and Hecht (doi: 10.3390/ijms23094559). In addition, Pol kappa, in particular, is associated with -1bp frameshifts during lesion bypass. It would be interesting to know whether such events are associated with the events under study.

The potential role of MMR is not adequately explored. In the context of replication-dependent strand asymmetric mutation, MMR at least in yeast, plays an important role in balancing out the differential mutagenesis of pol epsilon and pol delta (Lujan et al. PLoS Genetics 2012; Zhou et al. NSMB 2021). Further MMR is known to act on TLS-induced errors (see e.g. work of Niels de Wind). Indeed, does MMR have any role in direct recognition or processing of lesions created by DEN, or on the abasic sites created by excision?

The novelty of the data presented in Figures 5 and 6 is questionable given previous studies on the effect of DNA accessibility and transcription factor binding sites on genomic patterns of mutation.

Other points

The paper refers to the 'paradigm' that lesion bypass

The manuscript assumes a close knowledge of the authors previous work and there are several points that will be unclear to a more general reader.

- 'lesion segregation' is mentioned on line 40. This term is not intuitive at the best of times, although I appreciate the authors have used it before, but here it certainly needs explanation.

- DEN is used on line 54 but is not explained or introduced. Many readers will wonder what a DEN-induced tumour is.

- the heading (p10) 'Transcription coupled repair is stochastic' is not really very informative to a general reader.

Author Rebuttals to Initial Comments:

We thank the reviewers for their careful reading and constructive review of our manuscript. Guided by the referee feedback we have substantially revised this manuscript by including new experimental data, revised and extended analyses, and improved presentation for greater impact and clarity. We think the updated manuscript fully addresses all points raised by the referees.

Referee #1

#1.1 This is a fantastic study, which highlights the power of genomics if applied to the right system. It significantly extends the previous paper giving detailed insights into the mechanism of replication, damage tolerance and repair and how those processes are associated with mutagenesis. I therefore only have limited experimental points, which I suggest to be addressed before publication. My main criticism concerns the presentation of the manuscript, which in my eyes needs improvement.

We thank the referee for their resounding support, recognising the power of this system and the importance of our findings. We have extensively edited the manuscript for flow and clarity, revising the introduction to better introduce the system and removing or compressing multiple results sections that were not central to the main thread of connected results. Specifically we have removed the “Transcription coupled repair is stochastic” section that was tangential to the main themes and have substantially reduced and refocused the “Steric influences on DNA damage and repair” section.

#1.2 Fig. 1f – Why not show this analysis for both forward and reverse strand lesions (similar to Fig. 1 e, can be colour-coded)? Does “oriented so lesions on the reverse strand” mean forward lesions are included as well? If not, they should be.

The reviewer is correct, “oriented so lesions on the reverse strand” does mean that forward strand lesions are included as well. What matters for these strand-resolved analyses is whether the lesion containing strand is primarily the template for leading strand replication or lagging strand replication.

Preserving separate analyses for both forward strand and reverse strand lesions throughout the manuscript would substantially complicate figures (doubling the number of curves and data-points) and reduce statistical power. To avoid confusion we now define the metric replication strand bias (RSB) and use that through the main figures of the manuscript. In Extended Data Fig. 3 we demonstrate how replication strand bias is derived from the forward and reverse strand replication fork direction (RFD) measures (compare Extended Data Fig. 3 panels d and f). The text of the manuscript has also been updated to discuss mutation rates with respect to replication strand bias rather than RFD. We think these changes will help readers follow the work as replication strand bias is a far more precise description of the measurement being made.

#1.3 Fig. 2 – Should the ordering of upstream and downstream lesions not be done only for those regions that have a strong bias in fork direction? If this was done, I did not find the according information.

For the purposes of orienting upstream and downstream mutations, the direction of the replication fork is not relevant when the lesion strand is resolved as in this study: DNA synthesis always extends in the same direction (5'→3') regardless of leading or lagging strand. The schematic in Fig. 2d graphically illustrates this and we have expanded the corresponding methods section to explain in more detail:

“For each mutation cluster, if it was located within a lesion segregation mutation asymmetry segment, we annotated the mutations within the cluster with respect to the inferred lesion containing strand. For a genomic segment containing reverse-strand lesions, the leftmost mutation site would be the first used as a template for an extending DNA polymerase (as DNA synthesis extends 5’->3’), and the rightmost mutation site replicated over subsequently. These orientations are reversed for a genomic segment containing forward-strand lesions.”

We now guide the reader to the methods section in the Fig. 2 legend for the expanded explanation.

#1.4 Personally, it took me a long time to get through the manuscript and to understand data and conclusions. While undeniably there is a level of complexity in this study that cannot be changed – I think the authors need to improve presentation for the broad audience of Nature.

We agree that the originally submitted manuscript was dense with results, and that we tried to convey too many findings and narratives in a single manuscript. Taking into consideration the wider suggestions from the three reviewers, we have streamlined our resubmission by: (i) Removing the section titled “Transcription coupled repair is stochastic”, because these findings are tangential to the main themes of the manuscript and do not rely on the replication-strand analysis or insights from multiallelic variation, and (ii) reducing the text and main figure in the “Steric influences on DNA damage and repair” section of the manuscript. These analyses do fit with the overall narrative and depend on multiallelic variation, but the restructuring better presents that differential repair through DNA accessibility (rather than differential damage) is primarily responsible for shaping the mutational landscape.

Furthermore, we have more generally revised the figures and text to improve accessibility, clarity, and impact for the broad readership of Nature and, as noted in response to reviewer comment #1.6, we have restructured the Discussion. The result is a substantially shorter manuscript that concisely focuses on four key findings: (i) the unexpected replication strand symmetry of DNA damage tolerance, (ii) collateral mutagenesis from translesion synthesis, (iii) the utility of multiallelism to systematically disentangle differential repair from heterogeneous damage, and (iv) the mechanistic basis for mutagenic nucleotide excision repair.

#1.5 This includes (i) clarifying the experimental systems (e.g. DEN was not explained throughout the MS), (ii) clarifying the analysis (the MS sometimes contains phrases such as “after accounting for TCR” which are ambiguous; or mutation rates given as “nascent transcripts per million”), (iii) including whenever possible graphical illustrations (those included are very useful, but more would be desired).

(i) In the final two paragraphs of the introduction, we now describe diethylnitrosamine (DEN) and provide more context to explain the model system, the mechanism of hepatic specificity, the type of DNA damage, and how it relates to common and clinically relevant mutagenic exposures.

(ii) We have consulted with trial readers to improve readability and we now better explain terms where there was potential ambiguity. For instance, the nascent transcripts per million measure quoted by the referee is now explicitly defined as a measure of expression:

“Increased transcription decreases the mutation rate for template strand lesions up to an expression level of 10 nascent transcripts per million (Fig. 3b). Beyond this, the mutation rate plateaus and is not further reduced by additional transcription, suggesting that the remaining mutagenic lesions are largely invisible to TCR (Extended Data Fig. 6c,d).”

(iii) We have added further schematic panels to existing figures (Fig. 1b, Fig. 2g), incorporated additional schematic panels in newly added figures (Extended Data Figs. 2b, 5a, 5b, 5c, 5e), have added annotations to figures (Fig. 1a, Fig. 4d, and Extended Data Figs. 4d, 4g) using a consistent colour palette to further aid the interpretation of our results.

#1.6 In contrast to the previous study, this study is not about how DNA damage by nitrosamines causes liver cancer, but essentially it is about how cells deal with and repair those lesions at different places in their genome. Indeed, this may be the most powerful model in the mammalian system. While I generally find the authors conclusions convincing, they are very much based on deduction. Therefore, I think the authors need to put their story better in the context of the published literature including biochemical studies and experiments in model organisms. In this sense I would consider re-writing the discussion chapter. While I enjoyed reading this chapter and found that it nicely highlighted the brilliant “detective work” done in this study, I would have wished for a bit broader discussion of the key findings (e.g. a key result is mutagenic NER – it should therefore be discussed in more detail what others evidence exist about NER being mutagenic).

We very much appreciate the referee recognising the power of this system to understand the mechanisms of DNA damage and repair. As requested, we have substantially rewritten the discussion section, expanding on the wider context of the key findings: the replicative symmetry of damage tolerance for alkylation adducts of DNA, a counterpoint to existing models based substantially on UV damage; the context and implications of collateral mutagenesis from translesion synthesis; the utility of multiallelic variation to disentangle differential damage from differential repair; and the mechanism and consequences of mutagenic NER. In particular, this restructuring has allowed us to relate recent findings from bacterial mutagenesis and link them with mammalian biology, illustrating the deep conservation and highlighting the broad applicability of our results. Inspired by reviewer comments we have also extended our analysis of collateral mutagenesis, which in the revised discussion allows us to link our findings with *in vitro* biochemical studies of translesion synthesis.

Referee #2

In this manuscript, Anderson et al. provide a natural continuation of their prior work (Aitken et al., Nature, 2020) on chromosome-scale phasing of somatic mutations. In the current paper, the authors leverage strand-phased mutation patterns and multiallelic variation to better understand the strand asymmetry of replication and transcription. Indeed, the analyses reveal several unexpected and surprising findings, including the undistinguishable DNA repair fidelity of the leading and lagging replication strands as well as the high DNA damage tolerance observed during transcription.

We greatly value the constructive feedback of the referee, and their recognition of the novelty and value of the findings we present. Implementing their recommendations has strengthened our analyses and subsequent conclusions.

#2.1 First, biologically, the manuscript is written as finding a general result related to DNA damage and repair of all types of small DNA adducts. However, the authors are using a N-nitrosodiethylamine (DEN) mouse model which is a very specific mutagen. Perhaps, the presented results are correct for DEN (and maybe for many nitrosamines) but I am not sure whether this is the case for other small DNA adducts. For example, I am not sure whether their results with generalize for small alkylated bases such as 7-methylguanine. There is just not enough data to support the results as generalizable for other small DNA adducts. If the authors want to make this point, they should explore at least two other distinct classes of small DNA adducts. Otherwise, their results should be reported as specific for DEN.

In hindsight we agree with the referee that our results were presented as overly-generalised. To address this, the results from DEN mutagenesis are now presented as a counterpoint to the established models that are largely based on studies of UV induced intrastrand crosslinks (pyrimidine dimers). We have removed reference to the dichotomy of small versus large adducts because we agree that this is not a safe generalisation when we only have good evidence for two mutagens (DEN and UV).

The final paragraph of the introduction now makes the case that the multiple alkylation adducts generated by DEN overlap with those of other environmentally and medically relevant mutagens and behave in a similar manner. This establishes the importance of understanding the mechanisms of repair, tolerance and mutagenesis of small alkylation adducts generated by DEN and other common mutagens:

“The range of mutagenic alkylation adducts generated by activated DEN overlaps those from tobacco smoke exposure, unavoidable endogenous mutagens, and alkylating chemotherapeutics such as temozolomide (Singer 1985; Chen, Wang, and Lin 2012; Fu, Calvo, and Samson 2012). More generally, the mechanism of lesion segregation which the strand-resolved analysis relies on appears to be a ubiquitous property of base damaging mutagens (Aitken et al. 2020).”

The discussion now makes the direct comparison between our new findings and prior models based substantially on UV damage, highlighting that while replication strand asymmetric lesion tolerance has been widely anticipated and shown for UV, considerable statistical power is required to demonstrate an absence of asymmetry, and this is uniquely what our study provides:

“It has long been expected that the asymmetry of leading and lagging strand replication would lead to asymmetric replication fidelity on damaged DNA (Meneghini, Cordeiro-Stone, and Schumacher 1981; Yeeles et al. 2013; Gabbai, Yeeles, and Marians 2014; Hedglin and Benkovic 2017), and analysis of UV induced mutation patterns supports that expectation (Haradhvala et al. 2016; Seplyarskiy et al. 2019). However, with over 7.2×10^6 lesion strand resolved mutations and cell type matched measures of replication strand bias, our system is uniquely powered to question the generality of this model. Contrary to expectation, we find a remarkable symmetry of mutation rate for leading and lagging strand replication. Matched patterns of collateral mutagenesis - proximal downstream mutations thought to arise from continued synthesis by translesion (TLS) polymerases (Póti et al. 2022) - point to the recruitment of identical TLS polymerases for the bypass of small alkylation adducts on both replication strands.”

In combined response to this query and to Referee #3 (#3.2) we have also considered the question of replication strand mutational symmetry/asymmetry for a wider collection of mutagenic exposures in which we can strand-phase the lesions (see response to #3.2 for figure and methodological detail). We

do find nominally significant evidence of replication strand asymmetric mutagenesis following UV exposure, supporting prior work (Haradhvala et al. 2016; Seplyarskiy et al. 2019) but do not find replication strand asymmetry for the majority of exposures (11/13) - consistent with our DEN findings. However, by downsampling our own data we provide evidence that these additional analyses are likely under-powered to confidently conclude the absence of replication strand mutation asymmetry. For context our DEN model system has 1,183 times the number of lesion strand resolved mutations than are publicly available for any other mutagen ($n=7.2 \times 10^6$ in this study, $n=6,086$ for DBADE (Kucab et al. 2019)). Thus, our data and model system are uniquely well powered to demonstrate an **absence** of replication strand asymmetry.

#2.2 Second, some of the findings reported as novel have been previously shown and parts of the figures are quite similar to ones from the authors' previous manuscripts. For example, figure 1a-b in the current manuscript is almost identical to figures from their previous manuscript (Aitken et al., Nature, 2020). Similarly, the finding that mutations are largely shaped by the influence of DNA accessibility has been previously reported and the some of the authors on this study have reviewed it (Gonzalez-Perez et al., Cell, 2019).

Our original panels a and b in Figure 1 were intentionally styled to match (Aitken et al. 2020), in order to provide continuity for readers who are familiar with our previous work and help those cross-referencing to it. Additionally we recognise that it is important to not assume prior knowledge of these relatively new concepts, and sufficiently introduce them to the wider readership as highlighted by Reviewer #3 (#3.10.1). We have retained a modified version of Fig. 1a that concisely summarises the key concepts of lesion segregation (Aitken et al. 2020) that the current manuscript builds on. Complementing this, Extended Data Fig. 1 has been added to graphically illustrate the mutational asymmetries of these tumours and address reviewer point #3.3.

With regard to the DNA accessibility findings: It was not our intention to claim the discovery that mutation patterns reflect DNA accessibility, indeed the second sentence of that paragraph stated "*Accessibility is also influenced by nucleosome positioning and transcription factor binding, both of which have been shown to broadly influence mutation patterns (Reijns et al. 2015; Kaiser et al. 2016; Pich et al. 2018; Mao et al. 2019; Afek et al. 2020)*". Instead our intention was to show that the integration of multiallelic variation with mutation rate, our novel analysis approach that is first introduced in this manuscript, can be utilised to disentangle differential DNA damage from differential repair. This highlights that at multiple scales and in diverse contexts, DNA accessibility influences mutation rate, and identifies important exceptions.

We have now moved many of the main figure panels from the DNA accessibility section into Extended Data Figures. Correspondingly, we have re-written the "*Steric influences on DNA damage and repair*" section to be more concise and to better showcase the finding that correlation of mutation rate with accessibility represents differential repair rather than differential damage, and that this insight is made possible due to the use of multiallelic variation.

#2.3 Third, the manuscript makes a number of mechanistic claims based on the computational analyses. For example, the authors speculate that during the repair of small DNA adducts, the same translesion polymerases are recruited by both the leading and lagging strands replication machineries. Indeed, this is reasonable to infer from the performed analysis, but it is never experimentally validated. Further, this is not the only explanation of the observed data. For example, different translesion polymerases can be

recruited to the leading and lagging strands provided that they have similar fidelity of repair on these strands. As such, parts of the manuscript report interesting biological models but they do lack any mechanistic or experimental validation.

We have revised each of our major findings to more clearly present orthogonal and complementary lines of evidence that support our conclusions. As we demonstrate, the application of multifaceted and orthogonal analyses does allow hypothesis testing and can do so without the confounding effects of perturbation experiments.

Taking the reviewer's detailed example first: "Mechanisms of translesion repair on leading and lagging strand". We agree that we cannot definitively state which translesion polymerase - or polymerases - are responsible for the mutagenic translesion replication over DEN adducts and the associated collateral mutagenesis. We do however convincingly demonstrate that the (i) rate, (ii) mutation spectrum and (iii) tract-length of translesion synthesis is symmetrical between replication strands. This is in contrast to the apparent expectation that they should differ, based on previous characterisation of mutagens with alternate biophysical properties, such as helix-distorting pyrimidine adducts (Seplyarskiy et al. 2019). Further extending our analyses, as suggested by reviewers, identified different collateral mutagenesis signatures associated with either misincorporation or -1 frameshifting at probable lesion sites (Fig. 2e-g), providing evidence for alternate outcomes from the recruitment of different translesion polymerases.

We would also highlight the finding of "translesion resynthesis induced mutagenesis (TRIM)", where we have applied five orthogonal computational approaches to our multi-omic experimental datasets. First, we propose the mechanism "Nucleotide excision repair is mutagenic" and provide initial lines of evidence pointing to the existence of TRIM. We detail the (i) consistent elevation of apparent A->N mutations in situations where by all other measures, NER is more active; (ii) low multi-allelic variation of A->N mutations; and importantly (iii) the rate of A->N mutations reaching a plateau at higher expression levels that mirrors the saturation of transcription coupled repair. Second, we introduce a distinct prediction from the NER-TRIM model that is explicitly tested and confirmed in the subset of our mutationally symmetric "twin sister" tumours: (iv) Exceptionally high variant allele frequency of somatic mutations in highly expressed genes specifically in "mutationally symmetric" tumours, and (v) exceptionally high variant allele frequency of driver mutations, specifically in mutationally symmetric tumours. This demonstrates the remarkable propensity for NER-TRIM mutagenesis to drive oncogenic transformation.

In addition, we have substantially re-written other results sections to better articulate the complementary approaches taken and the conclusions drawn, for example detailing "Collateral mutagenesis" and "Transcription associated repair". As suggested by Reviewer #1 (#1.6), for each of these findings we have re-written the Discussion section to better explain their context with the established literature.

#2.4 The somatic mutations are taken from mouse liver cancers. However, replication fork directionality was determined based on mouse primary splenic B cells. Also, the data for B cell was generated using a different reference genome which required the authors to lift it over to the C3H_HeJ_v1 reference assembly. The authors should generate Repli-Seq for some of the tumors (if material is available) or, at least, for mouse hepatocytes as replication fork directionality in hepatocytes could differ from the one observed in B cells.

Following this important suggestion, in collaboration with the developers of E/L Repli-seq, we have now generated replication data for a mouse hepatocyte derived cell line (Hepa1-6; Extended Data Fig. 2). As the reviewer speculates, we do see differences in replication time and replication fork direction profile between cell types. These differences are pronounced between Hepa1-6 and mouse embryonic stem cells (Extended Data Fig. 2a) and more subtle, but still evident, between Hepa1-6 and the splenic B-cell OK-seq data used in our prior submission (Extended Data Fig. 2c).

However, although multiple previous high profile papers have used Repli-seq with no or very approximate matching between cell types to infer replication time or replication strand (Haradhvala et al. 2016; Seplyarskiy et al. 2019; Alexandrov et al. 2020; Kucab et al. 2019), that approach has not been independently validated for replication strand inference. In contrast to Repli-seq, OK-seq directly measures the ratio of leading:lagging strand synthesis across the genome and does so at higher resolution (~1kb versus ~50kb), but is not tractable on many cell types as it is a highly specialist technique requiring very large populations of synchronous or isolated S-phase cells.

To obtain the benefit of both high resolution direct measurement of RFD from OK-seq and cell-type matching with Hepa1-6 Repli-Seq, in our revised manuscript we have restricted replication strand (updated) analysis to genomic regions that were concordant in RFD between these two approaches (Fig. 1; Extended Data Figs. 2d, 3).

For all of these revised analyses, Repli-Seq and OK-seq FASTQ level data was aligned directly to the corresponding reference genome (C3H_HeJ_v1 for mouse analysis, GRCh37 for human RFD validation), removing the need for genome-alignment based coordinate transforms (liftover), as suggested by the reviewer.

All replication strand based analyses of the original submission have been replaced by new analyses based on the high-quality concordant-RFD measures. The previously noted directional correlation of liver expressed genes with leading strand replication was substantially strengthened in this concordant-RFD data, and an increase in mutation rate around RFD=0, subtly evident in the previous analysis, resolved as a clear signal in the updated analysis (Fig. 1). Further analysis and detailed exploration of replication time effects facilitated by the Hepa1-6 Repli-Seq data allowed us to demonstrate that all apparent effects of replication strand bias (RSB) on DEN-induced mutation rate can be attributed to transcription and, to a lesser extent, replication time (Fig. 1; Extended Data Fig. 3). This is consistent with, and extends our previous conclusion, and by demonstrating replication time effects also addresses reviewer point #3.7. Both the results text and discussion have been updated accordingly, the new data and validation of approach is shown in the new Extended Data Fig. 2 included below for convenience.

Extended Data Fig. 2 | Quantifying replication fork directionality. **a**, Replication time profile of an example 15 Mb of C3H genome chromosome 8 (x-axis, shared with panel **c**). Curves show early/late (EL) replication relative enrichment (E and L read counts normalised to their respective library read depth, then relative enrichment, $RE = (E-L)/(E+L)$) where more positive values indicate earlier replication and more negative values indicate later replication. Replication profiles shown for a mouse embryonic stem cell line (E14TG2a, tan) and mouse hepatocyte derived cell line (Hepa1-6, red). Blue dash line indicates the centre of a strong replication origin region (schematic) and is projected into panel **c** for comparison. **b**, Schematic illustrating two alternate strategies to generate replication fork directionality measures (RFD). Left side, E/L-Repli-seq (top) can be used to derive Repli-seq based replication fork RFD (repli-RFD; bottom). On the right side, Okazaki fragment sequencing based RFD (OK-RFD). **c**, Smoothed derivatives of Hepa1-6 E/L-Repli-seq data (red, panel **a**) provides an RFD estimate. Comparison to OK-seq data from another differentiated cell type (pink, activated B-cells) shows overall good concordance but captures some replication profile differences between cells (grey triangle). **d**, Kernel density plot with decile contours (grey) summarising the genome-wide correlation of B-cell derived OK-RFD (x-axis) and Hepa1-6 derived repli-RFD (y-axis), both at 10 kb resolution. Only high-concordance genomic intervals between blue stepped lines (21 quantile boundaries) were used for RFD based measures of liver tumour mutation rate. **e**, Validation of the E/L-Repli-seq to RFD measure in human RPE-1 cells where both OK-seq (grey) and E/L-Repli-seq (black) has been generated and used to calculate RFD. The curves are shown over a 15 Mb interval of human chromosome 8 and illustrate a high concordance of RFD profile. Although both traces are plotted at 10 kb resolution, the smoothing and processing required to calculate RFD from E/L-Repli-seq averages out some of the fine grained structure evident in the OK-seq derived profile. **f**, Kernel density plot summarising the OK-seq (x-axis) and E/L-Repli-seq (y-axis) RFD estimates for RPE-1 cells, as for panel **d**.

#2.5 In parts of the manuscript, the authors compared the observed mutation patterns to the expected ones. However, in most cases, it is not clear how these expected patterns have been generated. For example, figure 2a-b has an expected straight line without any confidence intervals. I am unsure how this expected value was derived. In contrast, figure 5a demonstrates extreme local distortions but it is unclear whether these are expected by chance due to the DNA strand sequence in accessible DNA regions or due to the actual repair of this sequence.

The expected patterns were described, but unfortunately rather deep in the methods section (original manuscript subsection: “Mutation clusters”, line 871). To improve clarity, we have added additional details to the corresponding figure legends (Extended Data Fig. 4), summarising the permutation of mutations across tumours to generate the expected distributions and direct the reader to the Methods section for expanded detail. We have added bootstrap 95% confidence intervals to both the “observed” and “expected” curves for plot Figure 2a to illustrate the statistical uncertainty of the estimates (although note that these are generally too small to visually resolve). For original plot Fig.2b (now Extended Data Fig. 4e) the “expected” curve is based on the naive expectation that two independently occurring mutations would co-occur phased on the same chromosome of a diploid pair (the conditional probability of either two heads or two tails in two consecutive flips of a fair coin: 50%, the rate we see the distribution decay to for more distant mutation pairs). Again this is now explained in the updated legend to Extended Data Fig. 4e.

Figure 5a as originally presented does show bootstrap 95% confidence intervals as a shaded area around the darker profile line, though with the ample power in our data those confidence intervals are narrow. Unfortunately we had not noted these confidence intervals in the plot or legend so we appreciate the reviewer flagging this oversight. The confidence intervals are now noted in the legend of the revised Figure 4, Extended Data Figs. 7 and 8.

#2.6 One of the more provocative findings in the paper was that, after accounting for transcription-coupled repair, there is no residual effect of replication strand on mutation rate. Again, it is quite unclear how the authors are accounting for transcription-coupled repair in regard to replication strand especially considering data from B cells was used to determine replication fork directionality. The authors should use better replication reference and they ought to focus on regions of the genome that are not transcribed to evaluate whether there is replication strand bias different than the one expected by chance. This result should be contrasted to regions of the genome that are commonly transcribed.

The joint stratification of the genome by replication strand (leading versus lagging) and potential transcription state (genic versus non-genic) proposed by the referee was originally presented in Fig.1f. We have now updated this analysis per the reviewer’s suggestion: (i) Now using newly generated cell-type matched concordant replication strand bias measures, shown in Extended Data Fig. 2 (see response to point #2.4). (ii) Contrasting non-genic with P15 mouse liver expressed genic regions (Fig. 1f).

We have further extended these analyses and included the new panel (Extended Data Fig. 3g) which shows that the mutation rate of replication strand bias stratified regions is almost perfectly (Pearson’s $\text{cor} = -0.97$ $p = 8.36 \times 10^{-14}$) explained by the fraction of sites transcribed using the lesion strand

template, but not by the replication strand bias (Fig. 3f). Additional multivariate regression analysis has been included to explore the independent contributions of replication strand, transcription strand and replication time on mutation rate (Extended Data Fig. 3k,l).

As noted in response to this Reviewer's point #2.4, all of our replication strand based analyses now use the newly generated, cell-type matched measures of replication strand bias. Implementing these suggested revisions has substantially strengthened the manuscript's prior conclusions, and further extended it to consider replication time effects.

Referee #3

#3.1 These are findings of potential broad interest. However, there are some significant limitations to the extent to which the data presented fully supports the generalisability of the conclusions. In addition, there are some statements that I think go beyond what is known or presented and some points that would need to be presented more clearly for a general reader.

We thank the reviewer for recognising the broad interest of this work and their helpful feedback. We have addressed the question of generalisability of these findings as described in detail in response to #3.2. We have also generated new experimental data, added substantial new analyses to support and, in some cases (e.g. collateral mutagenesis), significantly extended our initial findings. We have also revised the flow and clarity of the manuscript so that it is presented more clearly to the general reader. This includes a largely re-written discussion section, expanded use of summary schematics, making the "Steric influences on DNA damage and repair" section much more concise, and entirely removing the "Transcription coupled repair is stochastic" section of the manuscript which was tangential to the main narrative.

#3.2 My first point concerns the generalisability of the findings. The study is limited to a single DNA damaging agent, DEN. While I appreciate that the generation of this dataset represents an enormous investment, I believe that it is crucial to know to what extent the findings stand up when similar datasets generated with other mutagens are analysed e.g. those presented in Riva et al 202 (PMID 32989322). This is important as there are significant differences in the conclusions drawn here with regard strand specificity of lesion bypass-dependent mutagenesis to those in Ref 5 (Seplyarskiy et al. Nat. Gen. 2019), which focusses on UV-induced damage rather than an adduct. There is clearly diversity in the mutagenic responses, both in terms of enzymology and outcome, that are largely dictated by the mutagen and lesions it creates. Thus, drawing the sweeping conclusions that are made here from a study of DEN, especially when the results with UV appear quite distinct, seems unsafe.

We accept that as originally presented the manuscript made too broad a claim of generalisability for the replication strand symmetry finding. As noted above in response to comment #2.1 we have modified this presentation throughout the manuscript, now our narrative is not the dichotomisation of small versus large adducts, but counterpoint: there is good data that UV damage does have a pronounced replication strand effect and here we show that DEN damage does not. As we illustrate below, our DEN system is, to our knowledge, the only extant data in which the **absence** of replication

strand asymmetry can be adequately tested, due to limitations of model assumptions and lack of statistical power in other datasets.

The reviewer's proposal to look in other datasets is entirely reasonable. The ideal dataset in which to systematically explore replication strand biases in DNA damage tolerance would be the likes of the Compendium of Mutational Signatures (Kucab et al. 2019) that considered a diversity of mutagenic insults following a single burst of exposure. We have previously shown that for each of the human induced pluripotent cell-lines (iPSCs) in this experiment with >1,000 informative mutations there is significant mutation asymmetry from lesion segregation that can be used to phase the lesion strand. To address the reviewer's question we aggregated replicate data into the 13 distinct mutational exposures employing the same approach as for DEN induced tumours. Using human iPSC Repli-seq data (cell lines 7889SA3, H9, H1, WTC-11 from <https://data.4dnucleome.org>) intersected with OK-seq data (SRA:SRR7109016, SRR7109017), as described and validated in the updated manuscript (Extended Data Fig. 2), we partitioned the human genome into 21 quantile replication strand bias (RSB) bins. We also obtained human iPSC RNA-seq (GEO:GSE73211) for gene expression measures and repeated analysis of replication strand bias correlations with mutation rate both genome wide and stratifying into iPSC transcribed-genic and non-genic regions (Reviewer Fig. 1a). We find that almost no mutagenic exposures exhibit significant replication strand asymmetry. The exceptions are UV light (simulated solar radiation) and benzo[a]pyrene diol epoxide (BPDE) which, although nominally significant, are not robust to multiple testing corrections.

These results then are consistent with our DEN-based findings: most mutagenic DNA damage does not show evidence of replication strand mutational asymmetry, and suggests that UV light might be amongst rare outliers. However this is not a sound conclusion as it would be using the lack of statistical support to confirm a null hypothesis - to draw that conclusion we would additionally need to demonstrate that there is ample statistical power to detect some defined minimum level of replication strand asymmetry.

We can address this power question by down-sampling the DEN induced mutational data. Within transcribed genic regions of DEN induced tumours we do observe significant replication strand mutation rate asymmetries. We show this comes from the joint effects of transcription coupled repair and the correlation of transcriptional orientation with leading strand DNA synthesis (Fig. 1f; Extended Data Fig. edFigRepAsym). Using this genic-transcribed mutational asymmetry as a positive control for a significant effect, we down-sample the lesion strand resolved DEN mutations and ask what is the minimal number of such mutations required to robustly detect that asymmetry (Reviewer Fig. 1b). A downsampled set of 6,000 lesion strand resolved mutations is sufficient to nominally detect a significant asymmetry but is not robust to multiple testing correction. To detect such an effect in the Kucab study data (Kucab et al. 2019) we estimate 19,200 lesion strand resolved mutations would be required, the maximum available for an exposure in that large study is 6,086 mutations for DABDE. We conclude that other mutagenic exposures, outside of our DEN data, that can be lesion strand resolved are under-powered to detect replication strand mutational asymmetry. Replicating the Kucab study at several times (3 to 17) the scale of the original is not currently feasible.

The suggested (Riva et al. 2020) dataset could not be used because its data was generated by continuous exposure to mutagens, which results in multiple superimposed exposure patterns that progressively mask the mutational asymmetry used to resolve lesion strandedness. An alternate strategy to *de novo* lesion strand phasing could be to make assumptions regarding the known

biochemistry, for example assuming C->T mutations arise from UV photo adducts involving the C nucleotide rather than the G of the complementary strand. This strategy was used in the previous replication strand asymmetry work we cite (Seplyarskiy et al. 2019) but relies strongly on assumptions from biochemistry that have been shown to be confounded by atypical photoproducts (Vandenberg et al. 2023). The spectrum of mutagenic adducts formed by other mutagens is typically less well understood and therefore even more problematic.

Reviewer Fig. 1 | Statistical power estimation and evaluation of replication strand asymmetry for diverse DNA damaging mutagens. a, Reanalysis of the Compendium of Mutation Signatures (Kucab et al. 2019) data with lesion strand resolution (Aitken et al. 2020) and intersection with cell-type matched replication strand bias measures (described above). Mutagenic exposures are listed on the left with 3 separate odds-ratio measures (x-axis, log scale) calculated per exposure. Numbers to the right show the number of lesion strand resolved mutations per exposure dataset aggregated over replicates. Point estimates (circles) show the odds-ratio of mutation rates in the most extreme quantiles (1 and 21) of replication strand bias, compared against a null expectation. The null expectations are either equal rates (orange, brown) or the same rate difference between replication strands in genic-transcribed versus non-genic regions (black). Whiskers show 95% confidence intervals from 200 bootstrap replicates. Blue triangles indicate cases where the 95% confidence excludes log(odds)=0 though neither of these nominally significant rate differences are robust to multi-testing correction. A robustly significant deviation from log(odds)=0 for non-genic vs equal (orange) would indicate replication strand asymmetry in damage tolerance. Deviation from zero for genic transcribed versus equal (brown) could indicate replication or transcriptional strand bias because transcription strand correlates with replication strand. **b,** Odds ratio based quantification of replication strand mutation asymmetries in DEN induced tumours (top row) and down-sampled subsets of the full data as indicated by the number of lesion strand resolved mutations (y-axis). Calculated and plotted as for panel (a). The effect of transcription coupled repair through correlation with replication strand bias (Manuscript Fig. #repAsym.#) is tentatively evident with 6,000 lesion strand resolved mutations (blue arrows, black & brown points) but is not robustly found with datasets of <19,200 resolved mutations.

Prior to our current manuscript, there is only good evidence for replication strand asymmetry for the tolerance of and mutagenesis from a single type of DNA damage, UV (Haradhvala et al. 2016; Seplyarskiy et al. 2019). The mechanistic bases for those strand differences are not well understood.

Our paper shows that the situation differs for another type of DNA damage, base alkylation. This does not conflict with the UV findings but does provide evidence that there are at least two quite distinct pathways for damage tolerance during genome replication. Further than that, our analyses provide evidence for components of the strand symmetric pathway, for example similarity (probable parity) of translesion polymerases recruited and identical tract-length for translesion synthesis. We also reiterate that our data and model system is uniquely well powered to demonstrate an **absence** of replication strand asymmetry.

#3.3 The biology of DEN and the excision and bypass of its adducts is relatively poorly understood. It generates a very wide range of damage to both the phosphate backbone and bases. The authors argue that that 75% of DEN lesions occur at T, but DEN is also able to cause significant lesions at A (e.g. O4-ethylidA) and G (e.g. N7-ethylidG and O6-ethylidG) and C (e.g. O2-ethylidC and N3-ethylidC). Why is it only that T bases are persistent and able to cause persistent multiallelic variation?

The referee is correct that DEN causes damage to multiple bases and the DNA backbone (Singer 1985). As our readout is purely mutations rather than adducts, non-mutagenic damage will be invisible to our system.

Illustratively, we chose to focus on T lesions as they generate the most abundant mutations, it simplifies the narrative, and provides consistency between figures. As shown in the mutation spectrum plot, T bases show by far the highest rates of mutation from DEN damage, followed by C bases (Fig. 1c). All bases show mutation asymmetry, as illustrated in the new Extended Data Fig. 1. T, C and G all show appreciable rates of multiallelic variation demonstrating that lesions on each of these bases can persist for multiple cell generations. Except where otherwise stated (e.g. Fig. 5b; Extended Data Fig. 8a-d) all of the mutation rate and multiallelic rate estimates are based on the aggregate signal over all types of nucleotide. The first sentence of the corresponding methods section states: “*Mutation rates were calculated as 192 category vectors representing every possible single-nucleotide substitution conditioned on the identity of both the upstream and downstream nucleotides.*”

#3.4 The broad spectrum of base lesions caused by DEN is also relevant in thinking about mutations that appear downstream of what is inferred to be the reference lesion. These appear to largely be due to incorporation of A opposite template G, which is inferred to represent naturally unforced errors by a TLS polymerase, but could also represent A-rule bypass of e.g. an abasic site created by excision of a G lesion. Is so little mutation seen at G because it is so very efficiently excised? If so, what is the evidence for this? Such considerations are particularly important given the way in which the experiment was set up – induction of hepatic carcinoma with a single very high dose of DEN, which would be expected to produce a relatively dense distribution of lesions in the genome, increasing the chances of clustered mutations, which are indeed invoked by the authors as an explanation for mutagenesis associated with NER.

The A-rule bypass of DNA damage such as an adduct or abasic site could be generating C->T, G->T or A->T mutations with respect to the lesion containing strand across the genome, and that may be a significant contribution to the genome-wide mutation spectrum: C->T mutations are the most common mutations not involving mutation from T. A-rule bypass of T lesions would not generate a

mutation but would appear as accurate translesion synthesis over a damaged T, which we have previously provided strong evidence for in Aitken et al. 2020, (Aitken et al. 2020) in extended Data Fig.6. These effects are expected to be distributed over the genome and not specifically in spatially constrained clusters downstream of a mutation with typical DEN mutation spectrum and multiallelic variation.

As the referee correctly speculates, DEN adducts of G are efficiently repaired. G* (O6-ethG) lesions are rapidly repaired following DEN exposure (Connor et al. 2018), consistent with the existence of the known pathway for G* removal through non-replicative base repair by the enzyme MGMT (Armijo et al. 2023). The majority of tumours exhibiting low levels of G->N mutation genome-wide, implicating error free G* removal prior to genome replication. A subset of tumours exhibited a distinct mutation signature (denoted DEN2) that comprises G->N mutations, these are thought to represent tumours where the repair capacity of MGMT has been overwhelmed (the transfer of alkyl groups from guanine to a catalytic cystine residue on MGMT kills the catalytic activity, so one protein can only repair one damaged base). The direct reversal mechanism of MGMT does not lead to abasic sites (Christmann et al. 2011). Tumours with high DEN2 signature (>20% of mutations, (Aitken et al. 2020)) were excluded from analyses reported in this paper as we sought to focus on a single well defined mutation signature (DEN1).

The referees' more general point, that a recruited TLS polymerase may not be generating errors stochastically according to an intrinsic error profile, but doing so in collusion with other undefined variables is possible. However, TLS polymerases "by design" have reduced DNA replication fidelity and any colluding variables if randomly distributed would simply contribute to the mutation signature of the TLS polymerase in the same way that sequence context does. Colluding variables that are non-randomly distributed with respect to the replication strand would generate differences in mutation rate or spectrum between replication strands - the key finding is that they are demonstrably the same.

#3.5 The idea that some mutagenesis occurs during NER. This is perfectly plausible and is supported by their data. However, it is unclear whether it is due to clustered damage or simply that some TLS polymerases e.g. pol kappa are clearly able to carry out the gap filling reaction, which may provide an advantage in the low dNTP environment of G1 (Ogi & Lehmann Nat Cell Biol 2006). This should be discussed.

The evidence we present is not consistent with a simple explanation of low-fidelity gap-filling after the nucleotide excision of a lesion-containing single strand of DNA. In such a scenario, the variant allele frequency of NER-TRIM mutations would not be high in the symmetric tumours as observed (Fig. 5d,e). NER-TRIM predicts this increased allele frequency because the mutation was copied into both daughter lineages (Fig. 5a,c).

#3.6 Related to this, I was surprised that there was no analysis of indels in these datasets. This is relevant given the propensity for DEN to damage the phosphate backbone (see e.g. Li and Hecht (doi: 10.3390/ijms23094559)). In addition, Pol kappa, in particular, is associated with -1bp frameshifts during lesion bypass. It would be interesting to know whether such events are associated with the events under study.

This is an insightful and challenging question. Indels are a minor component of the DEN mutation signature (1.2% of mutations) and they are inherently difficult to resolve to a specific base and strand,

which is the core theme of this work. However, as the reviewer notes, it is reasonable to hypothesise that reduced fidelity TLS polymerases could have an elevated indel rate and as such they might be clustered with base substitutions in a similar manner to the collateral mutagenesis (cluster 3' base substitutions) we observe.

Comprehensively addressing this question has led to a series of important new findings around mammalian *in vivo* lesion bypass and translesion synthesis described in the results section (lines 188-199), illustrated (Fig. 2e-g; Extended Data Fig. 5) and considered in the discussion (lines 475-483). In brief we find exactly the -1 bp frameshift effect that the reviewer predicted, over likely lesion containing nucleotides (Ts on the lesion containing strand). Remarkably, we show that these 1 bp deletions correspond to a downstream base substitution signature that is clearly distinct from that of the base substitution mode of lesion bypass, arguing that the alternate substitution versus deletion outcomes reflect the recruitment of distinct combinations of TLS polymerases. We are extremely grateful to the reviewer for suggesting this analysis.

#3.7 The potential role of MMR is not adequately explored. In the context of replication-dependent strand asymmetric mutation, MMR at least in yeast, plays an important role in balancing out the differential mutagenesis of pol epsilon and pol delta (Lujan et al. PLoS Genetics 2012; Zhou et al. NSMB 2021). Further MMR is known to act on TLS-induced errors (see e.g. work of Niels de Wind). Indeed, does MMR have any role in direct recognition or processing of lesions created by DEN, or on the abasic sites created by excision?

This is a perceptive suggestion. We considered the role of NER by stratifying the genome into regions and strands where NER is expected to be more or less active (more active in open chromatin, template strand of transcribed genes). Analogously, MMR is known to be strongly biased towards earlier replicating regions of the genome. To address this reviewer question we have stratified the genome into 21 quantiles of replication time (RT) using the newly generated Hepa1-6 Repli-seq data, and further considered expressed genes and non-genic regions separately within these quantiles. The results presented in Extended Data Fig. 3j do show a correlation between mutation rate and replication time, supportive of mismatch repair, albeit to a much lesser extent than transcription coupled repair. We have applied multivariate regression analysis to quantify the relative contributions of replication time, transcription strand and replication strand to mutation rates (Extended Data Fig. 3k-l). These findings are considered in the corresponding results section (lines 126-130).

#3.8 The novelty of the data presented in Figures 5 and 6 is questionable given previous studies on the effect of DNA accessibility and transcription factor binding sites on genomic patterns of mutation.

It was not our intention to claim discovery of the mutation patterns around diverse protein binding sites and rather, as detailed in response to reviewer comment #2.2, we intended to show that in multiple scales these patterns are revealed by multiallelic variation analysis to be substantially driven by differential repair rather than differential damage. We have re-written this section to better showcase this core finding and to highlight the anomalous behaviour of apparent A->N mutations that are the subject of the subsequent NER-TRIM section. We have now consolidated the original Fig.5a,c and Fig.6 into a new Extended Data Fig. 7.

#3.9 The paper refers to the ‘paradigm’ that lesion bypass

We have revised this section of the discussion, which now reads:

“...our system, with over 7.2×10^6 lesion strand resolved mutations and cell type matched measures of replication strand bias, means we are uniquely powered to question the generality of this model.”

#3.10 The manuscript assumes a close knowledge of the authors previous work and there are several points that will be unclear to a more general reader.

We have revised the manuscript to make it accessible to the more general reader and ensure that familiarity with our previous work is not required to understand the analyses or conclusions drawn. This includes a better description of the DEN-induced model of liver tumours in the introduction (text included in response to #3.10.2) and a plain language introduction of lesion segregation (text included in response to #3.10.1). We have also included additional annotated schematics, as detailed in response to reviewer comment #1.5.

#3.10.1 ‘lesion segregation’ is mentioned on line 40. This term is not intuitive at the best of times, although I appreciate the authors have used it before, but here it certainly needs explanation.

We accept the reviewer’s point that we should clearly define the term in the introduction. The corresponding text now reads:

“The exposure results in mutagenic DNA base damage, referred to as DNA lesions, that are inherited and resolved as mutations in subsequent cell cycles (Aitken et al. 2020). This phenomenon of lesion segregation, in which damaged lesion-containing strands segregate into separate daughter cells, results in pronounced, chromosome scale mutational asymmetry. In a clonally expanded cell population, such as a tumour, this asymmetry can identify which damaged DNA strand was inherited by the ancestor of each tumour (Fig. 1a). Using this approach we can determine the lesion containing strand for approximately 50% of the autosomal genome and the entire X chromosome for each tumour (Extended Data Fig. 1) (Aitken et al. 2020).”

Additionally, an annotated schematic is provided in Fig. 1a, accompanied by the following text:

“Schematic of DNA lesion segregation on one haploid chromosome (Aitken et al. 2020). Mutagen exposure induces lesions (red triangles) on both DNA strands (forward blue, reverse gold). Lesions that persist until replication serve as a reduced fidelity template. The two sister chromatids segregate into distinct daughter cells, so new mutations are not shared between daughter cells of the first division. Since only one daughter lineage typically transforms and clonally expands, all damage-induced mutations in the tumour arise from lesions on only one of the originally damaged DNA strands.”

#3.10.2 DEN is used on line 54 but is not explained or introduced. Many readers will wonder what a DEN-induced tumour is.

We apologise for this omission. We now explain the tumour induction model system in the introduction. The corresponding text now reads:

“To understand the mechanistic asymmetries of DNA damage and repair on a genome-wide basis, we have exploited an established mouse model of liver carcinogenesis (Verna, Whysner, and Williams 1996; Connor et al. 2018), in which mutations are induced through a single DNA damaging exposure to diethylnitrosamine (DEN, an alkylating agent that is bioactivated by the hepatocyte expressed enzyme Cyp2e1).”

#3.10.3 the heading (p10) ‘Transcription coupled repair is stochastic’ is not really very informative to a general reader.

We agree with the referee. This whole section, though we think interesting and an important contribution to the community, does not fit well with the flow of the rest of the manuscript (other sections can be understood without it and it does not utilise multiallelic variation to draw its conclusions). Consequently we have removed this section from the current manuscript. This then tightens the focus and improves the narrative flow of the revised manuscript. Something that all referees have understandably requested.

References cited

- Aitken, Sarah J., Craig J. Anderson, Frances Connor, Oriol Pich, Vasavi Sundaram, Christine Feig, Tim F. Rayner, et al. 2020. “Pervasive Lesion Segregation Shapes Cancer Genome Evolution.” *Nature* 583 (7815): 265–70.
- Alexandrov, Ludmil B., Jaegil Kim, Nicholas J. Haradhvala, Mi Ni Huang, Alvin Wei Tian Ng, Yang Wu, Arnoud Boot, et al. 2020. “The Repertoire of Mutational Signatures in Human Cancer.” *Nature* 578 (7793): 94–101.
- Armijo, Amanda L., Pennapa Thongararm, Bogdan I. Fedeles, Judy Yau, Jennifer E. Kay, Joshua J. Corrigan, Marisa Chanchaoen, et al. 2023. “Molecular Origins of Mutational Spectra Produced by the Environmental Carcinogen N-Nitrosodimethylamine and SN1 Chemotherapeutic Agents.” *NAR Cancer* 5 (2): zcad015.
- Chen, Hauh-Jyun Candy, Yi-Ching Wang, and Wen-Peng Lin. 2012. “Analysis of Ethylated Thymidine Adducts in Human Leukocyte DNA by Stable Isotope Dilution Nanoflow Liquid Chromatography–Nanospray Ionization Tandem Mass Spectrometry.” *Analytical Chemistry* 84 (5): 2521–27.
- Christmann, Markus, Barbara Verbeek, Wynand P. Roos, and Bernd Kaina. 2011. “O6-Methylguanine-DNA Methyltransferase (MGMT) in Normal Tissues and Tumors: Enzyme Activity, Promoter Methylation and Immunohistochemistry.” *Biochimica et Biophysica Acta (BBA) - Reviews on Cancer* 1816 (2): 179–90.
- Connor, Frances, Tim F. Rayner, Sarah J. Aitken, Christine Feig, Margus Lukk, Javier Santoyo-Lopez, and Duncan T. Odom. 2018. “Mutational Landscape of a Chemically-Induced Mouse Model of Liver Cancer.” *Journal of Hepatology* 69 (4): 840–50.

- Fu, Dragony, Jennifer A. Calvo, and Leona D. Samson. 2012. "Balancing Repair and Tolerance of DNA Damage Caused by Alkylating Agents." *Nature Reviews. Cancer* 12 (2): 104–20.
- Gabbai, Carolina B., Joseph T. P. Yeeles, and Kenneth J. Marians. 2014. "Replisome-Mediated Translesion Synthesis and Leading Strand Template Lesion Skipping Are Competing Bypass Mechanisms." *The Journal of Biological Chemistry* 289 (47): 32811–23.
- Haradhvala, Nicholas J., Paz Polak, Petar Stojanov, Kyle R. Covington, Eve Shinbrot, Julian M. Hess, Esther Rheinbay, et al. 2016. "Mutational Strand Asymmetries in Cancer Genomes Reveal Mechanisms of DNA Damage and Repair." *Cell*, January. <https://doi.org/10.1016/j.cell.2015.12.050>.
- Hedglin, Mark, and Stephen J. Benkovic. 2017. "Eukaryotic Translesion DNA Synthesis on the Leading and Lagging Strands: Unique Detours around the Same Obstacle." *Chemical Reviews*, May. <https://doi.org/10.1021/acs.chemrev.7b00046>.
- Kucab, Jill E., Xueqing Zou, Sandro Morganella, Madeleine Joel, A. Scott Nanda, Eszter Nagy, Celine Gomez, et al. 2019. "A Compendium of Mutational Signatures of Environmental Agents." *Cell* 0 (0). <https://doi.org/10.1016/j.cell.2019.03.001>.
- Meneghini, R., M. Cordeiro-Stone, and R. I. Schumacher. 1981. "Size and Frequency of Gaps in Newly Synthesized DNA of Xeroderma Pigmentosum Human Cells Irradiated with Ultraviolet Light." *Biophysical Journal* 33 (1): 81–92.
- Póti, Ádám, Bernadett Szikriszt, Judit Zsuzsanna Gervai, Dan Chen, and Dávid Szüts. 2022. "Characterisation of the Spectrum and Genetic Dependence of Collateral Mutations Induced by Translesion DNA Synthesis." *PLoS Genetics* 18 (2): e1010051.
- Riva, Laura, Arun R. Pandiri, Yun Rose Li, Alastair Droop, James Hewinson, Michael A. Quail, Vivek Iyer, et al. 2020. "The Mutational Signature Profile of Known and Suspected Human Carcinogens in Mice." *Nature Genetics* 52 (11): 1189–97.
- Seplyarskiy, Vladimir B., Evgeny E. Akkuratov, Natalia Akkuratova, Maria A. Andrianova, Sergey I. Nikolaev, Georgii A. Bazykin, Igor Adameyko, and Shamil R. Sunyaev. 2019. "Error-Prone Bypass of DNA Lesions during Lagging-Strand Replication Is a Common Source of Germline and Cancer Mutations." *Nature Genetics* 51 (1): 36–41.
- Singer, B. 1985. "In Vivo Formation and Persistence of Modified Nucleosides Resulting from Alkylating Agents." *Environmental Health Perspectives* 62 (October): 41–48.
- Vandenberg, Brittany N., Marian F. Laughery, Cameron Cordero, Dalton Plummer, Debra Mitchell, Jordan Kreyenhagen, Fatimah Albaqshi, et al. 2023. "Contributions of Replicative and Translesion DNA Polymerases to Mutagenic Bypass of Canonical and Atypical UV Photoproducts." *Nature Communications* 14 (1): 2576.
- Verna, L., J. Whysner, and G. M. Williams. 1996. "N-Nitrosodiethylamine Mechanistic Data and Risk Assessment: Bioactivation, DNA-Adduct Formation, Mutagenicity, and Tumor Initiation." *Pharmacology & Therapeutics* 71 (1-2): 57–81.
- Yeeles, Joseph T. P., Jérôme Poli, Kenneth J. Marians, and Philippe Pasero. 2013. "Rescuing Stalled or Damaged Replication Forks." *Cold Spring Harbor Perspectives in Biology* 5 (5): a012815.

Reviewer Reports on the First Revision:

Referees' comments:

Referee #1 (Remarks to the Author):

I congratulate Taylor and colleagues to the revised version of their manuscript. They have more than adequately addressed my criticism and I find that rewriting and focusing have greatly improved the readability of the manuscript. It is now suited for the broad readership of Nature.

Referee #2 (Remarks to the Author):

The authors have performed substantial revisions of the manuscript and included additional experimental data and computational analyses.

From a technical perspective, the responses have clearly addressed my prior concerns about the Repli-Seq analysis by generating additional experimental data and by adding further computational analysis of these data. The authors have also answered my questions regarding the expected mutation patterns and clearly specified the 95% confidence intervals throughout the paper. Moreover, I find the newly submitted manuscript to be clearer and to have much better presentation.

Nevertheless, my concerns related to novelty and impact have not been addressed. First, this manuscript is very specifically focused on DEN mutagenesis which, as stated by the authors, has quite the unique properties:

"For context our DEN model system has 1,183 times the number of lesion strand resolved mutations than are publicly available for any other mutagen ($n=7.2 \times 10^{-6}$ in this study, $n=6,086$ for DBADE (Kucab et al. 2019)). Thus, our data and model system are uniquely well powered to demonstrate an absence of replication strand asymmetry."

While the utilized system may be "uniquely well powered", the reported results may also be unique for DEN. While the resubmitted manuscript has a more restrained message, it is unclear whether these observations will generalize to other mutagens despite the somewhat speculative paragraphs in Discussion. As per my prior review, the authors should explore multiple classes of mutagens to confirm the generalizability of their message.

Second, instead of performing additional experiments to confirm some of the proposed molecular mechanisms (as I suggested in my prior review) the authors have chosen to add additional computational examinations and, based on consistency with the prior literature, to logically argue certain molecular mechanisms. In this reviewer's opinion, experimental validation is required to prove at least some of the mechanistic claims made in the manuscript.

Referee #3 (Remarks to the Author):

Anderson et al.

Strand-resolved mutagenicity of DNA damage and repair

R1

The authors are to be commended on their thorough revision of this paper and for their detailed responses to the comments on the original version. I believe that the rewriting and focussing of the paper has significantly improved its reach and broad appeal. I am very pleased that some of my suggestions proved fruitful avenues for deeper exploration, particularly the analysis of indels and downstream mutagenesis, which leads to the interesting observation of distinct patterns of adjunct mutagenesis for index base substitution and indel mutations.

This is a very comprehensive, indeed tour de force, analysis that contains many points of general interest. It could be argued that fundamental issues remain: the focus on a very limited set of mutagen with the attendant risk of extrapolation of the conclusions to other lesion types, the lack of a single main conclusion and some overlap with existing studies. However, these points are not ever going to be readily addressable in any reasonable way. The authors' point about lack of power in other studies to draw the conclusions that they are able to draw is well taken. Overall, the work significantly advances the field of mutagenesis on a number of fronts, not least methodologically, and is beautifully performed and analysed. I feel that is deserving of publication as it stands.

Author Rebuttals to First Revision:

We thank the referees for their feedback, guidance and support for the work reported in this manuscript.

Referee #1

#1.1 I congratulate Taylor and colleagues to the revised version of their manuscript. They have more than adequately addressed my criticism and I find that rewriting and focusing have greatly improved the readability of the manuscript. It is now suited for the broad readership of Nature.

We agree that the rewriting and focussing of the revised manuscript, helped by all three referees, has substantially improved the accessibility of the manuscript. We share the view that the manuscript is now suited to the broad readership of Nature.

Referee #2

The authors have performed substantial revisions of the manuscript and included additional experimental data and computational analyses.

#2.1 From a technical perspective, the responses have clearly addressed my prior concerns about the Repli-Seq analysis by generating additional experimental data and by adding further computational analysis of these data. The authors have also answered my questions regarding the expected mutation patterns and clearly specified the 95% confidence intervals throughout the paper. Moreover, I find the newly submitted manuscript to be clearer and to have much better presentation.

We thank the reviewer for their previous guidance on streamlining and presenting the work. We agree that the updated presentation is much better suited to the broad audience of Nature.

We agree that we addressed the referee's concerns about the cell-type matching of replication strand bias measures. We have now gone further. The Hepa1-6 cell line used for Repli-Seq in the previous revision was well, but not precisely matched to the mouse strain or the DEN induced system. We have now replaced the Hepa1-6 data with new Hep74-3a repli-Seq data that is matched for mouse strain and derived from a DEN induced liver tumour, so is perfectly aligned to our experimental system, and shows improved quality control metrics in comparison to other E/L-repli-Seq data (Reviewer Fig. 1). The updated replication measures have been used for all replication based analyses throughout the revised manuscript. All conclusions remain the same.

Reviewer Fig. 1 | Updated quantification of replication fork directionality. **a**, Replication time profile of previous Hepa1-6 data (brown) and newly generated C3H strain matched Hep74-3a E/L-repli-seq data (red) showing a greater dynamic range more typical of other E/L-repli-seq data. **b**, Smoothed derivatives of Hep74-3a E/L-Repli-seq data (red, panel **a**) provides an RFD estimate. Comparison to OK-seq data from another differentiated cell type (pink, activated B-cells) shows overall good concordance but captures some replication profile differences between cells (grey triangle).

#2.2 Nevertheless, my concerns related to novelty and impact have not been addressed. First, this manuscript is very specifically focused on DEN mutagenesis which, as stated by the authors, has quite the unique properties:

"For context our DEN model system has 1,183 times the number of lesion strand resolved mutations than are publicly available for any other mutagen ($n=7.2 \times 10^{-6}$ in this study, $n=6,086$ for DBADE (Kucab et al. 2019)). Thus, our data and model system are uniquely well powered to demonstrate an absence of replication strand asymmetry."

While the utilized system may be "uniquely well powered", the reported results may also be unique for DEN. While the resubmitted manuscript has a more restrained message, it is unclear whether these observations will generalize to other mutagens despite the somewhat speculative paragraphs in Discussion. As per my prior review, the authors should explore multiple classes of mutagens to confirm the generalizability of their message.

We respectfully disagree with the reviewer on the unique properties of the DEN model system. While we are uniquely powered in this study, much of that power comes from the unprecedented scale of the experiment, including the whole genome sequencing of 371 DEN induced tumours (of which 237 tumours are the primary focus of this manuscript). This contrasts with typically 3-6 replicates of mutagen exposure in other animal or cell-line based studies. As we highlight in the manuscript, the range of mutagenic adducts generated by DEN overlaps a wide variety of environmental and clinically relevant mutagens. Further, the mutational asymmetry of lesion segregation is well illustrated for other (all tested) DNA damaging mutagens (Kucab et al. 2019; Aitken et al. 2020). The transcription coupled repair exhibited by DEN adducts is typical of other base-modifying mutagens (Alexandrov et al. 2020) and two recent studies have demonstrated multiallelic variation from persistent lesions that were not generated through DEN exposure (Campbell et al. 2023; Ginno et al. 2024) illustrating that the analysis of multiallelic variation we introduce in this manuscript will have broad applicability.

We propose including the “Statistical power estimation and evaluation of replication strand asymmetry for diverse DNA damaging mutagens” analysis presented in Reviewer Fig 1 of the previous rebuttal as a supplementary figure in the revised manuscript. This then highlights the statistical strength of the current work and shows that analysis of data from other mutagens is not inconsistent with our findings from DEN. It also represents a citable set of analyses that may inform the design of future mutagenesis studies.

#2.3 Second, instead of performing additional experiments to confirm some of the proposed molecular mechanisms (as I suggested in my prior review) the authors have chosen to add additional computational examinations and, based on consistency with the prior literature, to logically argue certain molecular mechanisms. In this reviewer’s opinion, experimental validation is required to prove at least some of the mechanistic claims made in the manuscript.

We did provide additional experiments, new data, and analyses during this review process. The referee requested “experimental validation” but did not propose any specific experiment in either the first or second round review. We are not aware of any additional, practical experiment that would usefully complement those already reported in the manuscript. Each of our key conclusions are supported by multiple lines of evidence, and we share the view of the other referees that these conclusions are robust and well evidenced.

Referee #2 provides one example of the mechanistic claims we have made, where they assert “experimental validation” is lacking:

“For example, the authors speculate that during the repair of small DNA adducts, the same translesion polymerases are recruited by both the leading and lagging strands replication machineries. Indeed, this is reasonable to infer from the performed analysis, but it is never experimentally validated.” (Round 1 review, point #2.3)

In our results section we clearly lay out 3 lines of evidence (mutation rate symmetry, collateral mutagenesis mutation rate and spectrum, and collateral mutagenesis tract length) that the leading and lagging strand replication over alkylation adducts is symmetric in terms of the mutations generated. This referee accepts that it is a reasonable inference from the analysis, that this reflects the recruitment of the same translesion polymerases to each strand. We frame this as an inference rather than

conclusion - “point[s] to the recruitment of identical TLS polymerases for the bypass of small alkylation adducts on both replication strands” but the referee requests experimental validation. This leading versus lagging strand contribution of specific polymerase is a question author Taylor has previously addressed for the main replicative polymerases (Reijns et al. 2015). This entailed modifying the active site of the polymerases in question, to alter nucleotide selectivity and identify consistent tracts of strand-asymmetric altered (ribo) nucleotide incorporation in synthesised DNA. Even if analogous modifications could be made to the translesion polymerases (unlikely) the short tract lengths of their synthesis at inconsistent locations between cells would make equivalent analysis impossible. Analysis based on knocking-out individual translesion polymerases would require mutational data acquisition on the same scale as the current 237 mouse tumour study for each knock-out, and would still not unequivocally identify synthesis from specific TLS polymerases, for example the loss of one TLS polymerase could influence the recruitment of another.

We are not averse to conducting well designed, appropriately powered and informative experiments that would significantly complement the study. This has been demonstrated with: (1.) the generation of extensive additional data during the review process and substantial reanalysis for its inclusion, (2.) the large scale of experiments this study is based on, and (3.) extensive experimental data reported for the first time with this manuscript. The questions we are addressing around the mechanisms of repair, damage tolerance and mutagenesis are, in our view, best addressed by well powered analysis studies where hypotheses can be directly tested by complementary strategies. The studies we build upon and whose insights we advance are also substantially of this design, e.g. (Haradhvala et al. 2016; Kucab et al. 2019; Seplyarskiy et al. 2019; Alexandrov et al. 2020), though the current manuscript is even better controlled and powered to address such mechanistic questions.

Referee #3

#3.1 The authors are to be commended on their thorough revision of this paper and for their detailed responses to the comments on the original version. I believe that the rewriting and focussing of the paper has significantly improved its reach and broad appeal. I am very pleased that some of my suggestions proved fruitful avenues for deeper exploration, particularly the analysis of indels and downstream mutagenesis, which leads to the interesting observation of distinct patterns of adjunct mutagenesis for index base substitution and indel mutations.

We agree that the revised manuscript is now more accessible and appealing to the broad readership of Nature. We again thank the referee for their important suggestions, especially for the indel analysis.

#3.2 This is a very comprehensive, indeed tour de force, analysis that contains many points of general interest. It could be argued that fundamental issues remain: the focus on a very limited set of mutagen with the attendant risk of extrapolation of the conclusions to other lesion types, the lack of a single main conclusion and some overlap with existing studies. However, these points are not ever going to be readily addressable in any reasonable way. The authors' point about lack of power in other studies to draw the conclusions that they are able to draw is well taken. Overall, the work significantly advances the field of mutagenesis on a number of fronts, not least methodologically, and is beautifully performed and analysed. I feel that is deserving of publication as it stands.

We agree with all of these comments, and their balanced and pragmatic summary that further experiments are beyond the reasonable scope of this manuscript. We especially appreciate the concluding sentence “Overall, the work significantly advances the field of mutagenesis on a number of fronts, not least methodologically, and is beautifully performed and analysed. I feel that is deserving of publication as it stands”.

References

- Aitken, Sarah J., Craig J. Anderson, Frances Connor, Oriol Pich, Vasavi Sundaram, Christine Feig, Tim F. Rayner, et al. 2020. “Pervasive Lesion Segregation Shapes Cancer Genome Evolution.” *Nature* 583 (7815): 265–70.
- Alexandrov, Ludmil B., Jaegil Kim, Nicholas J. Haradhvala, Mi Ni Huang, Alvin Wei Tian Ng, Yang Wu, Arnoud Boot, et al. 2020. “The Repertoire of Mutational Signatures in Human Cancer.” *Nature* 578 (7793): 94–101.
- Campbell, Peter, Michael Spencer Chapman, Emily Mitchell, Kenichi Yoshida, Nicholas Williams, Margarete Fabre, Anna Maria Ranzoni, et al. 2023. “Prolonged Persistence of Mutagenic DNA Lesions in Stem Cells.” *Research Square*. <https://doi.org/10.21203/rs.3.rs-3610927/v1>. Preprint.
- Ginno, Paul Adrian, Helena Borgers, Christina Ernst, A. Schneider, Mekaela Behm, Sarah J. Aitken, Martin S. Taylor, and Duncan T. Odom. 2024. “Single-Mitosis Dissection of Acute and Chronic DNA Mutagenesis and Repair.” *Nature Genetics*. In press
- Haradhvala, Nicholas J., Paz Polak, Petar Stojanov, Kyle R. Covington, Eve Shinbrot, Julian M. Hess, Esther Rheinbay, et al. 2016. “Mutational Strand Asymmetries in Cancer Genomes Reveal Mechanisms of DNA Damage and Repair.” *Cell*, <https://doi.org/10.1016/j.cell.2015.12.050>.
- Kucab, Jill E., Xueqing Zou, Sandro Morganella, Madeleine Joel, A. Scott Nanda, Eszter Nagy, Celine Gomez, et al. 2019. “A Compendium of Mutational Signatures of Environmental Agents.” *Cell*. <https://doi.org/10.1016/j.cell.2019.03.001>.
- Reijns, Martin A. M., Harriet Kemp, James Ding, Sophie Marion de Procé, Andrew P. Jackson, and Martin S. Taylor. 2015. “Lagging-Strand Replication Shapes the Mutational Landscape of the Genome.” *Nature* 518 (7540): 502–6.
- Seplyarskiy, Vladimir B., Evgeny E. Akkuratov, Natalia Akkuratova, Maria A. Andrianova, Sergey I. Nikolaev, Georgii A. Bazykin, Igor Adameyko, and Shamil R. Sunyaev. 2019. “Error-Prone Bypass of DNA Lesions during Lagging-Strand Replication Is a Common Source of Germline and Cancer Mutations.” *Nature Genetics* 51 (1): 36–41.

Author Rebuttals to Second Revision:

We thank the editor for seeking clarification from referee #2 regarding the outstanding point #2.3 (2nd round review) and we appreciate the referee providing their suggestions.

Addressing the outstanding point #2.3 (2nd round review)

Referee #2 clarification in full (grey):

I felt that the authors were making a very mechanistic claim regarding NER being mutagenic based specifically on translesion resynthesis purely based on associations. If this is indeed the case, they should see this effect disappearing if cells are exposed to DEN when either NER is knocked-out (i.e., XPC^{-/-}/CSB^{-/-} double mutant) or the Y-family DNA polymerases are knocked-out (i.e., POL η ^{-/-}, POL ι ^{-/-}, POL κ ^{-/-}). They could also do a rescue experiment with NER being first knocked-down and then reactivated in order to see the mutagenic effect disappearing and then re-appearing again

In all fairness, these experiments are quite tricky, and one needs to be very careful about off-target hits. Alternatively, the authors may want to reword their text to say that the observed associations are consistent with this specific model, but further experimental work is required to confirm these results.

We agree with the referee that the proposed experiments are tricky from several perspectives including scale, off-target concerns, and the interpretation of results. As the referee has suggested, we have reworded our text in the discussion to explicitly state that further experimental validation is required to confirm the involvement of NER.

Previous text:

“Building on evidence that transcription coupled NER can be mutagenic in bacteria (Carvajal-Garcia et al. 2023) and quiescent yeast (Kozmin and Jinks-Robertson 2013), we have now shown the existence of NER template resynthesis induced mutagenesis (NER-TRIM) *in vivo* in mammals and, additionally, show that it is not purely dependent on transcription.”

New text:

“Building on evidence that transcription coupled NER can be mutagenic in bacteria (Carvajal-Garcia et al. 2023) and quiescent yeast (Kozmin and Jinks-Robertson 2013), we present multiple orthogonal analyses supporting the conclusion that template resynthesis induced mutagenesis (TRIM) occurs *in vivo* in mammals, though confirming the involvement of NER requires further experimental validation. We also show that NER-TRIM is not purely dependent on transcription, but more generally results from the repair of lesions in close proximity, on opposite strands.”

Referees #1 and #3

There were not any outstanding matters from either referee #1 or referee #3.